# Endosomal Arl4A attenuates EGFR degradation by binding to the ESCRT-II component VPS36

Shin-Jin Lin [1,2], Ming-Chieh Lin [1,2], Tsai-Jung Liu [1,2], Yueh-Tso Tsai [1], Ming-Ting Tsai[1] & Fang-Jen S. Lee [1,2,3] ✉

Ligand-induced epidermal growth factor receptor (EGFR) endocytosis followed by endosomal EGFR signaling and lysosomal degradation plays important roles in controlling multiple biological processes. ADP-ribosylation factor (Arf)-like protein 4 A (Arl4A) functions at the plasma membrane to mediate cytoskeletal remodeling and cell migration, whereas its localization at endosomal compartments remains functionally unknown. Here, we report that Arl4A attenuates EGFR degradation by binding to the endosomal sorting complex required for transport (ESCRT)-II component VPS36. Arl4A plays a role in prolonging the duration of EGFR ubiquitinylation and deterring endocytosed EGFR transport from endosomes to lysosomes under EGF stimulation. Mechanistically, the Arl4A-VPS36 direct interaction stabilizes VPS36 and ESCRT-III association, affecting subsequent recruitment of deubiquitinating-enzyme USP8 by CHMP2A. Impaired Arl4A-VPS36 interaction enhances EGFR degradation and clearance of EGFR ubiquitinylation. Together, we discover that Arl4A negatively regulates EGFR degradation by binding to VPS36 and attenuating ESCRT-mediated late endosomal EGFR sorting.

Membrane trafficking plays a pivotal role in the transport of macromolecules to specific intracellular compartments and control of the molecular composition of cellular organelles, the functional regulation of which is essential for maintaining cellular homeostasis[1]. Among the various factors involved in these processes, small GTPases in the ADP-ribosylation factor (Arf) and Rab families have been identified as important regulators of membrane trafficking[2,3].

Arf and Arf-like (Arl) proteins function as major regulators of membrane trafficking in exocytotic, endocytic, and other pathways[3,4]. Arl4, of which there are three isoforms (Arl4A, Arl4C, and Arl4D) in vertebrates, can be distinguished from other Arf family members by the presence of a short basic extension at the C-terminus and a longer Ras-like interswitch region[5]. Arl4 expression is tissue-specific and developmentally regulated[6]. Arl4A expression is also increased in several cancer cells, such as colorectal SW620 cells, which exhibit an

aggressive metastatic phenotype[7]. Further studies have demonstrated that plasma membrane-localized Arl4A recruits its effectors to regulate actin cytoskeletal remodeling and cell migration[8–11]. Moreover, Arl4A complexes with GCC185 at the Golgi to regulate endosome-to-Golgi transport, while its extensive distribution was also found at early/late endosomes, and recycling endosomes, where the roles of Arl4A have not been elucidated[12].

Membrane receptors targeted for lysosomal degradation need to be distinguished from protein cargoes destined for other subcellular compartments, and the addition of ubiquitin was shown to mark these membrane receptors for lysosomal degradation[13]. In this regard, specific E3 ubiquitin ligases ubiquitinate membrane receptors and initiate endosomal sorting. These ubiquitinated membrane receptors are subsequently sorted into the intraluminal vesicles (ILVs) of endosomes, thereby forming multivesicular bodies (MVBs)[13,14], which

[1]Institute of Molecular Medicine, College of Medicine, National Taiwan University, 10002 Taipei, Taiwan. [2]Department of Medical Research, National Taiwan University Hospital, 10002 Taipei, Taiwan. [3]Center of Precision Medicine, College of Medicine, National Taiwan University, Taipei 10002, Taiwan. ✉e-mail: fangjen@ntu.edu.tw

eventually fuse with lysosomes, wherein the contents undergo lysosomal degradation. This ubiquitin-dependent protein sorting is coordinated by the endosomal sorting complex required for transport (ESCRT) machinery, which consists of four constituent complexes (ESCRT-0, I, II, and III) and several associated components. These ESCRT complexes act sequentially in cargo recognition and membrane sculpting processes, which involve a series of events characterized by a precise division of labor. ESCRT-I and ESCRT-II are responsible for membrane budding into the lumen of MVBs, the latter of which also participates in the recruitment of ESCRT-III, which contributes to membrane deformation[13–15].

ESCRT-II consists of VPS36, VPS22, and two VPS25 molecules[16]. The major role of ESCRT-II is to mediate the assembly of ESCRT-III components on the endosome surface where the cargo proteins are located[17]. Studies on *Arabidopsis* have indicated that VPS36 is involved in MVB biogenesis for the degradation of plasma membrane proteins[18,19]. VPS36 is important for the correct sorting of Toll-like receptor 3 (TLR3) into lysosomes[20] and has also been implicated in the ESCRT-dependent EGFR degradation pathway[21,22]. Knockdown of VPS22 in HeLa cells has been reported to delay EGF-induced EGFR degradation[21,23,24]. Furthermore, the ESCRT-II complex was required to prevent excess receptor signaling in *Drosophila*[25]. The accumulated findings of the studies conducted to date thus provide convincing evidence for the ESCRT-II-dependent degradation of EGFR; however, the detailed mechanisms have yet to be clarified.

It is well-documented that EGF stimulation induces the relocalization of EGFR to multivesicular compartments downstream of early endosomes, establishing an endosomal EGFR signaling center for oncogenic processing; these data support that EGFR signaling is not spatially restricted to the plasma membrane but also originates from late endocytic compartments[26,27]. In this study, we demonstrate that endosomal Arl4A acts with the ESCRT-II component VPS36 to attenuate EGF-induced EGFR degradation. We hypothesize that the Arl4A–VPS36 interaction provides a layer of control in maintaining the integrity of ESCRT-II for modulating EGFR sorting at late endosomes/MVBs, thereby negatively regulating EGFR degradation.

## Results

### Depletion of Arl4A enhances EGFR transport to endolysosomes

To determine the role of Arl4A in endosomal compartments, we first utilized EGF-Alexa-555 to track the movement of ligand-bound receptors. This approach allows monitoring of the lysosomal delivery of internalized EGF. We observed that depletion of Arl4A resulted in a significant decrease in EGF-Alexa-555 signals over time compared with those of the control cells (Fig. 1a, b). When we used an anti-EGFR external domain antibody to monitor the EGFR transport to endolysosomes, a similar result was observed in which Arl4A-depletion led to a rapid decrease in the EGFR signal (Supplementary Fig. 1). This finding prompted us to investigate whether Arl4A-depletion alters the trafficking of activated EGFR. Antibodies against EEA1 (early endosomes) and Lamp1 (late endosomes/lysosomes) were used to track the trafficking of internalized EGFR (Supplementary Fig. 2a, b). Compared with control cells, the internalized EGFR showed increased colocalization with EEA1 at 5 min but decreased colocalization at 15 min post-stimulation in the Arl4A-depleted cells (Supplementary Fig. 2a). Concomitantly, EGFR-Lamp1 colocalization quickly increased at 30 min but decreased at 45 and 60 min after Arl4A knockdown (Supplementary Fig. 2b). These indicate that a lack of Arl4A accelerates EGF triggered endolysosomal delivery and degradation of EGFR.

### Arl4A attenuates lysosome-mediated EGFR degradation

The cervical cancer cell line C33-A has been shown to exhibit a high level of Arl4A protein (Supplementary Fig. 3)[11]. To confirm Arl4A is involved in the lysosomal degradation of membrane receptors, we examined the protein levels of EGFR, hepatocyte growth factor

receptor (c-Met) and transferrin receptor (TfR) in Arl4A-depleted C33-A cells. Interestingly, we found that Arl4A depletion led to a significant decrease in EGFR and c-Met protein levels but not in TfR and $Na^+/K^+$ ATPase levels (Fig. 1c). Arl4A rescue, by expression of a short interfering RNA (siRNA)-resistant Arl4A (Arl4A$^{Res}$) could restored EGFR and c-Met protein levels in Arl4A-depleted cells, indicating that Arl4A specifically affects the protein levels of EGFR and c-Met (Fig. 1d). The decrease and rescue of EGFR was also observed using another Arl4A siRNA (siArl4A #2) (Supplementary Fig. 4a). Reproducibly, Arl4A depletion reduced EGFR in both HeLa and COS-7 cells, and c-Met protein levels in HeLa cells. Not all cells take up plasmids with transient transfection; thus, the result of the rescue experiment reached only 58% recovery (Supplementary Fig. 4b, c). Given that Arl4A expression is most abundant in adult testis and liver[6], we then determined the EGFR protein levels in the adult livers of Arl4A-knockout mice and found significant decreases in the protein level, but not the mRNA level, of EGFR (Supplementary Fig. 5a, b). Notably, the Arl4A-wild-type (WT)$^{Res}$ and the constitutive GDP-bound mutant Arl4-T51N$^{Res}$, but not the cytoplasmic myristoylation-deficient mutant Arl4A-G2A$^{Res}$, restored EGFR protein levels in the Arl4A-depleted cells (Supplementary Fig. 6). Because Arl4A-G2A$^{Res}$ is deficient for membrane binding, this result suggests that Arl4A sustained EGFR protein level relies on its membrane-binding ability while not being limited to its nucleotide-binding status.

We showed that Arl4A depletion did not downregulate EGFR mRNA levels, suggesting it results in increased protein degradation (Supplementary Figs. 4d and 5b). We further found that the lysosomal inhibitor $NH_4Cl$, but not the proteasome inhibitor MG132, reversed the decrease in EGFR protein levels in the Arl4A-depleted cells (Fig. 1e), indicating that Arl4A-depletion accelerates EGFR lysosomal degradation. Since the endogenous level of Arl4A in C33-A cells is high (Supplementary Fig. 3a) but low in HeLa cells, we knocked down Arl4A in C33-A cells and overexpressed it in HeLa cells. To confirm the role of Arl4A in ligand-induced receptor degradation, we treated cells with EGF and found that, compared with control cells, Arl4A-depletion significantly decreased the half-life of EGFR in C33-A cells (Fig. 1f and Supplementary Fig. 4e). Consistent with this observation, HeLa cells that express about three times more Arl4A protein than C33-A cells (Supplementary Fig. 4f), exhibit an increased EGFR half-life (Fig. 1g).

By examining phospho-MAPK ERK1/2, the hallmark of activated EGFR signaling, we observed that the phosphorylation peaks of EGFR and ERK1/2 were reduced upon Arl4A depletion in C33-A cells (Supplementary Fig. 7a, b). However, insignificant differences in p-EGFR/total EGFR ratio between control and Arl4A-depleted cells indicate that the decrease in p-EGFR level was mainly due to a decrease in EGFR protein level in Arl4A-depleted cells. EGFR depletion is known to inhibit cell growth and induce apoptosis[28,29]. We next examined the impact of Arl4A depletion on proliferation and apoptosis, respectively by 5-bromo-2-deoxyuridine (BrdU) incorporation assays and annexin V staining apoptosis assays. As shown in Supplementary Fig. 8a, b, Arl4A knockdown reduced C33-A cell proliferation in either the absence or presence of EGF stimulation. It also increased the number of apoptotic C33-A cells (Supplementary Fig. 8c) which accrued with cleaved PARP and caspase-7 (Supplementary Fig. 8d). These results indicate that Arl4A depletion inhibits proliferation and induces apoptosis. Though EGFR level is relatively low in C33A cells (Supplementary Fig. 3a), we evaluated the EGFR-dependent effect by treating cells with gefitinib (an EGFR inhibitor) and observed a decrease in cell growth in an inhibitor in a dose-dependent manner (Supplementary Fig. 9a). Further, the gefitinib inhibition of cell growth was exacerbated by concomitant Arl4A knockdown, with or without EGF stimulation (Supplementary Fig. 9b–d). These results suggest that aside from EGFR, Arl4A may modulate cell growth through other receptor-dependent signalings. Taken together, these data suggest that Arl4A plays a critical role in regulating EGFR degradation and signaling.

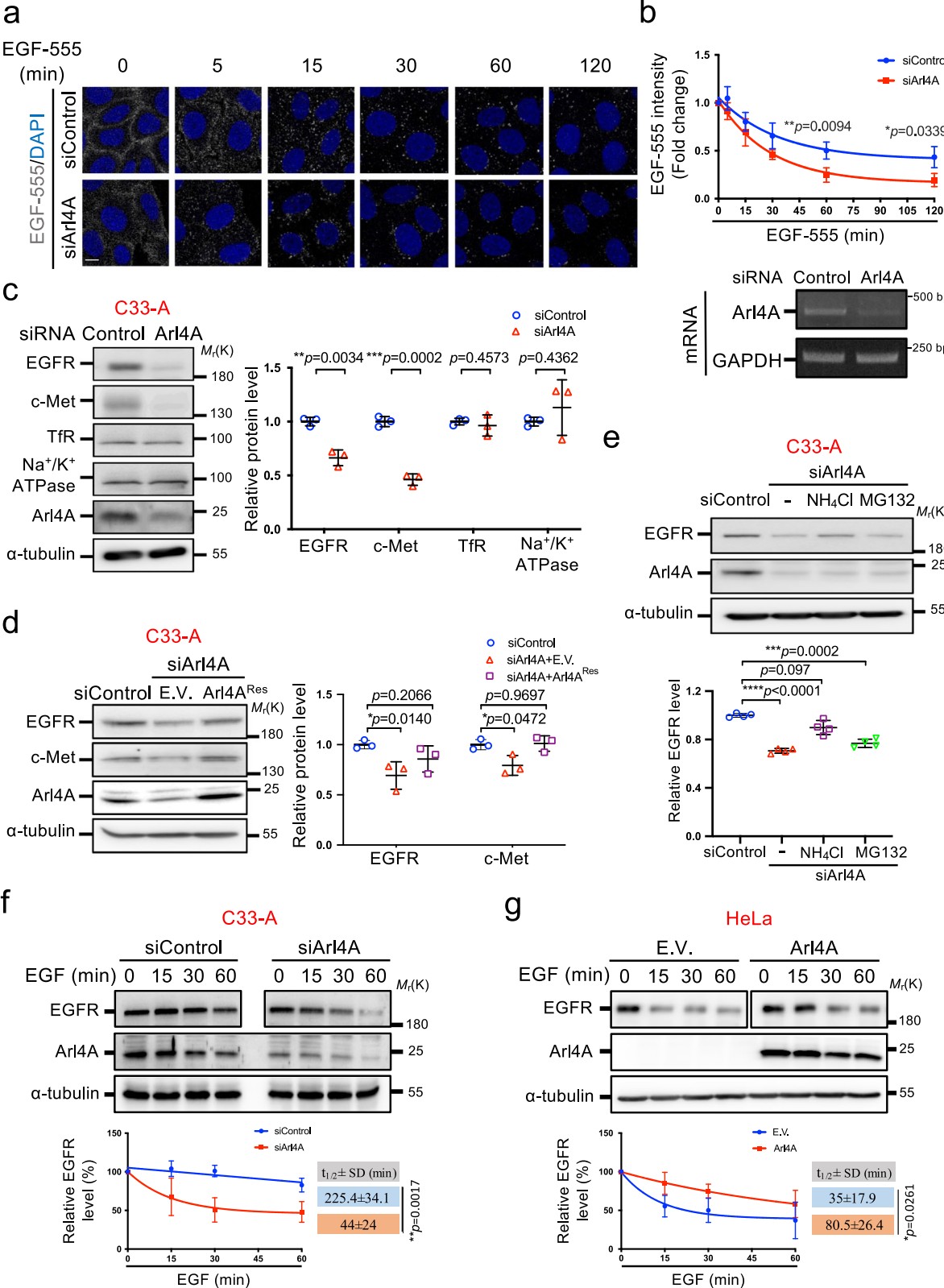

## Arl4A-depletion enhances EGFR mutant protein degradation

EGFR activation leads to multiple downstream signaling events that control various cellular processes, such as mitogenesis, cell survival, differentiation, and cell migration[30,31]. These processes are tightly regulated to prevent uncontrolled mitogenic signaling, cell survival, and cell migration, which may lead to tumorigenesis. Previous studies have shown that mutations in EGFR (such as L858R/T790M or ex19Del)

in non-small cell lung cancer (NSCLC) cell lines increase EGFR internalization and inhibit ligand-induced degradation[32,33]. We then tested Arl4A depletion effects on these EGFR mutants and observed that the protein levels of WT EGFR, EGFR[L858R/T790M] and EGFR[ex19Del] were all decreased in the Arl4A-depleted A549, H1975, and PE089 cell lines (Fig. 2a), whose c-Met protein levels were also decreased (Supplementary Fig. 4g). The half-lives of WT and mutant EGFRs were also

**Fig. 1 | Alr4A depletion promotes EGFR lysosomal degradation. a** HeLa cells transfected with the indicated siRNAs for 48 h were serum-starved and treated with EGF-Alexa Fluor 555 at a concentration of 80 ng/mL for the indicated times. Scale bar, 10 μm. **b** The EGF-Alexa-Fluor 555 intensity was determined using a nonlinear one-phase decay fit of the time course. The results represent the mean fluorescence intensity ± SD of three independent experiments and the *p*-values were assessed by a two-sided *t*-test. Arl4A-knockdown efficiency was assessed by RT-PCR. **c** C33-A cells were treated with the indicated siRNAs. The results represent the mean ± SD of three independent experiments and the *p*-values were assessed by a two-sided *t*-test. **d** The effects of Arl4A knockdown on EGFR protein levels were rescued by overexpression of Arl4A$^{Res}$. C33-A cells were treated with either the control or Arl4A siRNA as well as the indicated plasmids. Quantification of EGFR and c-Met levels is shown in the right panel. The results represent the mean ± SD of three independent

experiments and the *p*-values were assessed by one-way ANOVA with Tukey's test. **e** C33-A cells were treated with either the control or Arl4A siRNA and incubated with NH$_4$Cl (10 mM) or MG132 (10 mM) for 6 h. Quantification of EGFR levels is shown in the lower panel. The results represent the mean ± SD of four independent experiments and the *p*-values were assessed by one-way ANOVA with Tukey's test. **f** and **g** Arl4A knockdown and expression regulated the half-life of EGFR degradation upon EGF treatment. Arl4A was knocked down in C33A cells (**f**) and overexpressed in HeLa cells (**g**) with the indicated siRNAs and plasmids. The results represent the mean ± SD of three independent experiments. To compare the EGFR degradation rates, we adjusted the EGFR protein levels at time 0 in each group to be the same for western blot analysis. The $t_{1/2}$ ± SD of EGFR was obtained using a nonlinear one-phase decay fit of the time course; *p*-values were assessed by a two-sided *t*-test. Source data are provided as a Source Data file.

decreased upon EGF treatment (Fig. 2b–d). The Arl4A-knockdown efficiencies are shown in Supplementary Fig. 4h. We infer that the enhanced EGFR degradation following Arl4A-depletion is unrelated to the EGFR degradation resistance machinery elicited by these cancer-associated EGFR mutations.

We further tested whether Arl4A depletion accelerates EGFR transport to late endosomes in NSCLC cell lines. Consistent with previous studies, EGFR$^{L858R/T790M}$ and EGFR$^{ex19Del}$ already had higher colocalization with early endosomes than WT EGFR without EGF treatment (Supplementary Figs. 10–12a)[32,33]. Then Arl4A-depletion accelerated the delivery of EGFR to early endosomes after EGF treatment for 5 min in each of the cell lines examined. There was also an acceleration in EGFR transport to the late endosomes at 30 min in Arl4A-depleted cells (Supplementary Figs. 10–12b). Thus, we conclude that Arl4A also participates in endolysosomal delivery and degradation of cancer-associated EGFR mutants.

### Arl4A regulates EGFR ubiquitination levels and MVB sorting
Ligand-induced EGFR degradation is regulated by ubiquitination/deubiquitination of both EGFR and various sorting machinery constituents (ESCRT pathways)[34]. Ubiquitination of activated EGFR allows the receptor to engage with ESCRT machinery (ESCRT 0–III). The ESCRT-III machinery further deubiquitinates EGFR and sorts it into ILVs for degradation[35–37]. We examined whether the EGFR ubiquitination level is affected by Arl4A under EGF signaling. These experiments showed that Arl4A-knockdown in C33-A cells decreased the duration of EGFR ubiquitinylation with accelerated EGFR degradation (Fig. 3a). In contrast, Arl4A overexpression in HeLa cells increased the EGFR half-life and accumulated EGFR ubiquitination (Fig. 3b). Accumulating evidence suggests that ubiquitination serves as an essential sorting signal for endocytic trafficking of EGFR[38]. Therefore, we questioned whether Arl4A plays a role in delaying EGFR sorting into MVBs. Using the active Rab5 mutant Q79L to generate massive MVBs that enable high spatial resolution by super-resolution microscopy[39], we found that cells transfected with an empty vector displayed efficient translocation of EGFR to the MVBs represented by GFP-Rab5 Q79L after EGF stimulation. In contrast, Arl4A overexpression significantly suppressed EGFR sorting into the lumen of the MVBs (Fig. 3c). Together, we found that Arl4A regulates the duration of EGFR ubiquitinylation and sorting into the MVB lumen.

### VPS36 ESCRT-II component specifically interacts with Arl4A
To elucidate how Arl4A involves in ESCRT-driven EGFR degradation, we screened for putative Arl4A-interacting proteins using a yeast two-hybrid assay. We identified VPS36, a subunit of the ESCRT-II complex, which is required for MVB formation and the sorting of endosomal cargo proteins into MVBs, making it a valuable target for further analysis. Using the yeast two-hybrid assay, VPS36 (330-386) interacted specifically with WT, GTP-bound (Q79L), GDP-bound (T51N), and nucleotide-binding defective (T34N) mutants of Arl4A but not with Arl4C and Arl4D (Fig. 4a).

To confirm their direct binding, we further showed that VPS36 (201–386) interacts with Arl4A without nucleotide-binding preference in vitro using purified components (Fig. 4b). While in vivo co-immunoprecipitation in HeLa cells showed that, compared with Arl4A$^{WT}$ and Arl4A$^{Q79L}$, Arl4A$^{T51N}$ appeared to have a stronger interaction with VPS36 (Fig. 4c). Consistently, Arl4A$^{T51N}$ was most colocalized with VPS36 in HeLa cells by immunostaining (Fig. 4d). We found their interaction requires Arl4A membrane targeting for Arl4A G2A was unable to colocalize and be co-immunoprecipitated with VPS36 (Fig. 4d, Supplementary Fig. 13). These results reveal VPS36 as an interactor for Arl4A.

### EGF signals Arl4A GDP binding and Arl4A–VPS36 interaction
We observed that the GDP-bound form of Arl4A$^{T51N}$ exhibits increased colocalization with Lamp1 positive vesicles compared to Arl4A$^{WT}$ (Fig. 5a), suggesting that subcellular localization of Arl4A may depend upon different nucleotide-binding states. Due to Arl4A$^{T51N}$ endosomal preference, we next asked whether this could be triggered by EGF signaling. Interestingly, we observed that EGF stimulation elicited the translocation of Arl4A to Lamp1-positive vesicles, where Arl4A$^{T51N}$ is prone to localize (Fig. 5a). By assessing the GTP-bound levels of Arl4A using a PAK-PBD pulldown assay, we showed that GTP-bound Arl4A did decrease after EGF treatment within 15 min in HeLa cells (Fig. 5b), meanwhile, the Arl4A–VPS36 interaction was increased (Fig. 5c), These correlate with VPS36 binding and localization preference for GDP-locked Arl4A (Fig. 4c, d). Together, these data indicate that EGF signaling regulates the Arl4A nucleotide-binding state as well as its lysosomal distribution, and promotes the Arl4A–VPS36 interaction.

### Mutating VPS36 (352–356) region abates its binding to Arl4A
To dissect the Arl4A–VPS36 interaction site, we employed alanine scanning mutagenesis within the VPS36 (327–386) region. We found that the Ala-A3, A4, and A6 mutants of VPS36 showed decreased interactions with Arl4A in the yeast two-hybrid system (Fig. 6a). Using an in vitro GST pulldown assay, we also found that Arl4A interacted with GST-VPS36-WT (330–386) but not with the A3, A4, or A6 mutants (Fig. 6b). We aligned and found that the amino acid region $^{352}$LAKER$^{356}$ in which the A6 mutations reside was highly conserved among species (Fig. 6c). We therefore used the A6 mutant for further functional studies. The reduced co-immunoprecipitation of A6 with Arl4A was confirmed when comparing to WT VPS36 (Fig. 6d). Thus, we identified $^{352}$LAKER$^{356}$ of VPS36 as the critical region for the Arl4A–VPS36 interaction.

### Arl4A binding modulates the association of VPS36 with MVBs
A structural study showed that each ESCRT-II subunit contains two repeats of a WH domain that gather and contribute to the formation of the ESCRT-II complex. The residues ($^{352}$LAKER$^{356}$) required for Arl4A binding are in the VPS36 C-terminal WH domain that contacts VPS22. To examine whether Arl4A affects the association between VPS36 and VPS22. We assayed their interaction by depletion of Arl4A or

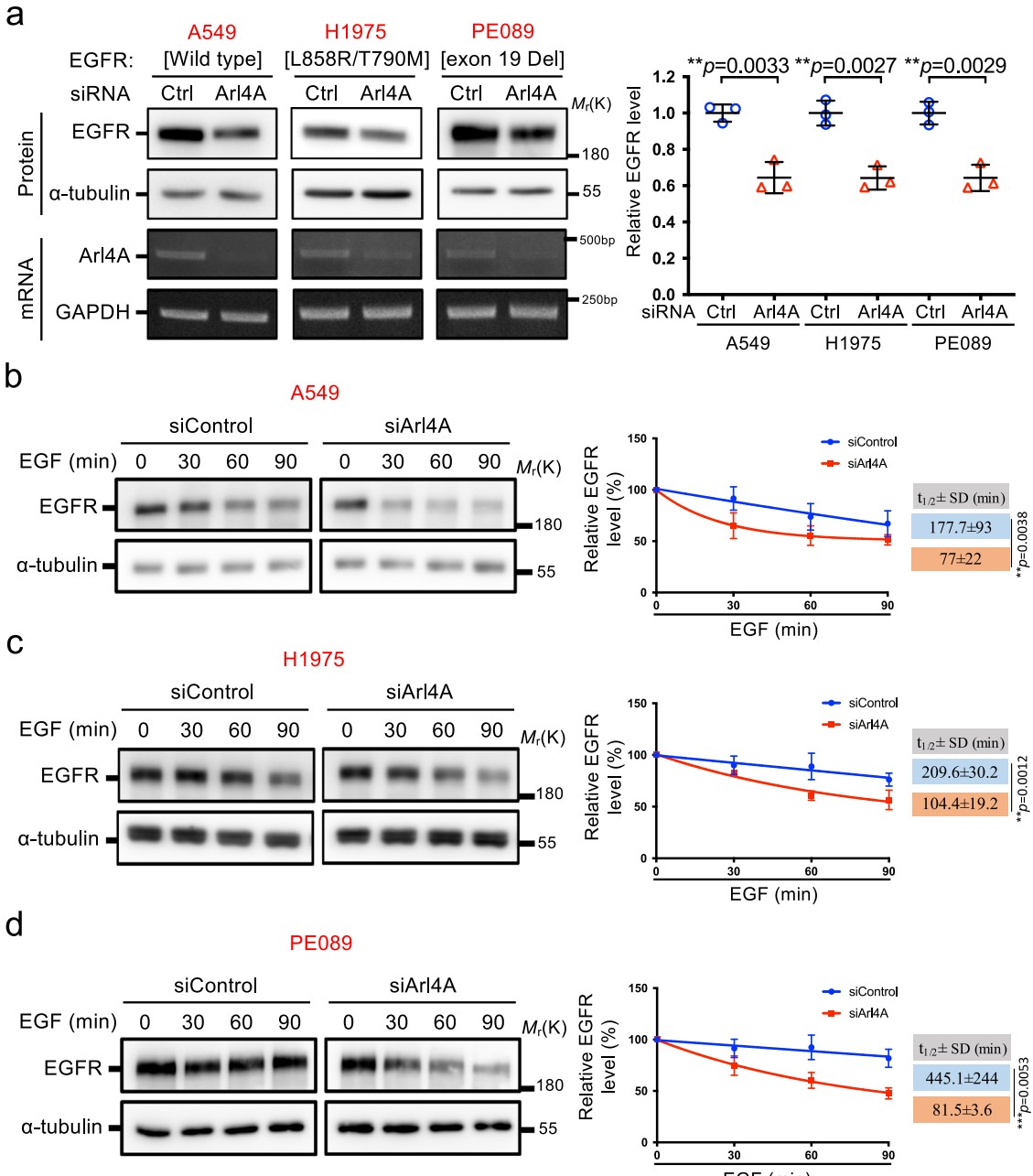

**Fig. 2 | Depletion of Arl4A accelerates EGFR mutant degradation. a** A549, H1975 and PE089 cells were treated with either the control or Arl4A siRNA. Cell lysates were analyzed by immunoblotting with anti-EGFR and anti-α-tubulin antibodies. Arl4A knockdown efficiency was assessed by RT-PCR amplification using Arl4A-specific primers, and GAPDH was used as an internal control (lower panel). The amounts of EGFR were determined by ImageJ software. The results represent the mean ± SD of three independent experiments and the *p*-values were assessed by a two-sided *t*-test. **b**–**d** The indicated cells were transfected for 48 h with siControl or siArl4A. Cells were sequentially subjected to EGFR degradation assays as described in the "Methods" section. To compare the EGFR degradation rates, we adjusted the EGFR protein levels at time 0 in each group to be the same for western blot analysis. The relative amount of EGFR was quantified by densitometric quantification from three experiments using α-tubulin intensity to normalize the EGFR signal, and error bars represent the mean ± SD. The $t_{1/2}$ ± SD of EGFR was obtained using a nonlinear one-phase decay fit of the time course; the *p*-values were assessed by a two-sided *t*-test. Source data are provided as a Source Data file.

expression of Arl4A$^{T5IN}$. We found that VPS36-VPS22 associates well in either event (Supplementary Fig. 14a, b). Furthermore, VPS22 remained bound to the VPS36-A6 mutant (Supplementary Fig. 14c). These results suggest that Arl4A binding to VPS36 does not interfere with the interactions between ESCRT-II complex.

To better define the function of Arl4A binding-defective VPS36, we compared the subcellular localization of VPS36-WT-Myc and Arl4A binding-defective VPS36-A6-Myc. In the VPS36-WT-Myc-expressing cells, VPS36 was observed in punctate structures

scattered throughout the cytoplasm that were clustered in the perinuclear region, as previously reported[40]. VPS36-Myc also localized to small punctate structures that were positive for the MVB marker CD63 and Rab5 Q79L. In comparison, VPS36-A6-Myc puncta showed lower MVB localization (Fig. 7a, b). In addition, Arl4A depletion that simulates perturbed Arl4A-VPS36 interaction also decreased the targeting of VPS36 at the MVBs (Supplementary Fig. 15). These results suggest that the binding of Arl4A can modulate VPS36 localization at MVBs. The ESCRT sorting pathway is

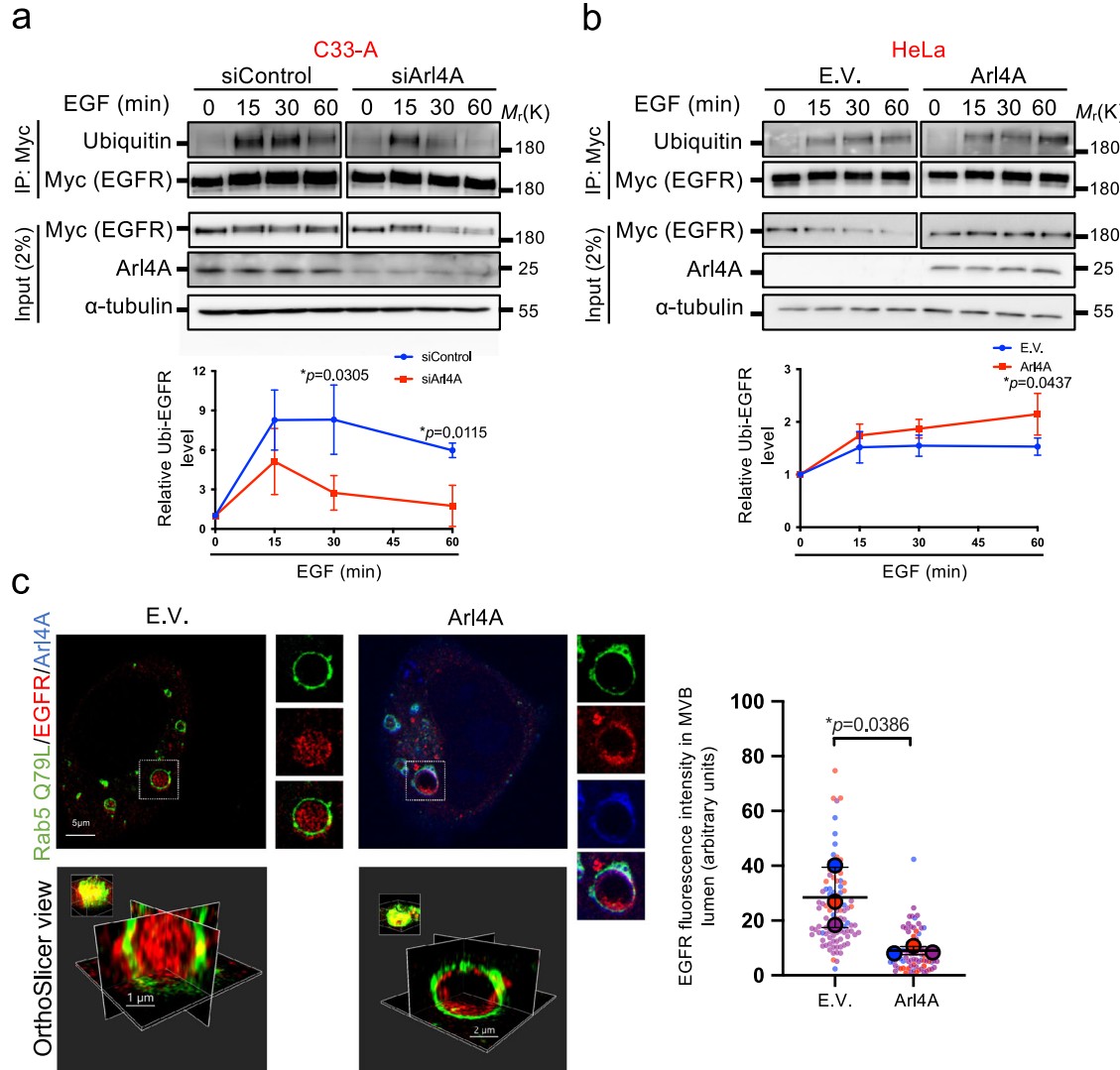

**Fig. 3 | Arl4A affects the level of EGFR ubiquitination and sorting of EGFR into MVBs. a** HeLa cells were transfected with empty vector (EV) or Arl4A WT in combination with EGFR-Myc (**b**) C33-A cells were transfected with either control or Arl4A siRNA in combination with EGFR-Myc. **a** and **b** After 24 h of transfection with plasmids or 48 h of transfection with siRNAs, the cells were starved and stimulated with EGF (100 ng/ml) for the indicated time. Cell lysates were immunoprecipitated with Myc-Trap for EGFR-Myc and analyzed by Western blotting with antibodies against ubiquitin or Myc. Total cell lysates (input) were immunoblotted with the indicated antibodies. The amounts of ubiquitin-EGFR were determined by ImageJ software. The results represent the mean ± SD of three independent experiments and the *p*-values were assessed by a two-sided *t*-test. **c** HeLa cells were transfected with either empty vector (EV) or Arl4A and GFP-Rab5A Q79L for 24 h. After serum

starvation, the cells were incubated with 100 ng/ml EGF for 10 min at 37 °C and then stained with anti-EGFR (red) and anti-Arl4A (blue) antibodies. Images were acquired using a Leica TCS SP8 STED microscope. Scale bar, 5 μm. Imaris 3D visualization software was used to create an enlarged endosome in a 3D orthoslicer view. Endosomes with diameters >2 μm were identified for analysis. Using the Rab5-Q79L outline to define the ROI, the EGFR intensities in the Rab5-Q79L-positive MVB lumen were measured. The results represent the mean ± SD of three independent experiments, each consisting of 26 or more endosome scans from ~10 cells each consisting of 26 or more endosome scans from ~10 cells (total numbers: EV = total 100 endosomes from 30 cells; Arl4A = total 81 endosomes from 30 cells; *p*-value was assessed by two-sided *t*-test). Source data are provided as a Source Data file.

composed of component assembly and finally disassembly after sorting cargos into the MVB. After sorting cargoes, ESCRT components are dissociated into the cytosol by VPS4 for the next round of assembly[41]. To examine whether VPS36 mutant A6, which showed less Arl4A binding and endosomal localization, triggers faster sorting of cargo into MVBs and dissociation into the cytosol, we blocked the final step of ESCRT sorting by knocking down VPS4A and VPS4B. We observed that VPS36 A6 is able to accumulate and well colocalize with CD63 (Fig. 7c). These results imply that the VPS36 A6 mutant did not lose its endosome binding ability, but rather rapidly assisted in cargo sorting, thus dissociating from the endosomal compartment.

## Arl4A affects the association of VPS36 and ESCRT-III members

We next asked whether Arl4A binding may affect the association of ESCRT-II and ESCRT-III. We found that, compared with VPS36-WT, VPS36-A6 exhibited both decreased colocalization (Fig. 8a) and association with the ESCRT-III component CHMP2A (Fig. 8b). In addition, Arl4A knockdown also abated VPS36 and CHMP2A interaction (Fig. 8c). These phenomena were also observed with other members of the ESCRT-III complex, such as CHMP6 (Supplemental Fig. 16). Taken together, these results suggested that Arl4A binding to VPS36 affected the ESCRT-II and ESCRT-III association. Several deubiquitinating enzymes (DUBs) are known to be recruited by ESCRT-III subunits through binding to the SH3 domain of STAM[42–45]

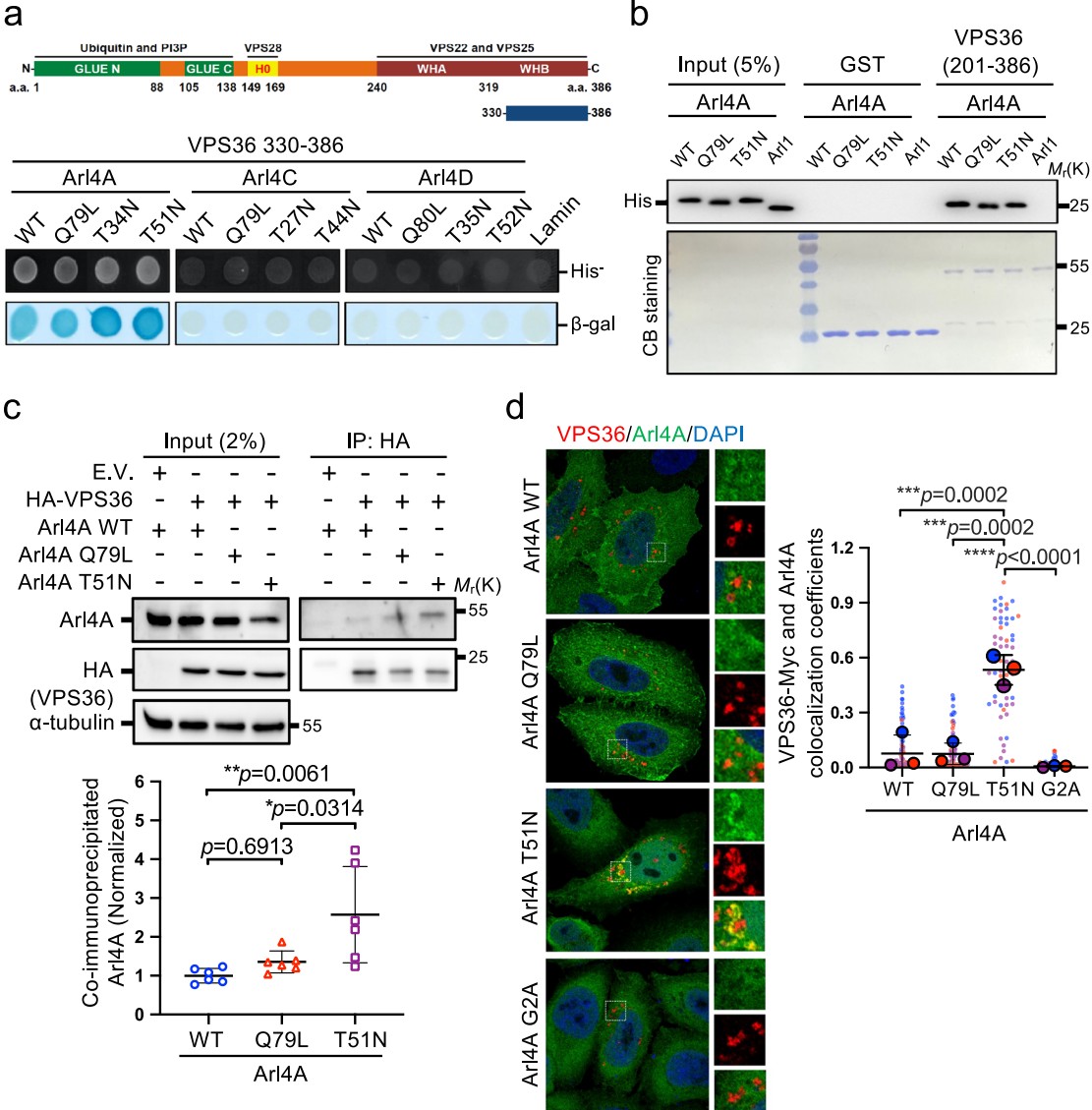

**Fig. 4 | Arl4A interacts with VPS36. a** The yeast reporter strain L40 was transformed with a construct encoding Gal4 transcriptional activation domain-fused VPS36 (330–386) and the indicated wild-type and mutant constructs of Arl4A, Arl4C, and Arl4D fused to the LexA DNA-binding domain. Colonies were patched on selective plates and assayed for β-galactosidase activity. Lamin was included as a negative control. **b** Equal amounts of His-tagged Arl4A WT, Q79L, T51N, and Arl1 (negative control) were pulled down by either GST or GST-VPS36 (201–386). Inputs and pulled-downed Arl4 proteins were probed using an anti-His antibody. Equal inputs of GST fusion proteins used in the assay were visualized by Coomassie Blue staining as shown at the bottom of each figure. **c** VPS36 interacted with Arl4A[WT], Arl4A[Q79L], and Arl4A[T51N] in vivo. Lysates of HeLa cells transfected with the indicated plasmids were immunoprecipitated with HA beads, and the bound proteins were

analyzed by Western blotting with anti-HA, anti-Arl4A, or anti-α-tubulin antibodies. The amounts of coimmunoprecipitated Arl4A were determined by densitometric quantification. The results represent the mean ± SD of six independent experiments and the p-values were assessed by one-way ANOVA with Tukey's test. **d** HeLa cells were cotransfected with VPS36 WT-Myc and Arl4A WT, Q79L, T51N, or G2A (myristoylation-defective) constructs. The cells were stained for Arl4A (green), VPS36-Myc (red), and DAPI (blue). Scale bar, 10 μm. Quantification of VPS36 and Arl4A colocalization was obtained from three biological replicates and is shown as the colocalization coefficient by ZEN software. The results represent the mean ± SD of three independent experiments (total WT = 103, Q79L = 79, T51N = 60, and G2A = 105 cells; the p-values were assessed by one-way ANOVA with Tukey's test). Source data are provided as a Source Data file.

to regulate MVB sorting[42,46]. In particular, two DUB molecules, AMSH (associated with the SH3 domain of STAM) and USP8 (ubiquitin-specific peptidase 8) have been extensively studied[42–45]. Because we have shown that Arl4A regulates the duration of EGFR ubiquitination (Fig. 3a, b), we wondered whether Arl4A affects the recruitment of DUBs by ESCRT-III and, leads to increased EGFR ubiquitination levels. We assayed the recruitment in Arl4A-overexpressing 293 T cells for that 293T cell line has a better ability to express DUBs. Arl4A overexpression resulted in a reduction in the interaction between CHMP2A and USP8/UBPY but not AMSH (Fig. 8d-e). In contrast, Arl4A depletion increased the association between CHMP2A and USP8

(Supplementary Fig. 17). These results indicate that Arl4A impedes the association of USP8/UBPY with ESCRT-III. To further clarify whether USP8 is involved in Arl4A-attenuated EGFR degradation, we first confirmed that USP8-knockdown delayed EGFR degradation (Supplementary Fig. 18a-b). Further knockdown of USP8 in Arl4A-depleted cells suppressed the accelerated EGFR degradation in Arl4A-depleted cells (Supplementary Fig. 18c). Combining Arl4A and USP8 knockdowns also suppressed the decreased duration of EGFR ubiquitinylation caused by Arl4A depletion (Supplementary Fig. 19); thus, we conclude that Arl4A and USP8 are involved in the same biological pathway.

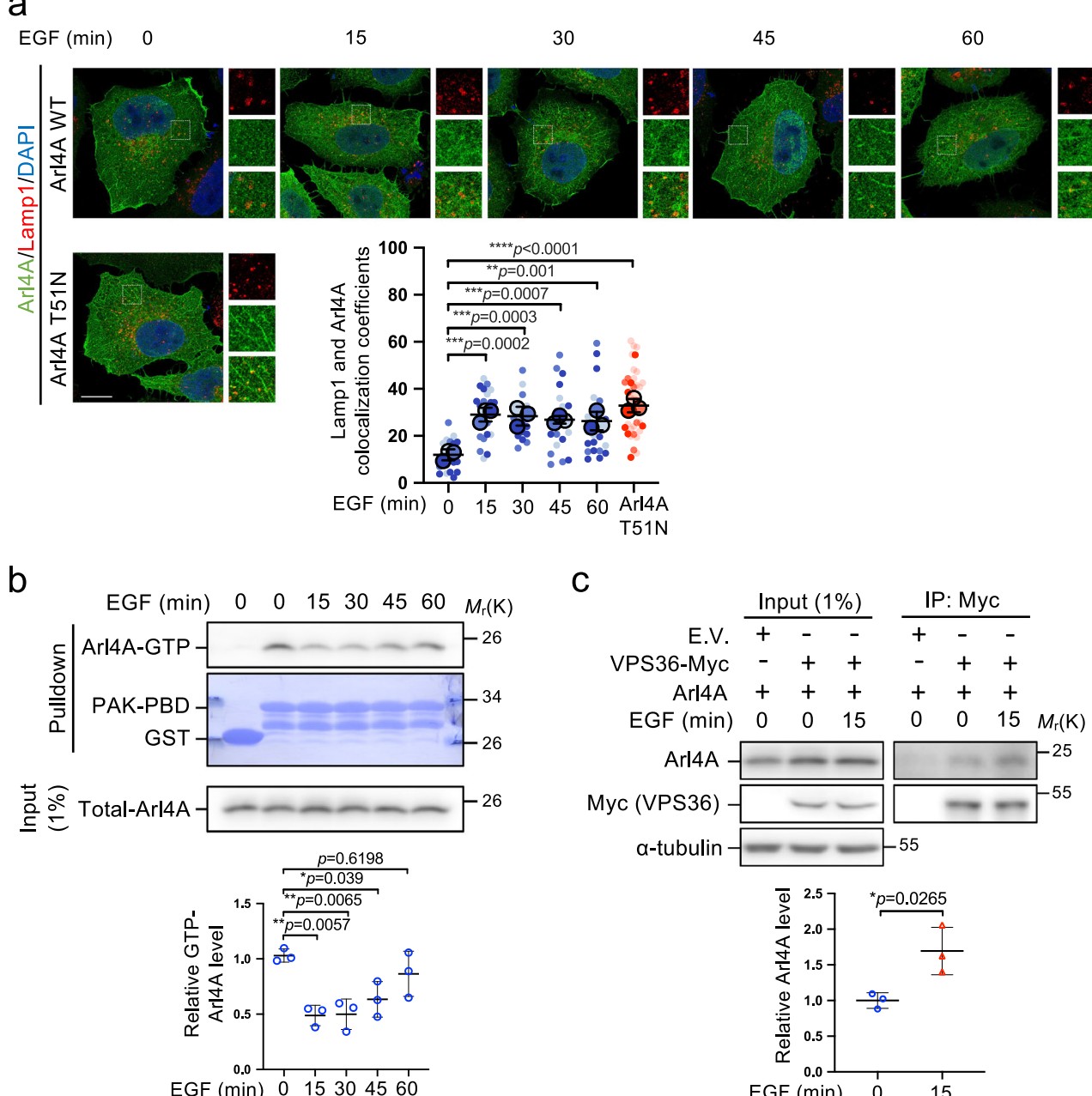

**Fig. 5 | EGF-mediated Arl4A-GTP conversion to the GDP-bound form is necessary for its interaction with VPS36. a** EGF-stimulated Arl4A translocates to the late endosomal compartment. HeLa cells serum starved for 3 h were treated with EGF (100 ng/ml) for the indicated times, and stained for Arl4A (green), Lamp1 (red), and DAPI (blue). The images were acquired using the Leica TCS SP8 STED microscope. Scale bar, 5 μm. The Arl4A signal served as a mask to define the region of interest (ROI) and was further used to quantify the colocalization coefficient between Lamp1 and Arl4A by Imaris software. The results represent the mean ± SD of three independent experiments (EGF 0 min = 22 cells, EGF 15 min = 22 cells, EGF 30 min = 22 cells, EGF 45 min = 19 cells, EGF 60 min = 24 cells, Arl4A T51N = 39 cells; the p-values were assessed by one-way ANOVA with Tukey's test). **b** HeLa cells were pretreated with serum-free medium and starved for 3 h, followed by EGF

(100 ng/ml) treatment for the indicated time. The amount of activated Arl4A was determined by the PAK-PBD pulldown assay. The amounts of GTP-Arl4A were determined by densitometric quantification. The results represent the mean ± SD of three independent experiments and the p-values were assessed by one-way ANOVA with Tukey's test. **c** EGF enhanced the interaction between VPS36 and Arl4A. Lysates of HeLa cells transfected with the indicated plasmids were immunoprecipitated with Myc-trap beads, and the amounts of bound proteins were analyzed by Western blotting with anti-Myc and anti-Arl4A antibodies. The amounts of coimmunoprecipitated Arl4A were determined by densitometric quantification and presented as the mean ± SD of three independent experiments and the p-value was assessed by a two-sided t-test. Source data are provided as a Source Data file.

## The Arl4A–VPS36 interaction attenuates EGFR degradation

The results depicted in Fig. 7 imply that losing Arl4A binding assists in more efficient cargo sorting. To further investigate the interaction between Arl4A and VPS36 in the context of EGF-induced EGFR degradation, we expressed VPS36-WT[res]-Myc or the Arl4A-binding deficient mutant VPS36-A6[res]-Myc after VPS36 knockdown. VPS36 knockdown leading to the deterrence of ligand-induced EGFR degradation is consistent with knockdown of the ESCRT-II complex reported in the literature[21,23,24,47] (Fig. 9a–c). Reintroducing VPS36-WT[Res]-Myc restored ligand-induced downregulation of EGFR whereas VPS36-A6[Res]-Myc further enhanced ligand-induced EGFR degradation compared to the knockdown control (Fig. 9a–c). These results show that

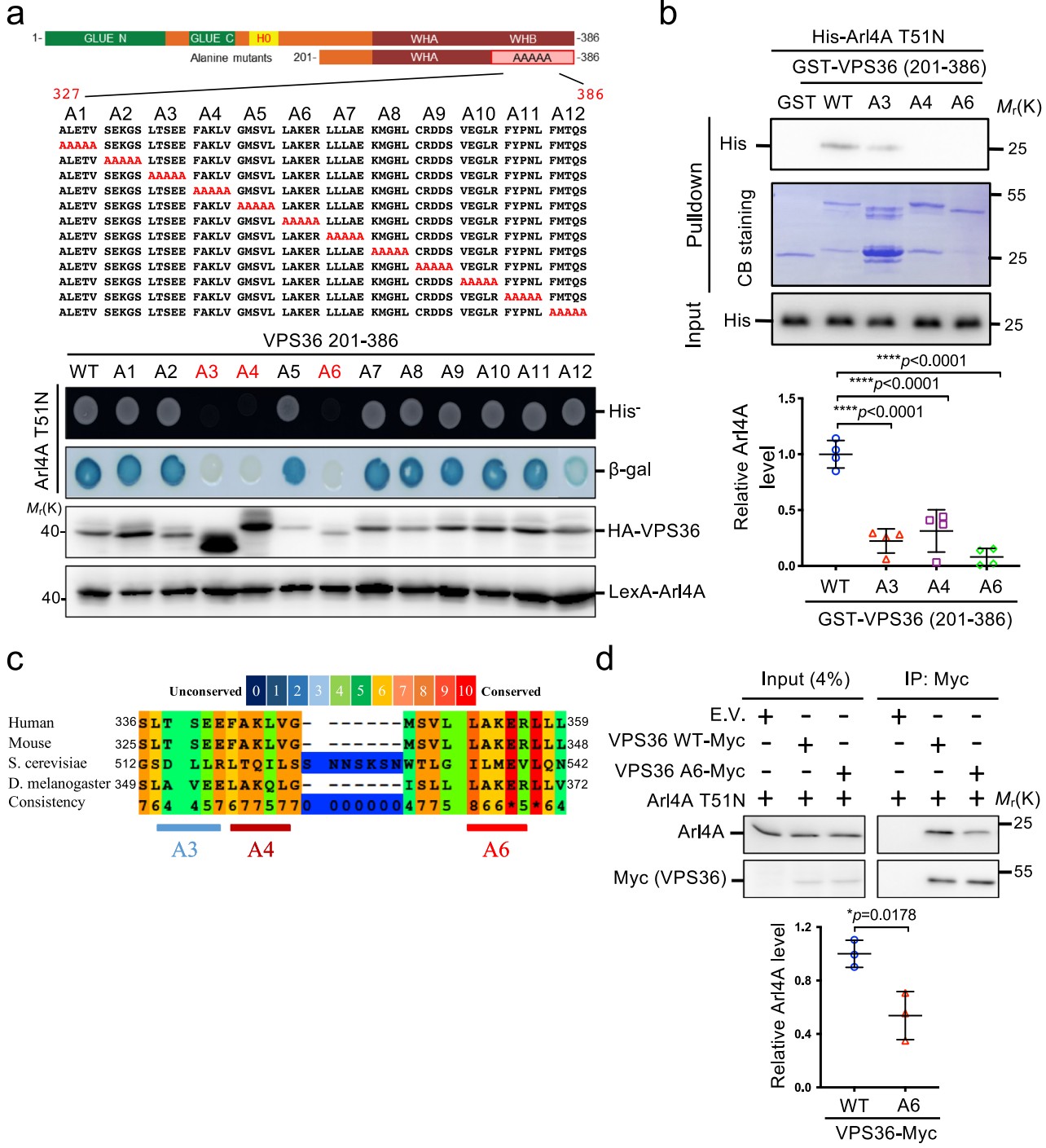

**Fig. 6 | Arl4A interacts with the WHB domain of VPS36. a** Schematic representation of VPS36 and its alanine mutants scanned from the C-terminus to a.a. 327 on VPS36 (201–386). Red letters indicate the amino acids mutated to alanine (upper). The yeast reporter strain L40 was transformed with constructs encoding VPS36 mutants fused to Gal4 AD and constructs encoding Arl4A T51N fused to LexA BD. **b** Purified His-tagged Arl4A T51N was incubated with the indicated GST fusion protein. Bead-bound His-Arl4A T51N was probed using an anti-His antibody. Arrows indicate the primary bands of GST-VPS36 (201–386). The amounts of Arl4A were determined by densitometric quantification and presented as the mean ± SD of four independent experiments (the *p*-values were assessed by ANOVA with Tukey's test). **c** Conservation of amino acid [351]LAKER[356] at the A6 region in different species.

The amino acid sequence alignment of the A3 to A6 regions at the VPS36 WHB domain in different species as indicated was analyzed by the PRALINE multiple sequence alignment tool. **d** Wild-type VPS36 or the VPS36-A6 mutant interacted with Arl4A in vivo. Lysates of HeLa cells transfected with the indicated plasmids were immunoprecipitated with Myc-Trap, and the bound proteins were analyzed by Western blotting with anti-Myc, anti-Arl4A, or anti-α-tubulin antibodies. The amounts of coimmunoprecipitated Arl4A were determined by densitometric quantification and presented as the mean ± SD of three independent experiments (the *p*-value was assessed by a two-sided *t*-test). Source data are provided as a Source Data file.

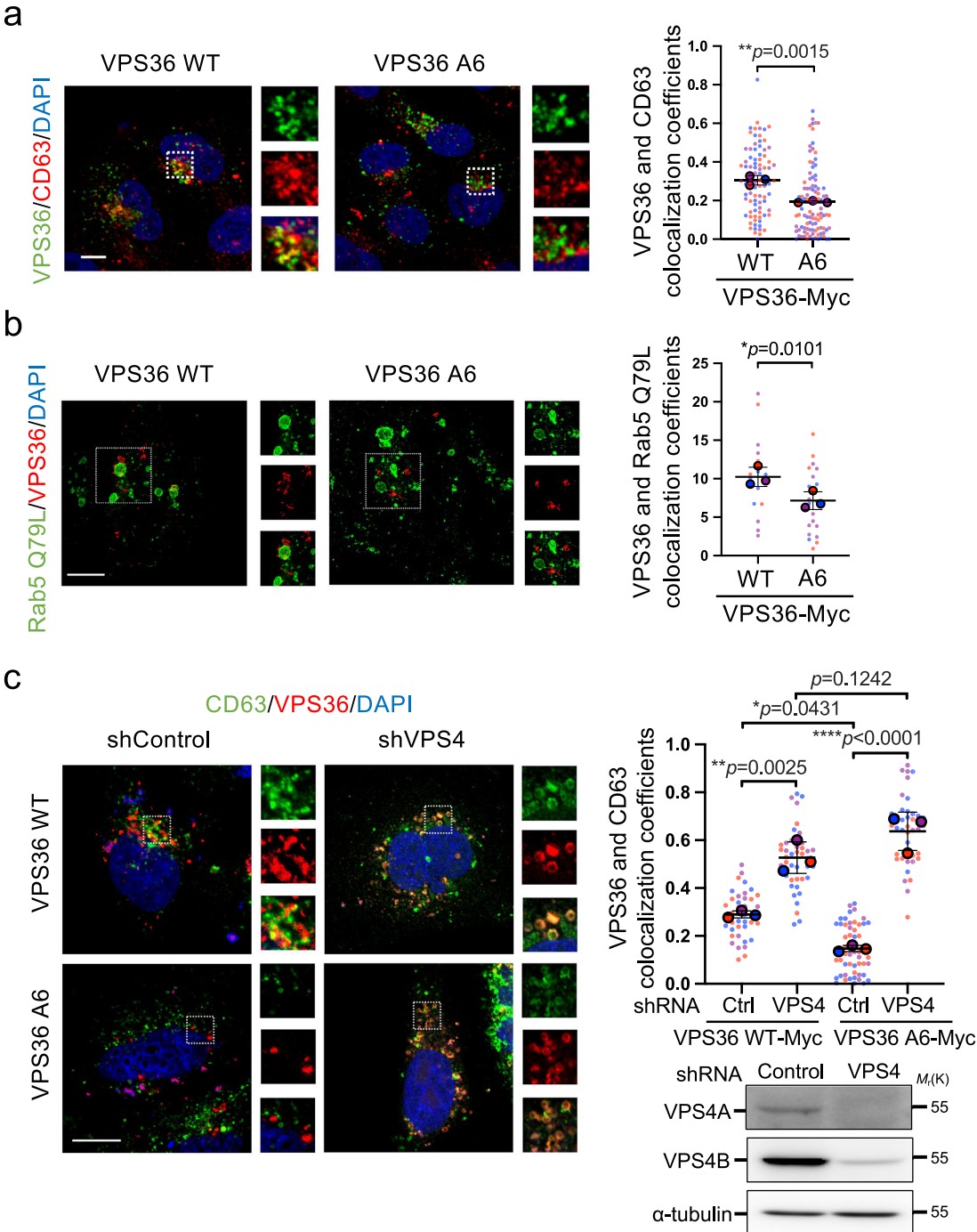

**Fig. 7 | Arl4A binding modulates the association of VPS36 with MVBs. a** HeLa cells were transfected for 24 h with the indicated plasmid. Confocal sections of HeLa cells stained with anti-Myc (VPS36-WT-Myc or VPS36-A6-Myc) (green), anti-CD63 (red), and DAPI (blue). Scale bar, 10 μm. Quantification of VPS36-Myc and CD63 colocalization is shown as the Pearson coefficient. The results represent the mean ± SD of three independent experiments and the *p*-value was assessed by a two-sided *t*-test (VPS36-WT = 84 cells, VPS36-A6 = 104 cells). **b** HeLa cells were co-transfected with VPS36-WT-Myc or VPS36-A6-Myc and GFP-Rab5A Q79L for 24 h. The cells were fixed and then stained with anti-Myc (red) and anti-GFP (green) antibodies. The images were acquired using the Leica TCS SP8 STED microscope. Scale bar, 5 μm. Imaris software was used to quantify the 3D-colocalization coefficient values of VPS36 and Rab5 Q79L. The results represent the mean ± SD of three independent experiments and the *p*-value was assessed by a two-sided *t*-test (VPS36-WT = 16 cells, VPS36-A6 = 22 cells). **c** HeLa cells stably expressing control shRNA or VPS4A and VPS4B shRNA using shRNA−expressing lentivirus. After puromycin selection, the cells were transfected with the indicated plasmids. Confocal sections of HeLa cells stained with anti-Myc (VPS36-WT-Myc or VPS36-A6-Myc) (green), anti-CD63 (red), and DAPI (blue). Scale bar, 10 μm. Quantification of colocalization of VPS36-Myc and CD63 presented as colocalization coefficient using the software ZEN. Results represent the mean ± SD of three independent experiments, and the p values were determined by one-way ANOVA with Tukey's test (shControl/VPS36- WT = 40 cells, shVPS36/VPS36- WT = 41 cells, shControl/VPS36-A6 = 52 cells, shVPS36/VPS36-A6 = 42 cells). Source data are provided as a Source Data file.

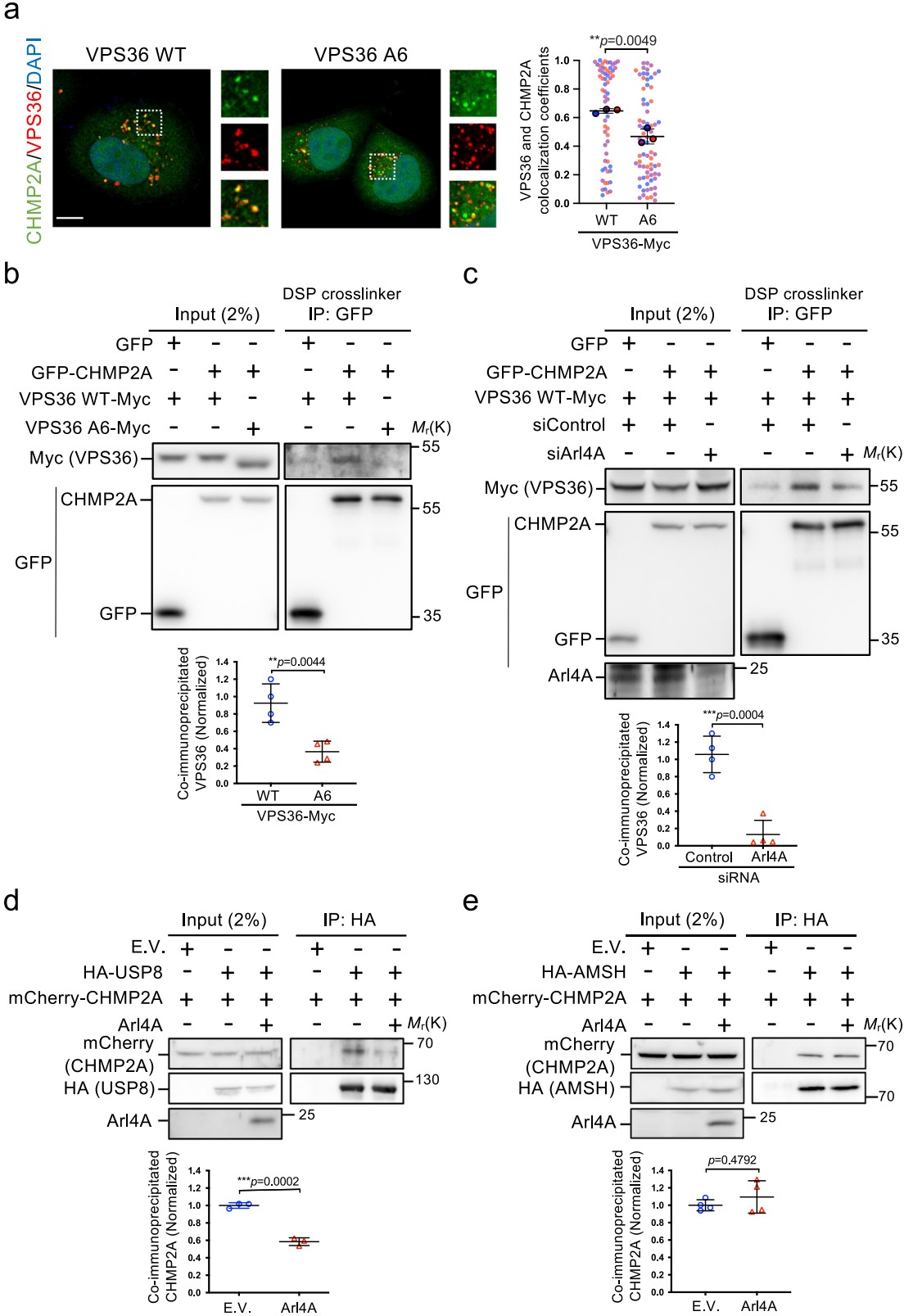

the lack of Arl4A binding to VPS36 accelerates ligand-induced EGFR degradation.

We found the VPS36 knockdown-caused EGFR degradation defect correlates positively with the inability of waiving ubiquitination signals from EGFR at the late sorting step (Fig. 9d–f). We further found that VPS36-WT[Res]-Myc restored the decrease of EGFR ubiquitination at 60 min in VPS36-depleted cells while VPS36-A6[Res]-Myc further

decreased these levels. This supports that Arl4A sustained EGFR ubiquitination and degradation defect may result from direct binding with ESCRT-II component VPS36.

## Discussion
Here, we report that an Arl4A-dependent mechanism attenuates EGFR degradation by binding to the ESCRT-II component VPS36. Several

**Fig. 8 | Arl4A affects the association of VPS36 and ESCRT-III components.**
**a** HeLa cells were transfected with the indicated VPS36 plasmids and GFP-CHMP2A for 24 h and stained with anti-Myc (VPS36-WT-Myc or VPS36-A6-Myc) (red) and DAPI (blue). Scale bar, 10 μm. Quantification of VPS36-Myc and GFP-CHMP2A colocalization was obtained from three biological repeats and shown as the colocalization coefficient by ZEN software. The results represent the mean ± SD and the *p*-value was assessed by a two-sided *t*-test (VPS36-WT = 62 cells, VPS36-A6 = 74 cells). **b** Comparison of VPS36-WT and VPS36-A6 binding ability toward ESCRT-III component CHMP2A in vivo. HeLa cells transfected with indicated plasmids were treated with 1 mM dithiobis(succinimidyl propionate) (DSP) for 2 h and quenched by Tris–HCl before cell lysis and immunoprecipitation with GFP-Trap. The bound proteins were analyzed by western blotting with anti-Myc or anti-GFP antibodies. The amounts of coimmunoprecipitated VPS36 were determined by densitometric quantification from four experiments as indicated. The results represent the mean ± SD and the *p*-value was assessed by a two-sided *t*-test. **c** Co-IP of VPS36 and CHMP2A in Arl4-knockdown HeLa cells. Cells transfected with the control or Arl4A siRNA as well as the indicated plasmids were treated with 1 mM DSP for 1 h and immunoprecipitated with GFP-Trap. The bound proteins were analyzed by western blotting with indicated antibodies. The amounts of coimmunoprecipitated GFP-CHMP2A were determined by densitometric quantification and represent the mean ± SD of three independent experiments (the *p*-value was assessed by a two-sided *t*-test). **d** and **e** Effects of Arl4A overexpression on the modulation of the **d** CHMP2A-USP8/UBPY interaction and **e** CHMP2A−AMSH interaction in 293T cells. Lysates of cells transfected with the indicated plasmids were immunoprecipitated with HA beads and bound proteins were analyzed by western blotting with anti-HA, anti-mCherry, or anti-Arl4A antibodies. The amounts of coimmunoprecipitated CHMP2A were determined by densitometric quantification from **d** three and **e** four experiments as indicated. The results represent the mean ± SD and the *p*-value was assessed by a two-sided *t*-test. Source data are provided as a Source Data file.

lines of evidence support this mechanism: (1) depletion of Arl4A promotes EGFR transport to lysosomes; (2) Arl4A depletion facilitates EGF-induced EGFR and NSCLC-related EGFR mutant degradation; (3) expression of Arl4A sustains EGFR ubiquitination levels; (4) Arl4A directly interacts with VPS36 in vitro and in vivo; and (5) and disrupting their interaction accelerates EGFR degradation. Currently, we hypothesize that Arl4A binding to VPS36 alters the association of VPS36 and the ESCRT-III component CHMP2A, which delays CHMP2A-deubiquitinating enzyme USP8 recruitment, which in turn affects EGFR ubiquitination levels (Fig. 9g).

In normal cells, receptor tyrosine kinase (RTK) activation is tightly regulated. Inappropriate activation of RTKs is associated with the development and progression of many human malignancies[48,49]. Ligand-induced RTK activation promotes internalization into the endocytic trafficking pathway, and subsequent degradation in lysosomes acts as an important mechanism that terminates receptor signaling[50]. Our results show that a lack of Arl4A results in strongly reduced protein levels of the RTKs EGFR and c-Met but not TfR and Na+/K+ ATPase, which is not cell line specific. We found that the expression levels of Arl4A and EGFR in two tested epithelial cervical carcinoma cell lines, HeLa and C33-A, were significantly different. Both the mRNA and protein levels of EGFR in C33-A cells were significantly lower than those in HeLa cells (Supplementary Fig. 2a, b); however, C33-A cells had much higher Arl4A expression than HeLa cells. Notably, our data showed that EGFR has a longer half-life in C33-A cells than in HeLa cells (Fig. 1f, g). These results also support that the expression level of Arl4A is critical for modulating EGFR degradation.

Although oncogenic mutations of EGFR (EGFR[L858R/T790M], EGFR[ex19Del]) increase its internalization, they also prevent its degradation, which helps sustain its signaling[32,33]. Interestingly, Arl4A depletion consistently led to significant protein decrease and accelerated degradation of EGFR[L858R/T790M] and EGFR[ex19Del] in NSCLC H1975 and PE089 cells. The present study revealed that Arl4A acts as a positive regulator of EGFR, which explains the accelerated degradation of EGFR mutants in Arl4A-depleted NSCLC cells. Because Arl4A is also involved in regulating the c-Met tyrosine kinase receptor, thus, high Arl4A expression may lead to the activation of multiple oncogenic pathways. It would be interesting to investigate whether Arl4A attenuates the degradation of other RTKs through the same mechanism. Overall, this mode of Arl4A regulation that bypasses EGFR degradation resistance in NSCLC cells offers a therapeutic approach and is worth investigating.

One key mechanism for decreased EGFR signaling is lysosomal trafficking and degradation of the activated receptor. Several mechanisms have been identified to regulate late endocytic trafficking and degradation of EGFR. LAPTM4B, an endosomal transmembrane oncoprotein, inhibits EGFR intraluminal sorting at late endosomes and lysosomal degradation[51]. PTEN promotes late endosomal maturation and controls endocytic trafficking of EGFR by dephosphorylating

Rab7[52]. The CD44s splice isoform triggers RTK-dependent signaling and was found to be internalized into endosomes to inhibit Rab7A-mediated EGFR late endocytic trafficking to lysosomes for subsequent degradation[53]. Our study reveals a mechanism by which Arl4A attenuates late endocytic trafficking and degradation of EGFR.

Our data indicate that EGF signaling spatiotemporally mediates Arl4A nucleotide switching from Arl4A-GTP to Arl4A-GDP. These results also suggest that EGF signaling-dependent Arl4A-GTPase-activating protein plays an important role in regulating the Arl4A-VPS36 interaction at the late endosomal compartment. A difficult challenge for the future will be to identify the upstream regulators of Arl4A to expand our knowledge of Arl4A function at late endosomes.

Arl4A interacts directly with the ESCRT-II component VPS36; however, the overexpression or depletion of Arl4A did not affect the interaction between VPS36 and VPS22, suggesting that Arl4A binding to VPS36 may not affect the assembly of the ESCRT-II complex. Instead, their binding may have affected the dynamics of ESCRT-II complex recruitment. ESCRT components are diffusely localized in wild-type yeast because of the rapid recycling of endosomal ESCRTs to the cytosol as part of their normal function[54]. However, Slagsvold et al. showed that GFP-tagged VPS36 assembles and partially colocalized with the MVB marker LBPA upon EGF activation[40]. Interestingly, our results showed that VPS36 reduces colocalization with the MVB marker CD63 in Arl4A-depleted cells. Moreover, the colocalization of Arl4A-binding-deficient VPS36 with CD63/MVBs was also decreased. However, the VPS36 A6 mutation does not affect its ability to localize to the endosome compartment. Therefore, our data suggest that Arl4A binding may delay the recycling of VPS36/ESCRT-II back to the cytosol. We also found that the Arl4A-binding-deficient VPS36-A6 mutant exhibited reduced coprecipitation of CHMA2A, which may alter the function/integrity of the ESCRT-III complex. Currently, we hypothesize that Arl4A binding to VPS36 sustains the association of VPS36 and ESCRT-III component CHMP2A, resulting in a reduction of CHMP2A-deubiquitinating enzyme USP8 interaction, which could further affect EGFR ubiquitination levels. Previous studies have shown that the recruitment of USP8/UBPY to the ESCRT-III complex deubiquitinates EGFR and thereby drives EGFR into ILVs, which fuse with lysosomes for further degradation[37,46]. However, further study is needed to clarify whether Arl4A-binding to VPS36 directly mediates the CHMP2A-USP8/UBPY interaction to deubiquitinate EGFR.

Previous studies have shown that plasma membrane-localized Arl4A is involved in actin remodeling[8,9] and cell migration[11,55]. Golgi-localized Arl4A participates in GCC185-mediated endosome-to-Golgi transport[12]. Arl4A dominates the above functions in the GTP-bound state. In this study, we showed that GDP-bound endosomal-localized Arl4A regulates EGFR endocytic transport to lysosomes. Therefore, the nucleotide-binding states of Arl4A are likely to be regulated at different membrane compartments, where it also may impact on EGFR transportation. Our data showed that Arl4A actually involved in the

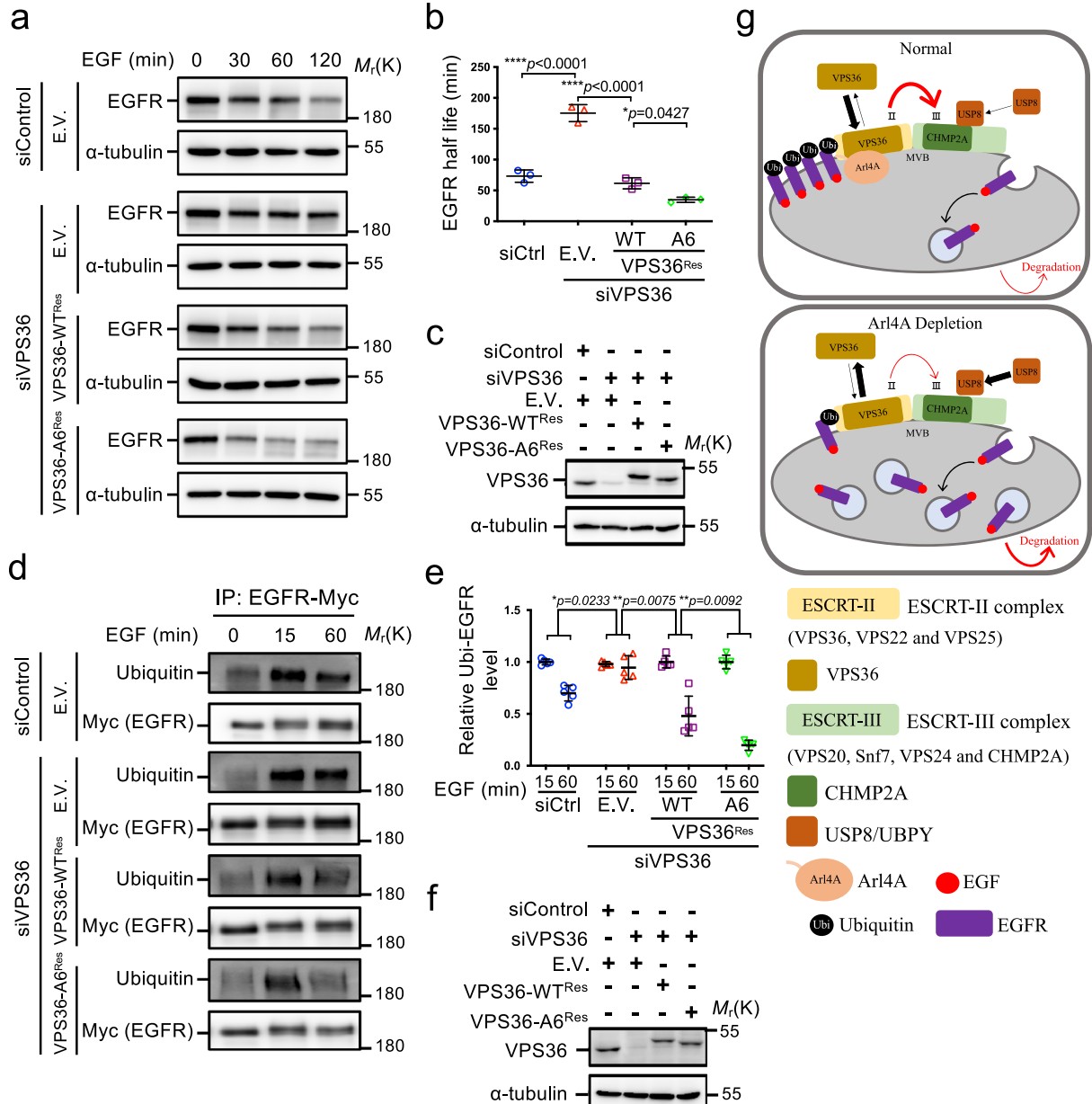

**Fig. 9 | The Arl4A–VPS36 interaction attenuates EGFR degradation. a** HeLa cells were transfected with siControl or siVPS36 as well as the indicated plasmids. After 48 h of transfection, the cells were starved and stimulated with EGF (50 ng/ml) for the indicated times. **b** Quantification of EGFR levels was performed using ImageJ software. The results represent the mean ± SD of three independent experiments and the *p*-values were assessed by one-way ANOVA with Tukey's test. **c** The lysates of the cells used in Fig. 9a were analyzed by Western blotting with antibodies against VPS36 and α-tubulin. **d** C33-A cells were transfected with either the control or VPS36 siRNA and combined with EGFR-Myc and the indicated plasmid. After 48 h, the cells were starved and stimulated with EGF (50 ng/ml) for the indicated times. Cell lysates were immunoprecipitated with Myc-Trap and analyzed by Western blotting with antibodies against ubiquitin. **e** Quantification of ubiquitin-EGFR was performed using ImageJ software. The graph resulted from densito-metric scanning of the 60-min ubiquitin-EGFR signal relative to the 15-min

ubiquitin-EGFR signal. The relative amount of ubiquitinated EGFR at 60 min was compared by one-way ANOVA. The results represent the mean ± SD of five inde-pendent experiments and the *p*-values were assessed by one-way ANOVA with Tukey's test. **f** The lysates of the cells used in Fig. 9d were analyzed by western blotting with antibodies against VPS36 and α-tubulin. Source data are provided as a Source Data file. **g** Model showing how Arl4A attenuates EGFR degradation. Endosomal Arl4, through the interaction of ESCRT-II component VPS36, mediates EGFR ubiquitination levels, sorting to MVBs, and lysosomal degradation. Arl4A-depletion enhances the endocytosed EGFR transport to late endosomes/lysosomes and accelerates EGF-stimulated EGFR degradation. Specifically, the Arl4A-VPS36 interaction facilitates the association of VPS36 with MVBs and sustains the asso-ciation of VPS36 and the ESCRT-III component CHMP2A, resulting in a reduction of CHMP2A-deubiquitinating enzyme USP8 interaction, which could further affect EGFR ubiquitination levels.

---

EGFR endocytic pathway from the plasma membrane to early endo-somes (Supplementary Figs. 2a, 10a, 11a, 12a). This is reminiscent of the Arf6 function at many locations along the plasma membrane, where multiple sites of Arf6 action influence the sorting of membrane pro-teins, endocytic pathways, and the structure of the plasma membrane[56,57]. In addition, Arl4A was reported to recruit ARNO to the

plasma membrane to regulate Arf6 activity[8,9], as well as to regulate the activity of CDC42 through Robo1[55]. These proteins are all related to endocytosis[58–60]. Therefore, the regulation of these proteins may be affected after Arl4A knockdown, thus accelerating the internalization of EGFR. How Ar4A is involved in early endocytic transport needs to be further investigated.

Together, our study provides a mechanism for the regulation of EGFR degradation by the Arl4A–VPS36 interaction. Future studies focused on the spatiotemporal regulation of Arl4A will improve our understanding of the molecular mechanism underlying Arl4A-based regulation of ESCRT biology during normal cellular function and oncogenesis.

## Methods

### Ethics statement

Our research complies with all relevant ethical regulations. The animal experiments were performed in accordance with the National Taiwan University Institutional Animal Care and Use Committee and the National Taiwan University College of Medicine Laboratory Animal Center.

### Cell culture and transfection

The A549 cell (BCRC Number: 60074), HeLa cell (BCRC Number: 60005), C33-A (BCRC Number: 60554), and 293t (BCRC Number: 60019) cell were purchased from Food Industry Research and Development Institute (Hsinchu, Taiwan). The PE089 was characterized as harboring an EGFR exon 19 deletion and derived from a Taiwanese female patient with adenocarcinoma of the lung (courtesy of Professor Ko-Jiunn Liu, National Health Research Institutes, Tainan, Taiwan). The H1975 cell line was provided by Professor Li-Chung Hsu (Institute of Molecular Medicine, National Taiwan University, Taipei, Taiwan). HeLa, 293T, and C33-A cells were cultured in DMEM. A549 cells were cultured in F12-k. H1975 cells were cultured in RPMI medium, and PE089 cells were cultured in MEM. All media were supplemented with 10% fetal bovine serum, penicillin, and streptomycin (Invitrogen) in a humidified incubator with 5% $CO_2$ at 37 °C. Transient transfection was performed using Lipofectamine 2000 reagent according to the manufacturer's protocol (Invitrogen).

### Antibodies

The antibodies used in this study were as follows: anti-VPS36 (#PA5-60561, Invitrogen,1:1000); and anti-HA (# MMS-101R, Biolegend,1:3000); anti-LexA (Covance,1:12,000); anti-Myc (#2276S,1:3000), anti-mCherry (#43590,1:3000), anti-EGFR (#4267,1:3000), anti-p-EGFR (#3777,1:2000), anti-ERK1/2 (#9107,1:3000), anti-p-ERK1/2 (4370,1:3000), anti-c-Met (#8198,1:3000), anti-cleaved PARP (#9541,1:1000) and anti-cleaved caspase-7 (#8438,1:1000) (Cell Signaling Technology); anti-GFP mouse (#sc-9996,1:3000), anti-ubiquitin (P4D1, #sc-8017,1:1000), anti-VPS4A (#sc-393428,1:1000), anti-VPS4B (#sc-377162,1:1000) and anti-Na/K ATPase (#sc-21712,1:1000) (Santa Cruz), anti-His (#2010762 A, Takara,1:5000), anti-α-tubulin (Sigma-Aldrich,1;5000). The anti-EGFP rabbit (generated in our laboratory). Arl4A (1:3000) antibodies were previously described (Lin et al. 2011). For the EGFR trafficking assay, we used an anti-extracellular region of EGFR (#ab30, Abcam) at a concentration of 18 µg/ml. For IF, dilutions of anti-Arl4A (1:200), anti-CD63 (#GTX28219, GeneTex,1:500), anti-Myc (#2276S,1:500), anti-Lamp1 (#sc-20011,1:500) and anti-EGFR (#4267,1:500) primary antibodies and Alexa Fluor 488/594 anti-rabbit and mouse IgG secondary antibodies (A-11012, A-11001 and A-11032, Invitrogen, 1:1000) were us. For western blot, the secondary antibodies used were goat horseradish peroxidase (HRP)-conjugated anti-rabbit/ mouse IgG (NA934V/NA931V, GE Healthcare, 1:5000).

### Plasmids and siRNA

In the mammalian expression system, untagged Arl4A WT, Arl4A Q79L, Arl4A T51N, and G2A mutants were cloned into the pSG5 vector (Stratagene) as described previously[11]. The Myc-tagged EGFR expression construct was kindly provided by Dr. Li-Chung Hsu (Institute of Molecular Medicine, National Taiwan University, Taiwan). CHMP2A-mCherry, USP8-HA, and AMSH-HA were purchased from Addgene. VPS36 (201-386) was cloned into pGEX vectors (GE Healthcare). The

VPS36 WT and A6 mutants were cloned into pcDNA3.1 vectors (Invitrogen). The Arl4A/VSP36 RNAi-resistant clones contained silent point mutations in the RNAi targeting sequences that decreased siRNA binding to the mRNA. These constructs were generated by two-step site-directed mutagenesis PCR primers as follows:
pSG5-Arl4A^Res
F: 5'-CAAGATTTGAGAAACTCTTTGTCACTGTCGTTGG-3'
R: 5'-CTCAATTTCTGACAGTGACAAAGAGTTTCTCAAATCTTG –3'.
pcDNA3.1-VPS36^Res
F: 5'-TTGCTGAGCATGGGCATCGCGAACCCAGTTACCAGAGAAACC-3'
R: 5'-TTGCTGAGCATGGGCATCGCGAATCCGGTGACCAGAGAAACC-3'.

The sequences of the siRNAs used in this study were as follows: Arl4A siRNA#1: GGAACTCATTGTCACT; Arl4A siRNA #2: TGCCTTCATTTCAGTCT; VPS36 siRNA: GGGAATAGCTAACCCAG; USP8 siRNA: TGAAATACGTGACTGTTTAT; VPS4A shRNA: GCAAAGA-GAAACACGGCAAGA; VPS4B shRNA: GTGAAGCCGCACGTAGAATTA.

### Yeast two-hybrid assay

Yeast strain L40 [MATα trp1 leu2 his3 LYS 2::(lexAop)4-HIS3 URA3::(lexAop)8-lacZ] was constructed with two readouts for an interaction of histidine auxotrophy and β-galactosidase expression using the LexA DNA-BD and GAL4-activation domain systems. Yeast strains were grown at 30 °C in a rich medium (1% yeast extract, 2% Bacto-Peptone, 2% glucose) or in a synthetic defined (SD) medium containing 2% glucose and all essential amino acids except for tryptophan and leucine. For expression of LexA DNA-binding domain (BD) fusion proteins in yeast, cDNA of Arl4A (WT, Q79L, T34N, and T51N), Arl4C (WT, Q72L, T27N, and T44N), Arl4D (WT, Q80L, T35N, and T51N) and lamin was cloned and inserted into a pBTM116 vector. After lithium acetate transformation, the double transformants were plated with a synthetic medium lacking histidine, leucine, tryptophan, uracil, and lysine. Plates were incubated at 30 °C for 3 days. His+ colonies were patched onto selective plates and assayed for β-galactosidase activity using a nitrocellulose transfer membrane filter (Immobilon, Merck Millipore), and assayed as described previously[61].

### In vitro binding assay

For the expression of GST fusion proteins in *Escherichia coli*, VPS36, and its deletion mutants were cloned into pGEX vectors (GE Healthcare). To produce the GST fusion protein, the indicated constructs were transformed into BL21 cells. Colonies expressing recombinant proteins were grown at 37 °C in Luria-Bertani medium. Protein was induced by 0.5 mM isopropyl-β-D-1-thiogalactopyranoside (IPTG) for 3 h at 37 °C. For purification of the GST-tagged protein, the pellet was incubated in phosphate-buffered saline (PBS), sonicated, and centrifuged at 14,000 × g for 20 min to separate the soluble fraction (supernatant) and insoluble fraction (pellet). The supernatant was incubated with glutathione Sepharose 4B beads (Amersham Pharmacia Biotech) at 4 °C for 3 h. The beads were washed five times with cold PBS. For His-tagged protein purification, For His-tagged protein purification, pellets were suspended in lysis buffer (20 mM Tris–HCl, pH 7.9; 500 mM NaCl; 5 mM imidazole; 10% glycerol; 0.1% Triton X-100; PI cocktail; and 100 µg/mL lysozyme) and incubated for 30 min at 4 °C in a shaker. Nitrogen was dissolved in the cell suspension using the nitrogen cavitation bomb (4639, Ashcroft Inc.) at a pressure of 1500 psi and at 4 °C for 10 min. Lysates were then centrifuged at 14,000 × g for 20 min. The supernatant was incubated with Ni-NTA resin (88222, Thermo Scientific) at 4 °C for 1 h. Unbound proteins were washed out by lysis buffer containing 0, 25, and 50 mM imidazole. Elution buffer (20 mM Tris–HCl, pH 7.9; 500 mM NaCl and 250 mM imidazole) was used to elute purified His-tagged proteins at 4 °C for 1 h. GST- or His-tagged proteins were quantified using SDS-PAGE. 2 µg GST-tagged proteins were incubated with 2 µg His-tagged Arl4A in 1 ml of binding buffer (20 mM Tris-HCl, pH 8.0, 150 mM NaCl, 5 mM MgCl2, 1 mM EDTA, 1 mM DTT, 10% glycerol, 1% Triton X-100, 1 mM NaN3, and

protease inhibitor cocktail) for 1 h at 4 °C in a shaker. After 1 h, the GST beads were washed three times with 1 ml of binding buffer. After the buffer was removed, the beads were boiled in 20 µl SDS–PAGE sample buffer and then subjected to Western blotting analysis.

## Immunoprecipitation assay

HeLa cells were lysed in lysis buffer (10 mM Tris–HCl, pH 7.5; 150 mM NaCl; 2 mM EDTA; 1% Triton X-100; protease inhibitor mixture) at 4 °C for 30 min. Lysates were clarified by centrifugation at $15,000 \times g$ for 30 min. The cell lysates were incubated with primary antibodies at 4 °C for 4 h with rotation. Then, the antibody-captured proteins were precipitated using protein G (Invitrogen). Coimmunoprecipitated proteins were analyzed by Western blotting. HeLa, 293T or C33-A cells were lysed in lysis buffer (10 mM Tris–HCl, pH 7.5; 150 mM NaCl; 0.5 mM EDTA; 0.5% NP40; protease inhibitor mixture) at 4 °C for 30 min. The lysates were clarified by centrifugation at $15,000 \times g$ for 30 min. The cell lysates were incubated with HA beads or Myc-Trap (ChromoTek) at 4 °C for 1 h with rotation. Coimmunoprecipitated proteins were analyzed by Western blotting.

## Western blotting analysis

Cell lysates were prepared in RIPA buffer [50 mM Tris–HCl, pH 7.4, 150 mM NaCl, 1% NP-40, 0.5% SDS, 50 µg/ml Nα-ρ-tosyl-L-lysine chloromethyl ketone, 1 mg/ml benzamidine and protease inhibitor mixture (Sigma-Aldrich)], and protein concentrations were determined using a Bio-Rad protein assay (Bio-Rad). The protein samples were separated by SDS–PAGE and transferred to PVDF membranes (Millipore). After incubation with antibodies, the membranes were developed with an ECL system (Amersham-Pharmacia Biotech) according to the manufacturer's instructions and imaged on an ImageQuant™ LAS 4000 biomolecular imager. The protein bands were quantified using ImageJ software.

## EGFR ubiquitination assay

Cells were harvested with lysis buffer [50 mM Tris–HCl pH 7.5, 150 mM NaCl, 10% glycerol, 1% Triton X-100, 2 mM EDTA, 20 mM NEM, protease inhibitor tablet (Sigma)]. The cell lysate was further incubated with Myc-Trap (ChromoTek) for 1 h. The Myc-Trap beads were then washed with lysis buffer three times and boiled in 30 µl of SDS–PAGE sample buffer before being subjected to Western blot analysis.

## EGFR degradation assay

The assay was performed as previously described, with some modifications. Briefly, cells were serum-starved (meant in serum-free medium) for 6 h before being pretreated for 1 h with 20 µg/ml cycloheximide, which prevented the synthesis of new EGFR during stimulation by EGF. Then, the cells were continuously stimulated with 100 ng/ml EGF for 0, 15, 30, 60, and 90 min. The levels of undegraded EGFR were determined by Western blotting using an anti-EGFR antibody (Cell Signaling). The intensity of the EGFR signal at each time point was quantified by ImageJ. EGFR protein degradation over time, a nonlinear one-phase exponential decay regression model, was used to fit a curve to the data (GraphPad Software, Inc., La Jolla, CA, USA). The half-life ($t_{1/2}$) of EGFR was estimated based on the best-fit curve. EGF was used at a concentration of 50 ng/ml to evaluate ESCRT-II complex-dependent ligand-induced EGFR degradation as previously described[21,23,24,47].

## EGF-Alexa 555 internalization and degradation assay

EGF-Alexa 555 internalization and degradation assays were performed as previously described[62]. HeLa cells grown on coverslips were transfected with either siControl alone or siArl4A. After 48 h, the cells were serum-starved for 6 h in starvation medium (DMEM without fetal bovine serum). For labeling of the cell surface EGF receptor, cells were incubated with 80 ng/ml EGF-Alexa 555 (EGF conjugated with Alexa 555) (Molecular Probes) in starvation medium for 1 h at 4 °C, followed by three washes with ice-cold starvation medium to remove unbound EGF-Alexa 555. Surface-bound EGF-Alexa 555 was internalized for degradation by incubation in complete DMEM at 37 °C for 0, 15, 30, 60 and 120 min. Cells were fixed and stained with 4′,6-diamidino-2-phenylindole (DAPI). The intensity of the EGF-Alexa 555 signal at each time point was quantified using a Carl Zeiss LSM880 Airyscan system with ×63/1.3 NA oil objective lenses and 405 or 543 nm lasers.

## Immunofluorescence staining and confocal microscopy

For immunofluorescence staining, cells were washed with PBS followed by fixation with 4% paraformaldehyde in PBS for 15 min (followed by permeabilization with 0.2% Triton X-100 in PBS for 5 min). After the cells were blocked with 2% BSA in PBS for 30 min, they were incubated with primary antibodies in PBS containing 1% BSA for 2 h. The cells were washed three times with PBS for 10 min and incubated with secondary antibody in PBS with 1% BSA for 1 h. After another three washes in PBS, the coverslips were mounted on slides with mounting medium (Mowiol® 4-88, Carl Roth). For confocal microscopy, the samples were imaged using a Carl Zeiss LSM880 Airyscan system with a ×63/1.3 NA or ×100/1.3 NA oil objective lenses and 405, 488, or 543 nm lasers.

For super-resolution microscopy, cells were cultured on Zeiss high-performance cover glasses. The fluorescent secondary antibodies used in super-resolution microscopy were anti-rabbit IgG-Abberior STAR 580 or anti-mouse IgG-Abberior STAR 635P. The coverslips were mounted on slides with a mounting medium (ProLong Diamond antifade mountant). The samples were imaged using a Leica TCS SP8 STED system with an HC PL APO ×100/1.40 oil STED white objective lens. The excitation wavelength was 580 or 635 nm, and the depletion wavelength was 775 nm.

## The crosslinker dithiobis (succinimidyl propionate) (DSP) stabilizes protein interactions in vivo

Cells were treated with 1 mM of the cleavable cross-linker dithiobis (succinimidyl propionate) (DSP) for 2 h on ice as described previously[63]. After 2 h, cross-linking was quenched by adding Tris–HCl to a final concentration of 15 mM for 15 min on ice. After DSP quenching, the cells were washed with PBS/Ca/Mg (PBS with 2 0.1 mM CaCl and 1 mM MgCl₂) two times, and then lysis buffer was added for 15 min (50 mM Tris–HCl, pH 7.5; 150 mM NaCl; 0.5 mM EDTA; 0.5% NP40; protease inhibitor mixture). The cell lysates were incubated with GFP-trap at 4 °C for 1 h with rotation. Coimmunoprecipitated proteins were analyzed by Western blotting.

## Image analysis

In Fig. 1a, the intensity of EGF-555 was measured using ImageJ software. The outline of EGF-555-positive areas was used to define a whole cell body referred to as the region of interest (ROI). This mask was then used as a reference to quantify EGF-555 intensity/cell in the image.

In Fig. 3c, EGFR signals localized inside the Rab5 (Q79L)-enlarged endosomes (MVBs) were quantified using ImageJ. Using the GFP-Rab5 Q79L signals as the mask to define the outline of the endosomes, endosome diameters >2 µm were taken for calculation of EGFR intensities.

In Figs. 4d, 7a, and 8a and Supplementary Figs. 2, 10, 11, 12, and 15 we used ZEN software to determine the colocalization coefficient of two proteins of interest. In Fig. 4d, the Arl4A signal served as the mask for the ROI per cell. VPS36-Arl4A colocalized pixels/total VPS36 pixels within the mask were calculated to obtain the colocalization coefficients in the images. Similar settings were used for VPS36-CD63 colocalized pixels in Fig. 7a, b and VPS36-CHMP2A colocalized pixels in Fig. 8a as numerators and total VPS36 pixels as the denominator. In Supplementary Figs. 2, 10, 11, and 12, the EGFR signals served as the mask for defining the ROI per cell. EGFR-EEA1 or -Lamp1 colocalized

pixels/total EGFR pixels within the mask were calculated as the colocalization coefficients.

In Fig. 5a, we used Imaris (Bitplane) software to quantify the 3D-colocalization of Arl4A and Lamp1 in super-resolution images. The Arl4A signals served as the mask for defining the ROI per cell. Using these masks, the 3D-colocalization coefficient values were calculated by Lamp1 pixels that colocalized with Arl4A/total Lamp1 pixels in these masks.

In Fig. 7b, we used Imaris (Bitplane) software to quantify the 3D-colocalization of VPS36 and GFP-Rab5 Q79L in super-resolution images. The GFP-Rab5 Q79L signals served as the mask for defining the ROI per cell. Using these masks, the 3D-colocalization coefficient values were calculated by VPS36 pixels that colocalized with GFP-Rab5 Q79L/total VPS36 pixels in these masks.

## EGFR trafficking assay
Serum-starved HeLa cells were treated with EGF (80 ng/ml) and EGFR antibody (18 μg/ml) (anti-extracellular region of EGFR, #ab30), fixed at indicated times, and stained with secondary antibody anti-mouse Alexa-488 and DAPI, as previously described[64]. The EGFR intensity was determined using a nonlinear one-phase decay fit of the time course.

## Cell proliferation assay
We evaluated the effect of Arl4A depletion on C33-A cell proliferation using a commercial BrdU incorporation assay kit (Merck Millipore, BrdU Cell Proliferation Kit). C33-A cells were seeded on 96-well cell culture plates at a density of 5000 cells per well. After 24 h, the cells were exposed to EGF (100 ng/ml) to induce cell proliferation for 24 h. Subsequently, the assay was carried out according to the protocol outlined by the manufacturer. The absorbance of each well was detected at dual wavelengths of 450–540 nm using a Thermo Fisher Scientific Multiskan™ GO plate reader.

## Apoptosis assay
Cells were stained with an annexin V-FITC apoptosis detection kit (Invitrogen), and apoptotic cells were identified and quantified by flow cytometry according to the manufacturer's instructions. In brief, after depletion of Arl4A, the cells were washed with PBS and collected via trypsin-EDTA solution (Invitrogen). The cell suspensions were then centrifuged at 1000 rpm for 5 min to remove the trypsin-EDTA solution. Then, the cells were resuspended and incubated with propidium iodide (PI), annexin V-FITC, and annexin V binding buffer for 15 min at room temperature. The stained cells were analyzed via flow cytometry with Muse Cell Analyzer system (Luminex).

## WST-1 assay for cell viability
In the WST-1 assay, the same number of cells from different experimental groups was seeded in a 96-well plate and WST-1 reagent (TOOLS; #TAAR-WBF9) was added to each well according to the manufacturer's guidelines. The plate was incubated for 2 h at 37 °C and the absorbance at a wavelength of 450 nm was measured using a spectrophotometer. The absorbance was then used to determine the number of viable cells in the samples compared to the control groups.

## The Arl4A knockout mice
The Arl4Atm1a(EUCOMM)Hmgu ES cells were purchased from the European Mouse Mutant Cell Repository (EuMMCR). The Arl4A knockout mice were generated by the National Taiwan University College of Medicine Laboratory Animal Center. In this study, we adopted aged 6–7 months C57BL/6N Arl4a KO and WT mice. Male and female mice were used for the experiments. All mice were kept in a specific pathogen-free (SPF) room at temperature (22 ± 2 °C), and humidity (55 ± 10%) with a dark/light cycle (12/12 h), and received food and water provided ad libitum.

## Statistics and reproducibility
Statistical comparisons between treatments were performed by parametric *t*-tests (Student's *t*-tests) or one-way analysis of variance (ANOVA) in GraphPad Prism 8. For the post hoc test, we used Tukey's method to compare all possible group pairings. Significant differences are indicated in the figure (*$P < 0.05$. **$P < 0.01$. ***$P < 0.001$. ****$P < 0.0001$). All experiments were performed at least three times unless otherwise indicated, and the figures show representative results. Figs. 4a, b and 6a were independently repeated twice with similar results. Fig. 9c and f were independently repeated three and five times, respectively, with similar results.

## Reporting summary
Further information on research design is available in the Nature Portfolio Reporting Summary linked to this article.

## Data availability
All data generated or analyzed during this study are included in this published article and its supplementary information files. Source data are provided with this paper.

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

## Acknowledgements

We thank Drs. Joel Moss, Randy Haun, Chia-Jung Yu, and Ya-Wen Liu for their critical review of this manuscript. We also thank Drs. Chia-Jung Yu, Ko-Jiunn Liu, Gee-Chen Chang, and Li-Ting Jang for generously providing us with cell lines and plasmids in this study. We also thank the Transgenic Mouse Core Facility at National Taiwan University for providing MEFs from Arl4A$^{-/-}$ mice and the imaging core and biomedical resource center at the First Core Labs, National Taiwan University Medicine College for technical assistance. This work was supported by grants from the National Health Research Institutes (NHRI) of Taiwan (NHRI-EX110-10902B1), the Ministry of Science and Technology in Taiwan (NSTC111-2634-F-002-017), and the Center of Precision Medicine from the Ministry of Education in Taiwan awarded to F.S.L.

## Author contributions

S.-J.L., M.-C.L., and F.S.L. designed the study and interpreted the results; S.-J.L. performed the majority of the experiments and analyzed the data; S.-J.L., M.-C. L., T.-J.L., Y.-T.T., and M.-T.T. contributed new reagents/analytic tools and conducted and supported the biological experiments; S.-J.L., M.-C. L., and Y.-T.T. prepared the draft of the manuscript; S.-J.L., M.-C. L., and F.S.L. wrote and edited the manuscript; and F.S.L. provided supervision, acquired funding, and performed project administration.

## Competing interests

The authors declare no competing interests.
