## [Peer Review File · Nature Communications]

Endosomal Arl4A attenuates EGFR degradation by binding to the ESCRT-II component VPS36REVIEWER COMMENTS

Reviewer #1 (Remarks to the Author):

The authors provide evidence linking (Arf)-like protein 4A with the regulation of EGFR degradation in a mechanism that involves interaction with the ESCRT complex member VPS36 to control EGFR deubiquitination. While I found this report of interest with some novel information regarding the role of Arl4A in EGFR trafficking and regulation of ESCRT complex function, there are some points that will require clarification. I also found at times the report quite dense with some figures not following the order of appearance in the text, so the manuscript could be improved with a good degree of rewording and reorganisation.

The major points that need experimental attention are as follows:

In Figure 1, fluorescent EGF is not a good measure of receptor degradation as the fluorescent moiety might continue to fluoresce even after protein breakdown. Reduced levels could also be the consequence of reduced ligand binding. The authors could use for example SNAP-tagged EGFR as a better alternative to follow its trafficking (there are cleavable SNAP tag probes or surface impermeable ones that can be used to answer this question). Moreover, binding affinity measurements could be performed to discard this possibility.

Arl4A KD levels should be assessed by WB.

EGFR staining profile looks strange at 0 minutes. The receptor should be localised at the plasma membrane. The Ab used should be validated in the manuscript with a control experiment. Experiments with SNAP-tagged receptor would help answer this question, as well as quantification of internalisation with a total versus surface receptor level assessment.

Need to clarify n numbers: are these cells from the same experiment or from individual experiments that can be considered biological repeats?

Rescue experiment don't seem to work for c-Met in Cos-7 or for either in HeLa but this is not discussed and no reason is given to explain this.

"siRNA-resistant construct Arl4AWT": can they clarify the name? Is WT not wild type? why is it siRNA-resistant?

Results at 0 minutes on Fig 1F don't match those of Fig 1E? Same problem in fig 2A and the rest of Fig 2

Increased apoptosis is worrying - does this happen in absence of egfr? Might be a general apoptosis phenotype rather than egfr specific? Specificity issue should be discussed when suggesting Arl4A KD as a treatment strategy.

Suppl Fig 8: why is wt egfr in endosomes?
PE089 pictures don't match quantification given for LAMP1 co-localisation.

Fig 3: why is Arl4A over expression not performed in C33-A?

Fig 3E needs a much higher resolution to be able to observe ILVs: need to employ an EM strategy or super-resolution microscopy

Fig4: what about the mirystoylation mutant?

Fig5c is not clear: again need EM or alternative approach

Fig6: why was A3 and A4 not further investigated?

Suppl fig 10 not convincing: EM or pulldowns are needed

Minor point:

Page 11, line 372, "Beccause" - please correct.

Reviewer #2 (Remarks to the Author):

In the study by Lin et al., the authors provide evidence to suggest that downregulation and eventual degradation of ubiquitin-modified EGFR via the endo-lysosomal sorting pathway is modulated by the expression of Arl4A. Specifically, they observe that depletion of Arl4A accelerates EGFR degradation and overexpression of Arl4A increases the half-life of EGFR following EGF stimulation. Additionally, they find that overexpression of Arl4A somehow inhibits the interaction between the ESCRT-III subunit CHMP2A and the deubiquitinating enzyme USP8/UBPY, suggesting that overexpressed Arl4A indirectly slows EGFR deubiquitination. However, no mechanistic understanding of this inhibition is explored further, which is a major shortcoming of the study. Instead, the authors turn to another finding that Arl4A binds to the ESCRT-II subunit Vps36 via a conserved motif, and disruption of this association (by mutating the Vps36 domain that binds to Arl4A) enhances the rate of EGFR degradation, mimicking the effect of Arl4A depletion. Based on these data, the authors conclude that the interaction between Arl4A and Vps36 decreases EGFR deubiquitination. Unfortunately, this conclusion represents an overinterpretation/overextrapolation of the data presented, and in its current state, the study appears too preliminary for further consideration.

Major concerns:

1. Although defects in ESCRT-mediated MVB formation lead to the accumulation of ubiquitinated substrates, it is unclear why at a mechanistic level a delay in EGFR deubiquitination specifically would slow its internalization into MVBs. There are many examples of ubiquitinated substrates still being internalized into MVBs.
2. Is the delay in EGFR degradation observed in Arl4A depleted cells similar to that of USP8/UBPY depletion? Can overexpression of USP8/UBPY suppress the delay in EGFR degradation observed in Arl4A depleted cells?
3. It is unclear how overexpression of Arl4A inhibits the interaction between CHMP2A and USP8/UBPY. This observation needs to be better defined at a mechanistic level to link Arl4A to deubiquitination of EGFR.
4. How does the interaction between Arl4A and Vps36 relate to the effect of Arl4A overexpression on the interaction between CHMP2A and USP8/UBPY. Without a better understanding of this interplay, the

study fails to provide a mechanistic understanding of the role that Arl4A plays in directly regulating EGFR degradation.

Reviewer #3 (Remarks to the Author):

EGFR is a potent regulator of cell proliferation and survival. Upon binding ligand, the magnitude and duration of EGFR is regulated by several factors, including signal termination resulting from receptor internalization and degradation in lysosomes. While some aspects of EGFR signaling regulation have been studied, the factors that regulate EGFR degradation remain incompletely understood. This study examines the role of the GTPase Arf4A in the regulation of EGFR degradation.

Arf4 silencing enhanced the internalization of fluorescent EGF, used to label ligand-bound EGFR, to early endosomes and lysosomes and also decreased the expression levels of EGFR and other signaling receptors such as Met. The loss of EGFR expression upon Arl4A silencing could be rescued by inhibitors of lysosomal acidification but not of the proteasome. The loss of EGFR expression upon Arl4A silencing was observed in cell lines that express wild type as well as cancer associated EGFR mutants. Arl4A overexpression increased and Arl4A silencing decreased the duration of EGFR ubiquitinylation following stimulation with EGF. Arl4A over-expression decreased the association of the de-ubiquitinylation enzyme USP8 with the ESCRT component CHMP2A. Consistent with a role for Arl4A in regulating EGFR ubiquitinylation and ESCRT-mediated sorting of EGFR, silencing Arl4A impaired the sorting of EGFR into the lumen of enlarged endosomes formed by expression of mutant Rab5.

Arl4A associated with the ESCRT-II component Vps36 in a GTPase-independent manner, as determined by co-IP and some immunofluorescence co-localization analysis. EGF stimulation increased Arl4A association with GTP as determined by PAK-pulldown and also increased Arl4A association with LAMP1 (determined by colocalization analysis). Scanning mutagenesis of Vps36 coupled to yeast 2 hybrid revealed regions essential for interaction with Arl4A, and a Vps36 mutant of this region exhibited reduced co-precipitation of Arl4A and Vps36. While silencing of Vps36 resulted in delayed EGFR degradation, the rescue of this effect was more pronounced upon expression of Arl4A defective in Vps36 interaction than wild-type Arl4A.

The study concludes and presents a model that Arl4A is a novel regulator of EGFR degradation and sorting to the lysosome, due to the association of Arl4A with ESCRT-II, which in turn allows regulation of EGFR ubiquitylation. In general, the study is well-designed and the experiments are carefully performed. The rescue of silencing experiments (e.g. Arl4A) with both wild-type and mutant constructs adds significantly to strengthen the conclusions and provide mechanistic understanding of the role of Arl4A. The role of Arl4A in regulation of EGFR degradation is novel and will be of interest to a broad readership. For the most part, the conclusions are well-supported by the data, with the exception of the conclusions pertaining to a specific role of Arl4A in controlling EGFR deubiquitylation, which are too strong for the data shown. Some comments should be addressed prior to publication.

Major comments:

1. The overexpression and silencing of Arl4A to monitor EGFR ubiquitylation are each done in different cell lines (Figure 3A-B). This is understandable since HeLa cells appear to not express Ar4A and are

thus suitable to study the effect of (over) expression of Arl4A, while C33A cells express Arl4A and are thus suitable to study the effect of Arl4A silencing. It might be useful to show a comparison of the relative levels of Arl4A in HeLa cells upon overexpression relative to that of C33A cells.

2. For many of the experiments (Fig. 3E, 4D, 5A, 5C), it is not clear what the individual data points refer to. Is this data from individual cells from a single experiment? Statistical analyses should only be applied to measurements from independent experiments, not on measurements obtained from individual cells from a single experiment.

3. The blots shown in Figure 3D are not of great quality and should be replaced.

4. In Figure 5D, could the authors comment on why 15 min was selected? It seems from Figure 5C that Arl4A and LAMP1 may exhibit higher overlap at later timepoints of EGF stimulation.

5. Some of the immunofluorescence experiments showing colocalization do not show evident overlap of the Arl4A and the other marker(s) examined. For example, Figure 4D (Arl4A WT), Figure 5A (Arl4A WT), Figure 5C, etc. It would be helpful to clarify how the Pearson's coefficient analysis was conducted (how ROIs were selected for analysis, etc). Showing some control analyses (for example, on pairs of images in which one of the channels is randomized by rotation 180 degrees or similar) would be helpful to delineate what is indeed colocalization. Also, some of the colocalization of Vps36 and Arl4A appear to have a Pearson's coefficient of nearly 1.0 (Figure 4D) in the Arl4A T51N condition – this would nearly perfect correlation of fluorescence intensities of these two channels, and it is difficult to imagine how this could be the case. These colocalization experiments should be carefully reviewed and revised.

6. Are there any differences in expression of (myc)-tagged Arl4A mutants versus wild-type?

7. The C33A cell line appears to have very little EGFR (Figure S2A). Are these cells EGFR dependent under the conditions examined, for cell survival or proliferation? Some of these experiments were conducted under conditions in which EGF is added at 100 ng/mL, which should lead to rather rapid downregulation of EGFR near the start of the observation period. Is the effect seen under these conditions (BrdU, or apoptosis) sensitive to EGFR inhibition by TKIs? This would be relevant to allow interpretation of the results shown in Figure S7. It is difficult to demonstrate that the effects of Arl4A siRNA on cell viability are due to the regulation of EGFR by Arl4A, but the dependence of cell viability on EGFR under the conditions examined should at least be included.

8. It is not clear how the enhanced arrival of EGFR to EEA1 endosomes (Figure S1A) at upon Arl4A silencing is related to a role for Arl4A in ESCRT sorting or EGFR deubiquitylation. Could the authors comment on this?

9. In figure S2A, there appears to be very little Arl4A expressed in HeLa, A549, H1975 and PE089 cells. Nonetheless, there are effects of Arl4A knockdown for each of these cell lines shown in Figure 2. The detection of mRNA levels of Arl4A upon knockdown are helpful to appreciate that knockdown took place. The lack of evident detection of Arl4A protein in these cells makes some of the interpretation of the data less straightforward.

10. The statement in the discussion on line 370: "The present study revealed that Arl4A acts as a negative regulator of EGFR" is confusing. For example, some of the data is summarized by the statement on line 357-358 "...our results show that lack of Arl4A results in strongly reduced levels of EGFR..." – this indicates that Arl4A is a positive regulator, at least of EGFR expression levels. Most of

the experiments in this paper demonstrate that Arl4A enhances EGFR expression, consistent with enhanced EGFR function.

11. The statement on line 188 that "Arl4A depletion may reactivate EGFR mutant degradation..." could be revised. It is not clear that EGFR degradation is a binary switch that can only be in the on/off state and it is not clear that Arl4A silencing counteracts the specific perturbation of EGFR degradation seen in mutant receptors. More accurate would be to just state that EGFR degradation is enhanced by Arl4A depletion even in cells with EGFR mutations that have intrinsically reduced EGFR degradation rates.

12. In Figure S8A, what is the time of EGF stimulation? This is important to allow understanding of how this relates to the distinct effects of Arl4A silencing on EEA1 overlap of EGFR following 5 vs 15 min shown in Figure S1A.

13. A description of the quantification for the localization of EGFR to the lumen of enlarged endosomes shown in Figure 3E should be added beyond the brief statement in the figure legend for Figure 3E. For instance, was the lumen of the enlarged endosomes identified manually? What is the size distribution of these ROIs selected to represent the endosome lumen in each condition studied?

14. It is perhaps more accurate (lines 331-333) to say of the experiments in Figure 7 that Arl4A binding to Vps36 regulates EGFR ubiquitinylation levels, rather than extending this conclusion to state that this is specifically due to EGFR deubiquitylation modulation. A similar tempering of conclusions in the Results and Discussion section that indicate that effect of Arl4A is to regulate EGFR deubiquitylation is suggested throughout the manuscripts. This is due to the evidence to support this being largely limited to a displacement of USP8 from co-IP with CHMP2A with Arl4A overexpression (Figure 3C), which could be impacted by overexpression of these proteins and the effect of silencing of Arl4A on EGFR ubiquitinylation duration, which could be affected by either ubiquitylation or de-ubiquitylation.

Minor:

1. The sentence in lines 283-284 is awkward and difficult to understand: "Using an in vitro GST pulldown assay, we also found that GST-VPS36-WT-(330-386) interacted with Arl4A but not with the A3, A4, or A6 VPS36 mutants (Fig. 6B)"

Reviewer #4 (Remarks to the Author):

In the manuscript "Endosomal Arl4A attenuates EGFR degradation by delaying its deubiquitination at the late endosome", Lin et al. report that the small GTPase Arl4A reduces EGFR degradation by binding to the ESCRT-II component VPS36. They further describe that Arl4A delays EGFR degradation by decreasing the deubiquitination of the receptor at late endosomes.

The findings reported in the paper are novel and of significance to the field; nevertheless, many of the data in the current form are not convincing and not able to support the authors' claims and conclusions. There are fundamental issues with the data presentation and interpretation. Specific points are listed below:

1) One of the main issue concerns all IF results presented in the manuscript. First, it is not clear whether the data presented derive from three independent experiments or from only one. This should be specified. If they originate from a single experiment, this is not sufficient to make any conclusion and the experiments should be repeated at least two more times. If they originate from at least three

independently performed experiments, the results from independent experiments should not be pooled for statistical analysis. Second, the image resolution is very low. For example, it is not possible to see the red signal on the merged, not magnified, images in suppl. Fig.8. Nevertheless, even when it is clearly visible that there is no (or poor) colocalization, the conclusion is often misleading. For example, the statement in lines 197-199: "Colocalization between EGFR and the late endosomal/lysosomal marker Lamp1 was barely observed in the control cells, suggesting that EGFR remains in early endosomal compartments (Supplementary Fig. 8A-B)." is not supported by data. Indeed, in Supplementary Fig. 8, the colocalization measured between EGFR and early endosomes is equally low as the colocalization between EGFR and late endosomes for control cells.

In the same figure, in PE089 cells, it is visible that EGFR colocalizes similarly with EEA1 in control and siArl4A cells, even though the EGFR signal is barely visible in siArl4A cells. It is not clear why the EGFR signal is barely visible in siArl4A cells stained for EEA1, but much stronger in the same cells stained for Lamp1 (but the Lamp staining is poorly visible). Based on these images, low Pearson's correlation coefficient, and lack of information about the number of experiments/statistics, it is impossible to draw any conclusion, including what is written in line 997, that "Depletion of Arl4A promotes EGFR mutant trafficking to lysosomes". It might help to repeat these experiments by stimulating the cells with EGF as done in suppl. Fig.1.

2) Another concern is that the results are based on the use of only one siRNA targeting Arl4A (only one experiment in suppl. Fig. 3 and 5 shows that the reintroduction of Arl4A rescues the defect). Arl4A rescue experiments need to be performed more consistently throughout the study (as done for Vps36), or at least a second siRNA targeting Arl4A should be included to confirm the specificity of the siRNAs used in this study.

3) There are three isoforms of Arl4 (Arl4A, Arl4C, and Arl4D). Is the Arl4A siRNA used in this study specific for Arl4A isoform only?

4) In many experiments, the WBs including control siRNA and Arl4A siRNA (or control and Arl4A overexpression) are shown on separate panels (e.g. Fig. 1f-g, Fig.3, suppl. Fig. 6A, etc.). For a direct comparison of the levels of Arl4A in the different conditions, these samples should be presented on the same membrane and not on separate panels.

5) Why the EGFR receptor is often barely degraded in control cells even at late time points (e.g. in Fig.1 f, Fig. 2, etc.)? And why two different concentrations of EGF (100 or 50 ng/ml) are used to stimulate the cells in different experiments?

6) It is not clear why in suppl. Fig. 6B, p-EGFR level relative to tubulin graph, the curve relative to siArl4A decreases over time, while in the blot shown in suppl. Fig. 6A there is a clear increase in the levels of p-EGFR at 15 minutes.

7) The only image showing that EGFR is present in Arl4A-positive endosomes is Fig.3E. However, in this experiment, the authors also express the constitutively active mutant of Rab5, which is very well known to dramatically affect the endosomal pathway and the sorting of EGFR (e.g. Wegener et al, 2010). GFP-Rab5 wt should be included as control.

In addition, this image is not fully supporting the conclusion stated in lines 233-236 as in the control cells are also visible enlarged endosomes without EGFR in the lumen, and in the siArl4A cells are also visible enlarged endosomes with EGFR in the lumen. In addition, the analysis is also performed on only 18 cells, and as for the other IF experiments, it is not specified whether the results are from one experiment or more.

8) In some experiments, both GTP- and GDP- bound Arl4A interacts with Vps36. However, in the co-IP (and IF), Vps36 seems to preferentially interact with the dominant negative mutant. How do the authors explain this? In addition, according to the analysis in Fig. 5C, there is an increase in colocalization between Arl4A and Lamp-1 45 and 60 minutes after EGF stimulation, and Fig. 5B shows that Arl4A is more active at these time point. How does this fit with the statement that the dominant negative mutant is mostly associated with the lysosome? Also, the images in Fig.5A do not support this conclusion, as no Arl4A T51N-positive vesicles are present in the magnified green channel. It is not clear how the authors measure colocalization, as the signal from Arl4A T51N in the magnified

region shown is clearly cytosolic and not overlapping with Lamp-1-positive vesicles, even though the artificial-coloured image seems to illustrate that. It seems that this is rather the consequence of measuring the background/cytosolic signal due to a too low threshold in the image analysis than a real result. The same for Fig.5C, where the Pearson's correlation coefficient is close to zero for the different conditions, indicating almost no colocalization. As many of the images seem to have a low signal-to-noise ratio, the colocalization values measured may simply be the consequence of a very low threshold level. It may help to reduce the Arl4A cytosolic signal by permeabilizing the sample before fixation.

9) Line 319: the conclusion "the VPS36-A6 mutant exhibited decreased colocalization with CHMP2A" is not supported by data as in the images in suppl. Fig. 10C, CHMP2A-positive vesicles are all clearly positive for Vps36 (wt or mutant).

10) In many IP experiments, a proper negative control is missing, making difficult to evaluate the specificity of the results. For example, in Fig. 3C-D and Fig. 4C, the negative control would include cells transfected with an empty HA-vector (and not untransfected cells), and in Fig. 5D, 6D, 9A and 9C, cells transfected with an empty myc-vector. It would also be interesting to verify whether the silencing of Arl4A increases USP8 recruitment by CHMP2A.

11) Similarly, a negative control is missing in the pull down experiment in Fig. 4B. Also, the entire comassie gel for all the GST- and His- tagged purified proteins should be included to evaluate their purity. The pulldown shown in Fig.6B should include anti-His or anti-Arl4A staining for the pull down membrane containing all the samples and not only the input, to verify that the amount of the protein present in the different sample is comparable.

12) The membrane shown in Suppl. Fig. 9a should include anti-Arl4A to verify that the lack of difference is not a consequence of a poor silencing. Arl4A levels should also be shown in Suppl. Fig. 4.

13) The authors use different cell lines for different experiments (e.g. HeLa, 293T, C33A, Cos-7, etc.), and it is not often clear why some experiments are performed with one particular cell line and others with a different cell line. This should be clarified. For example, why in Fig.1 or Fig. 3 the EGFR degradation upon Arl4A overexpression is performed in HeLa while EGFR degradation upon Arl4A silencing is performed in C33A cells? Why 293T cells are used for the experiment in fig. 3D?

14) Not sufficient details are provided in the methods for the work to be reproduced. For example, information about the siRNAs used in this study is not provided. It is also not described which point mutations have been introduced to generate the RNAi-resistant clones. The description about how image analysis has been performed is also missing.

Minor comments:

-line 51: the references are separated

- In Figure 6B, and in the relative figure legend, it is mentioned GST-VPS36 (201-386), however in the main text (line 283), this is referred as GST-VPS36 (330-386).

- line 302 (and relative figure legend): expression of Arl4A dominant negative mutant (and not overexpression)

- line 372: "because"

-line 433: "Arl4Acan"

- line 866: Arl4A WT is not the GTP-bound form.

-line 953: "Cell lysates were analyzed by immunoblotting with anti-Arl4A", but this is not shown in suppl. Fig. 3D. It should be included.

- line 956: "Arl4a"

- lines 1023-1025: The immunoprecipitation shown in suppl. Fig. 9c is not a GFP-TRAP. The figure legend should be corrected.

May 5, 2023

Nature Communications (NCOMMS-122-42833)

Title "Endosomal Arl4A attenuates EGFR degradation by delaying its deubiquitination at the late endosome"

Revised title "Endosomal Arl4A attenuates EGFR degradation by binding to the ESCRT-II component VPS36"

Specific response to the reviewers

Reply to Reviewer #1

Reviewer #1 (Remarks to the Author):

The authors provide evidence linking (Arf)-like protein 4A with the regulation of EGFR degradation in a mechanism that involves interaction with the ECSRT complex member VPS36 to control EGFR deubiquitination. While I found this report of interest with some novel information regarding the role of Arl4A in EGFR trafficking and regulation of ESCRT complex function, there are some points that will require clarification. I also found at times the report quite dense with some figures not following the order of appearance in the text, so the manuscript could be improved with a good degree of rewording and reorganisation.

Response: We thank the reviewer for the positive comments and opportunity to improve the manuscript. As suggested by the reviewer, we have reworded and reorganized the revised manuscript. We also appreciate the comments aimed at improving the manuscript, and we have addressed these below, both in the comments for the author as well as the major and minor comment sections.

Major comments:

In Figure 1, fluorescent EGF is not a good measure of receptor degradation as the fluorescent moiety might continue to fluoresce even after protein breakdown. Reduced levels could also be the consequence of reduced ligand binding. The authors could use for example SNAP-tagged EGFR as a better alternative to follow its trafficking (there are cleavable SNAP tag probes or surface impermeable ones that can be used to answer this question). Moreover, binding affinity measurements could be performed to discard this possibility.

Response: We thank the reviewer for these comments. SNAP-tags indeed have numerous applications in biochemistry and for the investigation of the function and localization of proteins in living cells. Although SNAP-fused EGFR has been used to screen drug leads (Fu et al. 2021), we did not find published papers using SNAP-fused EGFR to study the EGF-

induced EGFR endocytic route. This may be because the 182 residue polypeptide length of the SNAP-tag may interfere with EGF binding to the EGFR and/or EGF-induced EGFR signaling or degradation. In addition, a previous study indicated that selection of the dye for labeling SNAP-fusion proteins is crucial, as many dyes suffer from either rapid photobleaching or high levels of nonspecific staining (Bosch et al., 2014). Therefore, we used an anti-EGFR external domain antibody to measure EGFR fate during the endocytic route as described by Savio et al. (2016).

As shown in the **new Supplementary Fig. 1**, we found that the half-life of EGFR was decreased in Arl4A-depleted cells compared to siControl cells after EGF treatment. The result, as shown in Fig. 1, supports the notion that Arl4A depletion enhances EGFR transport for degradation.

References:

Fu J, Jia Q, Liang P, et al. Targeting and Covalently Immobilizing the EGFR through SNAP-Tag Technology for Screening Drug Leads. *Anal Chem.* 2021;93(34):11719-11728.

Bosch PJ, Corrêa IR Jr, Sonntag MH, Ibach J, Brunsveld L, Kanger JS, Subramaniam V. Evaluation of fluorophores to label SNAP-tag fused proteins for multicolor single-molecule tracking microscopy in live cells. *Biophys J.* 2014 Aug 19;107(4):803-14.

Savio MG, Wollscheid N, Cavallaro E, Algisi V, Di Fiore PP, Sigismund S, Maspero E, Polo S. USP9X Controls EGFR Fate by Deubiquitinating the Endocytic Adaptor Eps15.

Curr Biol. 2016 Jan 25;26(2):173-183.

Arl4A KD levels should be assessed by WB.

Response: We thank the reviewer for this suggestion. Using our Arl4A-specific antiserum in western blotting analysis, we previously showed that endogenous Arl4A could only be detected in cervical cancer C33A cells among 21 human cancer cell lines examined (Chen et al., 2020) (**Fig. R1A, for reviewer only**). This is why we had difficulty detecting endogenous Arl4A when we applied 20-30 µg total protein from HeLa, A549, H1975, and PE089 cell lysates, as shown in Fig. 2.

To overcome the low abundance of Arl4A in these cells, we increased the cell lysate amounts to 120 µg total protein/lane for western blotting analysis in Arl4A knockdown cell lysates. With this extremely high amount of cell lysates, we detected endogenous Arl4A in these cell lines (**Fig. R1B, for reviewer only**). Western blotting analysis of HeLa, A549, H1975, and PE089 cells has been added to the **new Supplementary Figs. 1, 9, 10, and 11**.

Fig. R1. Detection of endogenous Arl4A. (A) Endogenous Arl4A could only be detected in cervical cancer C33A cells among 21 human cancer cell lines examined. Approximately 20 μ g of total protein from each cancer cell line lysate was used for western blot analysis. (B) Approximately 120 μ g of total protein from HeLa, A549, H1975, and PE089 cell lysates was used for western blot analysis.

Reference:

Chen KJ, Chiang TC, Yu CJ, Lee FS. Cooperative recruitment of Arl4A and Pak1 to the plasma membrane contributes to sustained Pak1 activation for cell migration. *J Cell Sci.* 2020;133(3):jcs233361.

EGFR staining profile looks strange at 0 minutes. The receptor should be localized at the plasma membrane. The Ab used should be validated in the manuscript with a control experiment. Experiments with SNAP-tagged receptor would help answer this question, as well as quantification of internalization with a total versus surface receptor level assessment.

Response: We thank the reviewer for the suggestion. In this figure, our main purpose is to determine the colocalization coefficient between EGFR and EEA1 by confocal microscopy. The confocal z-stack figure indicated that EGFR localization at the plasma membrane could not be observed at certain focal planes that do not focus on the plasma membrane. Specifically, the EEA1 signal is clearest in the upper focal plane, whereas the EGFR plasma membrane signal usually appears at the bottom focal plane. Therefore, the EGFR staining profile at the plasma membrane could not be clearly observed at 0 minutes.

For example: The confocal z-stack images below (our data in Fig. R2A, for reviewer only) show that the EGFR antibody (Cell Signaling Technology D38B1 XP® Rabbit mAb #4267) can detect the plasma membrane signal of EGFR at the bottom layer, but becomes blurry at the top layer. The same EGFR antibody (CST #4267) has also been used to observe EGFR localization at the plasma membrane (Fig. R2B) (Yang et al., 2022) and its specificity was confirmed by showing the lack of EGFR signal in EGFR-knockout HeLa cell lysates (Fig. R2C, data from Cell Signaling Technology). In response to the reviewer's first comment, we have used an anti-EGFR external domain antibody to measure EGFR levels during the endocytic

route and determined the levels of EGFR per cell at different time points by immunofluorescence staining in the new Supplementary Fig. 1. The data showed EGFR plasma membrane signals at time 0 in both siControl and siArl4A cells.

Fig. R2. EGFR staining profile and antibody specificity. (A, B) The EGFR antibody (CST #4267) has been successfully used to detect EGFR at the plasma membrane by confocal microscopy. (A) Our data in staining steady-state EGFR and EEA1 in HeLa cell monolayers. (B) Steady-state HeLa cell monolayers stained with EGFR antibody (Cells. 2022;11:3358). (C) The absence of an EGFR signal in EGFR-knockout HeLa cell lysates confirms the specificity of the EGFR antibody (data from Cell Signaling Technology). Western blot analysis of cell lysates from control HeLa cells (lane 1) or from EGFR knockout HeLa cells (lane 2) using EGFR (D38B1) XP® Rabbit mAb #4267.

References:

- Yang Z, Feng Z, Li Z, Teasdale RD. Multifaceted Roles of Retromer in EGFR Trafficking and Signaling Activation. *Cells*. 2022;11(21):3358.
- Rivera C, Lee HG, Lappala A, et al. Unveiling RCOR1 as a rheostat at transcriptionally permissive chromatin. *Nat Commun*. 2022;13(1):1550.
- Al-Akhrass H, Naves T, Vincent F, et al. Sortilin limits EGFR signaling by promoting its internalization in lung cancer. *Nat Commun*. 2017;8(1):1182.

Need to clarify n numbers: are these cells from the same experiment or from individual experiments that can be considered biological repeats?

Response: We thank the reviewer for the comments. All the dot plots in this manuscript represent the results from at least 3 different biological repeats of the experiments. We have adjusted our quantified figures into super plots as previously described (Lord et al., 2020) to visually represent and compare large amounts of data from multiple experiments. We also

clarified the n numbers calculated in the plots in all the figure legends.

Reference:

Lord SJ, Velle KB, Mullins RD, Fritz-Laylin LK. SuperPlots: Communicating reproducibility and variability in cell biology. *J Cell Biol.* 2020;219(6):e202001064.

Rescue experiment don't seem to work for c-Met in Cos-7 or for either in HeLa but this is not discussed and no reason is given to explain this.

Response: Thank you for the comment. A previous study showed that c-Met protein cannot be detected in COS-7 cells (Higuchi et al., 1992). Similar to their findings, we did not detect the c-Met signal in COS-7 cells by western blotting analysis. To avoid misunderstanding, we deleted the c-Met data in COS-7 in the **revised Supplemental Fig. S4c**.

For the rescue experiment of c-Met in HeLa cells, we acknowledge that expressing Arl4A^{Res} can only achieve 58% rescue of c-Met in Arl4A-depleted HeLa cells. We have modified our discussion, and a detailed description is presented in the revised manuscript (Lines 125-127).

Reference:

Higuchi O, Mizuno K, Vande Woude GF, Nakamura T. Expression of c-met proto-oncogene in COS cells induces the signal transducing high-affinity receptor for hepatocyte growth factor. *FEBS Lett.* 1992;301(3):282-286.

“siRNA-resistant construct Arl4AWT”: can they clarify the name? Is WT not wild type? why is it siRNA-resistant?

Response: Thank you for your kind reminder. We have added details to explain how we constructed an RNAi-resistant clone of Arl4A. The Arl4A RNAi-resistant clone contains silent point mutations in the RNAi targeting sequence that decrease siRNA binding to the mRNA. Therefore, the protein sequence of the Arl4A-resistant clone is exactly the same as that of the Arl4A wild type (WT). In the main text and in the figures, we carefully labeled this kind of resistant clone Arl4A-WT^{Res}, and we also noted the meaning of this name in the figure legends. The pSG5-Arl4A siRNA-resistant construct was generated by two-step site-directed mutation primers.

The primer sequences for PCR are as follows:

F: 5'-CAAGATTTGAGAACTCTTTGTCAGTGTCTGTTGG-3'

R: 5'-CTCAATTTCTGACAGTGACAAAGAGTTTCTCAAATCTTG-3'.

Similarly, the pcDNA3.1 VPS36 siRNA-resistant PCR primer sequences are as follows:

F: 5'-TTGCTGAGCATGGGCATCGCGAACCCAGTTACCAGAGAAACC-3'
R: 5'-TTGCTGAGCATGGGCATCGCGAATCCGGTGACCAGAGAAACC-3'.

Results at 0 minutes on Fig 1F don't match those of Fig 1E? Same problem in fig 2A and the rest of Fig 2

Response: Thank you for the comment. In Fig. 1E, knockdown of Arl4A decreased EGFR protein levels, which was also observed when we conducted the experiment in Fig. 1F. As shown below (Fig. R3 for reviewer only), if the same amounts of total proteins were loaded, there was a difference in EGFR levels between the siControl and siArl4A groups at 0 minutes. To determine the half-life of EGFR in Fig. 1F and Fig. 2B-D, we adjusted the EGFR protein to equal amounts between siControl and siArl4A at 0 minutes for western blotting. This makes the EGFR degradation rates between the two groups more comparable and easier to determine. This adjustment can avoid the argument that faster degradation is due to less total protein at time 0 or due to the limitation of Ab detection when the amount of protein is too low. This is the reason why we loaded different amounts of total protein on different SDS-PAGE gels for EGFR blotting in siControl and siArl4A and why we presented the EGFR western blots in separate images.

Fig. R3. Analysis of EGFR protein degradation in Arl4A knockdown cells. C33-A cells were transfected for 48 hours with siControl or siArl4A. Cells were sequentially subjected to EGFR degradation analysis by western blotting.

Increased apoptosis is worrying - does this happen in absence of EGFR? Might be a general apoptosis phenotype rather than EGFR specific? Specificity issue should be discussed when suggesting Arl4A KD as a treatment strategy.

Response: Thank you for the comments. We agree that the cell apoptosis caused by Arl4A knockdown is a general phenotype. Inhibition of c-Met oncogene signaling induces apoptosis in a variety of cancers. (Campos-Viguri et al., 2020; Hung et al., 2015; Wang et al., 2020; Zagouri et al., 2013). Previous studies also showed that reducing EGFR expression increases apoptosis (Okamoto et al., 2010; Yao et al., 2020). Our data showed that Arl4A knockdown decreased the protein levels of growth factor receptors, such as EGFR and c-Met. Therefore,

we conclude that increasing apoptosis in Arl4A-knockdown cells is a general, but not EGFR-specific, phenotype.

Furthermore, Arl4A plays different roles at different subcellular locations, such as binding to Pak1 or Robo1 at the plasma membrane to promote cell migration (Chen et al., 2020; Chiang et al., 2019) and binding to VPS36 at the endosome compartments to mediate receptor degradation. These Arl4A functions are related to cancer progression. In this study, knockdown of Arl4A also led to a decrease in proliferation and apoptosis. Therefore, we infer that inhibiting Arl4A might be applicable to cancer therapy. We have included these specificity issues in the discussion section.

References:

Campos-Viguri GE, Peralta-Zaragoza O, Jiménez-Wences H, et al. MiR-23b-3p reduces the proliferation, migration and invasion of cervical cancer cell lines via the reduction of c-Met expression. *Sci Rep.* 2020;10(1):3256.

Wang Y, Zhang H, Zhang Y, Li X, Hu X, Wang X. Decorin promotes apoptosis and autophagy via suppressing c-Met in HTR-8 trophoblasts. *Reproduction.* 2020;159(6):669-677.

Zagouri F, Bago-Horvath Z, Rössler F, et al. High MET expression is an adverse prognostic factor in patients with triple-negative breast cancer. *Br J Cancer.* 2013;108(5):1100-1105.

Okamoto K, Okamoto I, Okamoto W, et al. Role of survivin in EGFR inhibitor-induced apoptosis in non-small cell lung cancers positive for EGFR mutations. *Cancer Res.* 2010;70(24):10402-10410.

Yao N, Wang CR, Liu MQ, et al. Discovery of a novel EGFR ligand DPBA that degrades EGFR and suppresses EGFR-positive NSCLC growth. *Signal Transduct Target Ther.* 2020;5(1):214.

Chiang TS, Lin MC, Tsai MC, Chen CH, Jang LT, Lee FS. ADP-ribosylation factor-like 4A interacts with Robo1 to promote cell migration by regulating Cdc42 activation. *Mol Biol Cell.* 2019;30(1):69-81.

Chen KJ, Chiang TC, Yu CJ, Lee FS. Cooperative recruitment of Arl4A and Pak1 to the plasma membrane contributes to sustained Pak1 activation for cell migration. *J Cell Sci.* 2020;133(3):jcs233361.

Suppl Fig 8: why is wt egfr in endosomes?

Response: Thank you for the comment. Previous studies have shown that wild-type EGFR can colocalize with the early endosomal markers Rab5 and EEA1 without EGF treatment (Fig. R4 A and B)(Al-Akhrass et al., 2017; Yu et al., 2020). We observed that wild-type EGFR localized

at endosomes in the A549 cell line, which is consistent with previous findings. In addition, the cells in Suppl Fig 8 were cultured in medium supplied with FBS, which contains low amounts of EGF that can bind and activate EGFR at basal levels. Under these culture conditions, EGFR is endocytosed and transported to early endosomes. However, low concentrations of EGF do not trigger EGFR to move to the lysosomal degradation pathway; alternatively, EGFR is recycled to the plasma membrane once it reaches early endosomes (Tanaka et al., 2018). Thus, the localization of wild-type EGFR in the early endosomal compartments can indeed be observed in this culture condition.

Fig. R4. EGFR localized at endosomes. (A) EGFR can colocalize with Rab1 without EGF treatment in A549 cells. (B) EGFR can colocalize with EEA1 without EGF treatment in A549 cells.

References:

- Yu JJ, Zhou DD, Yang XX, et al. TRIB3-EGFR interaction promotes lung cancer progression and defines a therapeutic target. Nat Commun. 2020;11(1):3660.
- Al-Akhrass H, Naves T, Vincent F, et al. Sortilin limits EGFR signaling by promoting its internalization in lung cancer. Nat Commun. 2017;8(1):1182.
- Tanaka T, Zhou Y, Ozawa T, et al. Ligand-activated epidermal growth factor receptor (EGFR) signaling governs endocytic trafficking of unliganded receptor monomers by non-canonical phosphorylation. J Biol Chem. 2018;293(7):2288-2301.

PE089 pictures don't match quantification given for LAMP1 co-localisation.

Response: We apologize that the inappropriate selection of enlarged image in the representative PE089 panel do not match the quantification given for Lamp1 colocalization. In the original supplementary Fig. 8b, EGFR intensity was shown to be decreased after Arl4A knockdown. We originally selected an enlarged region in PE089 panel that had brighter EGFR in order to clearly see the colocalization of EGFR and Lamp1. This is why the inappropriate selection of the enlarged image in PE089 did not match the quantification given for Lamp1 colocalization.

In addition, the cells were grown in normal growth medium without the trigger from EGF, and EGFR colocalization with the endosomal compartment was therefore low overall for all of the cell lines we used. As requested by another reviewer, we performed experiments to show the colocalization of EGFR and EEA1/Lamp1 markers in A549, H1975, and PE089 cell lines at different time points after EGF treatment. These results are consistent with previous findings that Arl4A-depleted cells showed increased colocalization of EGFR and Lamp1 compared with control cells. In the revised manuscript, we have provided the data in the new Supplementary Figs. 9, 10, and 11 to replace Supplementary Fig. 8.

Fig 3: why is Arl4A over expression not performed in C33-A?

Response: Thank you for the comment. Previously, we showed that the expression of endogenous Arl4A in the cervical cancer C33-A cell line is the highest among 21 human cancer cell lines examined (Chen et al., 2020) and the cell lines we assayed in this study (Fig. S2A). The Arl4A protein level in C33-A cells is much higher than that in HeLa cells. Since the expression level of Arl4A in C33-A cells is high enough, we only performed Arl4A overexpression in HeLa cells (Fig. 3a, 3e) and Arl4A knockdown in C33-A cells.

For analyzing the effect of Arl4A overexpression on the CHMP2A-USP8 interaction in Fig. 3 C and D, we found that HeLa and C33-A cells did not express enough USP8 to perform IP experiments. We noticed that 293T cells have been widely used to efficiently produce proteins of interest (Tan et al., 2021). We then examined the protein expression level of USP8 in 293T, HeLa, and C33-A cell lines and confirmed that 293T cells efficiently express USP8 (Fig. R5, for reviewer only). Therefore, we chose an alternative cell line, 293T, to examine the effect of Arl4A overexpression on the CHMP2A-USP8 interaction.

Fig. R5. Exogenous expression of USP8-HA in 293T, HeLa, and C33-A cell lines.

References:

Chen KJ, Chiang TC, Yu CJ, Lee FS. Cooperative recruitment of Arl4A and Pak1 to the plasma membrane contributes to sustained Pak1 activation for cell migration. *J Cell Sci.* 2020;133: jcs 233361.

Tan E, Chin CSH, Lim ZFS, Ng SK. HEK293 Cell Line as a Platform to Produce Recombinant Proteins and Viral Vectors. *Front Bioeng Biotechnol.* 2021; 9:796991.

Fig 3E needs a much higher resolution to be able to observe ILVs: need to employ an EM strategy or super-resolution microscopy

Response: As suggested. We replaced the data in the revised Fig. 3c by using super-resolution microscopy to verify whether Arl4A prevents the sorting of EGFR into MVBs. The result is consistent with the original Fig. 3E.

Fig4: what about the mirystoylation mutant?

Response: As suggested, we performed immunofluorescence staining (IF) and observed that myristoylation mutant Arl4A G2A did not colocalize with VPS36 (revised Fig. 4d). We also validated the interaction between VPS36 and the G2A mutant by a coimmunoprecipitation assay. We observed that Arl4A WT, but not the Arl4A G2A mutant, can interact with VPS36 (Fig. R6, for the reviewer only). These data indicated that the membrane association ability of Arl4A is important for Arl4A-VPS36 colocalization and interaction. We added Arl4A G2A and VPS36 coexpression IF as the negative control in the new Fig. 4d.

Fig. R6. VPS36 interacted with Arl4A WT but not Arl4A G2A *in vivo*. Lysates of HeLa cells transfected with the indicated plasmids were immunoprecipitated with Myc-trap, and the bound proteins were analyzed by western blotting with anti-Myc and anti-Arl4A antibodies. The star (*) indicates the non-specific bands.

Fig5c is not clear: again need EM or alternative approach

Response: We appreciate the reviewer's suggestion. To advance the image resolution, we used super-resolution microscopy to provide new images in new Fig. 5a to replace both original Fig. 5a and 5c. We used Imaris (Bitplane) software to quantify the 3D colocalization of Arl4A and Lamp1 in super-resolution images. The quantified data showed that EGF treatment induces Arl4A transport to late endosomal compartments, where GDP-locked Arl4A is prone to localize. These results suggest that the Arl4A nucleotide binding state as well as its late

endosome distribution is regulated in response to upstream EGF signaling.

Fig6: why was A3 and A4 not further investigated?

Response: We thank the reviewer for this comment. Based on the conservation scores assessed by PRALINE software (Simossis and Heringa, 2005), we found that the conservation score of the A3 region is 5.2 ((6+4+4+5+7)/5), that of A4 is 6.4 ((6+7+7+5+7)/5), and that of A6 is 7 ((8+6+6+10+5)/5). In addition, the amino acid residue 355E in the A6 region is highly conserved in many species (see below). We therefore concluded that the region of A6 is more conserved than that of A3 and A4 and used the VPS36 A6 mutant for further functional studies.

Reference:

Simossis VA, Heringa J. PRALINE: a multiple sequence alignment toolbox that integrates homology-extended and secondary structure information. Nucleic Acids Res. 2005;33(Web Server issue): W289-W294.

Suppl fig 10 not convincing: EM or pulldowns are needed

Response: We appreciate the reviewer's comment and suggestion. As suggested, we performed an IP pulldown experiment to demonstrate that, compared to VPS36 WT, VPS36 A6 exhibited reduced coprecipitation of CHMP2A (new Fig 8b). To further demonstrate that VPS36 A6 exhibits decreased MVB localization, we used super-resolution microscopy-based images to determine the colocalization of VPS36 on Rab5 Q79L- enlarged MVBs. Imaris (Bitplane) software was used to quantify the 3D colocalization of VPS36 and Rab5 Q79L. Our data showed that VPS36 A6 is less capable of targeting MVBs than VPS36 WT (new Fig. 7c). These data support the notion that Arl4A binding can facilitate the anchoring of VPS36 to MVBs. We replaced the images in the original Supplementary Fig. 10 with the new Fig. 7.

Minor point:

Page 11, line 372, "Because" - please correct.

Thank you for the correction. The mistake has been corrected on line 366.

Reply to Reviewer #2

Reviewer #2 (Remarks to the Author):

In the study by Lin et al., the authors provide evidence to suggest that downregulation and eventual degradation of ubiquitin-modified EGFR via the endo-lysosomal sorting pathway is modulated by the expression of Arl4A. Specifically, they observe that depletion of Arl4A accelerates EGFR degradation and overexpression of Arl4A increases the half-life of EGFR following EGF stimulation. Additionally, they find that overexpression of Arl4A somehow inhibits the interaction between the ESCRT-III subunit CHMP2A and the deubiquitinating enzyme USP8/UBPY, suggesting that overexpressed Arl4A indirectly slows EGFR deubiquitination. However, no mechanistic understanding of this inhibition is explored further, which is a major shortcoming of the study. Instead, the authors turn to another finding that Arl4A binds to the ESCRT-II subunit Vps36 via a conserved motif, and disruption of this association (by mutating the Vps36 domain that binds to Arl4A) enhances the rate of EGFR degradation, mimicking the effect of Arl4A depletion. Based on these data, the authors conclude that the interaction between Arl4A and Vps36 decreases EGFR deubiquitination. Unfortunately, this conclusion represents an overinterpretation/overextrapolation of the data presented, and in its current state, the study appears too preliminary for further consideration.

Response: We thank and agree with the reviewer's comments and concerns. To avoid overinterpretation of the data presented, we have toned down our conclusions throughout the text that the interaction between Arl4A and VPS36 decreases EGFR deubiquitination. In addition, we changed the title of our manuscript to "Endosomal Arl4A attenuates EGFR degradation by binding to the ESCRT II component VPS36". We have modified statements on lines 198-216 and 318-335 in the results and on lines 343-345 and 411-423 in the discussion. We also appreciate the comments and have addressed each of the specific points as follows:

Major comments:

1. Although defects in ESCRT-mediated MVB formation lead to the accumulation of ubiquitinated substrates, it is unclear why at a mechanistic level a delay in EGFR deubiquitination specifically would slow its internalization into MVBs. There are many examples of ubiquitinated substrates still being internalized into MVBs.

Response: We thank the reviewer for the comment. Despite that many proteins could be internalized into MVB without deubiquitination, previous studies have shown that knockdown of USP8 delayed EGFR degradation and accumulated ubiquitinated EGFR on the MVBs (Fig. R1 A, B and C)(Ali et al., 2013; Row et al., 2006; Savio et al., 2016). In addition, the Notch receptor also needs to be deubiquitinated by the USP12/UAF1 complex, allowing its sorting

into MVBs for degradation (Moretti et al., 2012). These results suggest that deubiquitination of ubiquitinated proteins is important for cargo sorting into the intraluminal vesicles of MVBs. Indeed, it is not entirely understood why a delay in EGFR deubiquitination specifically slows its internalization into MVBs at a mechanistic level.

This figure was redacted due to third-part rights. It can be found in doi: 10.1074/jbc.M512615200, Figures 1a and 8a.

This figure was redacted due to third-part rights. It can be found in doi: 10.1016/j.cub.2013.02.033, Figures S6.

This figure was redacted due to third-part rights. It can be found in doi: 10.1016/j.cub.2015.11.050, Figure 3.

Fig. R1. USP8 mediates EGFR downregulation. (A) Left panel shows that depletion of USP8 inhibits EGFR-dependent EGFR downregulation. The right panel shows that depletion of USP8 promotes the accumulation of ubiquitin on aberrantly enlarged endosomes. (B) Left panel shows that depletion of USP8 inhibits EGF-dependent EGFR downregulation. The right panel shows that depletion of USP8 leads to the accumulation of ubiquitinated proteins on endosomes. (C) Depletion of USP9X and USP8 alters EGFR fate during the endocytic route.

References:

- Henne WM, Buchkovich NJ, Emr SD. The ESCRT pathway. *Dev Cell*. 2011;21(1):77-91.
- Raiborg C, Stenmark H. The ESCRT machinery in endosomal sorting of ubiquitylated membrane proteins. *Nature*. 2009;458(7237):445-452.
- Row PE, Prior IA, McCullough J, Clague MJ, Urbé S. The ubiquitin isopeptidase UBPY regulates endosomal ubiquitin dynamics and is essential for receptor down-regulation. *J Biol Chem*. 2006;281(18):12618-12624.
- Ali N, Zhang L, Taylor S, Mironov A, Urbé S, Woodman P. Recruitment of UBPY and ESCRT exchange drive HD-PTP-dependent sorting of EGFR to the MVB. *Curr Biol*. 2013;23(6):453-461
- Moretti J, Chastagner P, Liang CC, Cohn MA, Israël A, Brou C. The ubiquitin-specific protease 12 (USP12) is a negative regulator of notch signaling acting on notch receptor trafficking toward degradation. *J Biol Chem*. 2012;287(35):29429-29441.
- Savio MG, Wollscheid N, Cavallaro E, Algisi V, Di Fiore PP, Sigismund S, Maspero E, Polo S. USP9X Controls EGFR Fate by Deubiquitinating the Endocytic Adaptor Eps15. *Curr Biol*. 2016 Jan 25;26(2):173-183.

2. Is the delay in EGFR degradation observed in Arl4A depleted cells similar to that of USP8/UBPY depletion? Can overexpression of USP8/UBPY suppress the delay in EGFR degradation observed in Arl4A depleted cells?

Response: In order to address whether UPS8 and Arl4A function at the same pathway, we performed additional experiments and the results are shown in the **new Supplementary Fig. S14**. We first observed that USP8 knockdown delayed EGFR degradation in HeLa and C33-A cells upon EGF treatment (**new Fig. S14a-b**). Similar to this result, we found that knockdown of USP8 suppresses the acceleration of EGFR degradation in Arl4A-depleted cells (**new Fig. S14c**).—This result indicated that Arl4A and USP8 are both involved in the same biological pathway.

3. *It is unclear how overexpression of Arl4A inhibits the interaction between CHMP2A and USP8/UBPY. This observation needs to be better defined at a mechanistic level to link Arl4A to deubiquitination of EGFR.*

Response: We thank the reviewer for the comment. To confirm the effect of Arl4A on the interaction between CHMP2A and USP8/UBPY, we further showed that depletion of Arl4A increased the interaction between CHMP2A and USP8/UBPY (new Supplementary Fig. 13). In addition, we demonstrated that, compared to VPS36 WT, the VPS36-A6 mutant exhibited reduced coprecipitation of CHMA2A (new Fig. 8b), mimicking that loss of Arl4A interferes the VPS36-CHMP2A interaction. Currently, we hypothesize that Arl4A-mediated stronger interaction between VPS36 and CHMP2A interferes the CHMP2A affinity toward USP8/UBPY. However, it is also possible that Arl4A affects an unknown molecule that can mediate the CHMP2A and USP8/UBPY interaction. We have added this statement in the discussion section (Page 13, line 411).

We also modified the proposed model in Fig. 9g and stated that “The endosomal Arl4, through the interaction of ESCRT-II component VPS36, mediates EGFR ubiquitination levels, sorting to MVBs, and lysosomal degradation. Arl4A-depletion enhanced the endocytosed EGFR transport to late endosomes/lysosomes and accelerated EGF-stimulated EGFR degradation. Specifically, Arl4A-VPS36 interaction facilitates the association of VPS36 with MVBs and sustains the association of VPS36 and ESCRT-III component CHMP2A, resulting in a reduction of CHMP2A-deubiquitinating enzyme USP8 interaction, which could further affect EGFR ubiquitination level.”

4. *How does the interaction between Arl4A and Vps36 relate to the effect of Arl4A overexpression on the interaction between CHMP2A and USP8/UBPY. Without a better understanding of this interplay, the study fails to provide a mechanistic understanding of the role that Arl4A plays in directly regulating EGFR degradation.*

Response: We thank the reviewer for the comment and concerns. In response to the reviewer’s concerns, we have toned down our conclusions, modified the proposed model in the revised Fig. 9g and stated that “Arl4A binding to the ESCRT-II component VPS36 can mediate EGFR ubiquitination and lysosomal degradation.” In addition, as suggested by another reviewer, we have changed the title of our manuscript to "Endosomal Arl4A attenuates EGFR degradation by binding to the ESCRT II component VPS36".

The focus of our study is to reveal that Arl4A is a novel regulator of EGFR degradation and sorting to the lysosome. We provided evidence that Arl4A associated with the ESCRT-II component VPS36, which in turn allows regulation of EGFR degradation and ubiquitylation.

As such, we do not make conclusions about whether and how the Arl4A-VPS36 interaction directly contributes to the effect of Arl4A on the CHMP2A-USP8/UBPY interaction. We agree that it would be interesting to extend such analyses in the future and do indeed plan to further characterize the regulation of EGFR degradation and signaling in this context in future work.

Reply to Reviewer #3

Reviewer #3 (Remarks to the Author):

EGFR is a potent regulator of cell proliferation and survival. Upon binding ligand, the magnitude and duration of EGFR is regulated by several factors, including signal termination resulting from receptor internalization and degradation in lysosomes. While some aspects of EGFR signaling regulation have been studied, the factors that regulate EGFR degradation remain incompletely understood. This study examines the role of the GTPase Arf4A in the regulation of EGFR degradation.

Arf4 silencing enhanced the internalization of fluorescent EGF, used to label ligand-bound EGFR, to early endosomes and lysosomes and also decreased the expression levels of EGFR and other signaling receptors such as Met. The loss of EGFR expression upon Arl4A silencing could be rescued by inhibitors of lysosomal acidification by not of the proteasome. The loss of EGFR expression upon Arl4A silencing was observed in cell lines that express wild type as well as cancer associated EGFR mutants. Arl4A overexpression increased and Arl4A silencing decreased the duration of EGFR ubiquitinylation following stimulation with EGF. Arl4A overexpression decreased the association of the de-ubiquitinylation enzyme USP8 with the ESCRT component CHMP2A. Consistent with a role for Arl4A in regulating EGFR ubiquitinylation and ESCRT-mediated sorting of EGFR, silencing Arl4A impaired the sorting of EGFR into the lumen of enlarged endosomes formed by expression of mutant Rab5.

Arl4A associated with the ESCRT-II component Vsp36 in a GTPase-independent manner, as determined by co-IP and some immunofluorescence co-localization analysis. EGF stimulation increased Arl4A association with GTP as determined by PAK-pulldown and also increased Arl4A association with LAMP1 (determined by colocalization analysis). Scanning mutagenesis of Vps36 coupled to yeast 2 hybrid revealed regions essential for interaction with Arl4A, and a Vps36 mutant of this region exhibited reduced co-precipitation of Arl4A and Vps36. While silencing of Vps36 resulted in delayed EGFR degradation, the rescue of this effect was more pronounced upon expression of Arl4A defective in Vps36 interaction than wild-type Arl4A.

The study concludes and presents a model that Arl4A is a novel regulator of EGFR degradation

and sorting to the lysosome, due to the association of Arl4A with ESCRT-II, which in turn allows regulation of EGFR ubiquitylation. In general, the study is well-designed and the experiments are carefully performed. The rescue of silencing experiments (e.g. Arl4A) with both wild-type and mutant constructs adds significantly to strengthen the conclusions and provide mechanistic understanding of the role of Arl4A. The role of Arl4A in regulation of EGFR degradation is novel and will be of interest to a broad readership. For the most part, the conclusions are well-supported by the data, with the exception of the conclusions pertaining to a specific role of Arl4A in controlling EGFR deubiquitylation, which are too strong for the data shown. Some comments should be addressed prior to publication.

Response: We thank the reviewer for the positive comments and highlighting the interest and novelty of this work. We also thank the reviewer for the comments and opportunity to improve the manuscript. As suggested by the reviewer, we have toned down our conclusions throughout the text that Arl4A has a specific role in controlling EGFR deubiquitylation. Specifically, we also changed the title of our manuscript to "Endosomal Arl4A attenuates EGFR degradation by binding to the ESCRT II component VPS36". We have addressed the comments for the author and in the major and minor comment sections.

Major comments:

1. The overexpression and silencing of Arl4A to monitor EGFR ubiquitylation are each done in different cell lines (Figure 3A-B). This is understandable since HeLa cells appear to not express Arl4A and are thus suitable to study the effect of (over) expression of Arl4A, while C33A cells express Arl4A and are thus suitable to study the effect of Arl4A silencing. It might be useful to show a comparison of the relative levels of Arl4A in HeLa cells upon overexpression relative to that of C33A cells.

Response: We thank the reviewer for the suggestion. We have shown that the amount of exogenous expression of Arl4A in HeLa cells in this study is approximately three times higher than that of endogenous Arl4A protein in C33-A cells (Fig. R1, for reviewer only).

Fig. R1. The relative levels of Arl4A in HeLa cells upon overexpression and in C33-A cells. HeLa cells were transfected with 1 μ g Arl4A plasmid/well in 6-well plates for 24 hours. Cell

lysates (~20 µg) were analyzed by immunoblotting with anti-Arl4A and anti- α -tubulin antibodies.

2. For many of the experiments (Fig. 3E, 4D, 5A, 5C), it is not clear what the individual data points refer to. Is this data from individual cells from a single experiment? Statistical analyses should only be applied to measurements from independent experiments, not on measurements obtained from individual cells from a single experiment.

Response: We thank the reviewer for the suggestion. All the individual data points were from at least three biological repeats and are specified in the figure legend. We also changed the quantitative panel into super plots as described by Lord et al. (2020) to visually represent and compare large amounts of data from multiple experiments.

Reference

Lord SJ, Velle KB, Mullins RD, Fritz-Laylin LK. SuperPlots: Communicating reproducibility and variability in cell biology. *J Cell Biol.* 2020;219(6):e202001064.

3. The blots shown in Figure 3D are not of great quality and should be replaced.

Response: As suggested by the reviewer, we have replaced Figure 3d with a good quality image in the revised Figure 8d.

4. In Figure 5D, could the authors comment on why 15 min was selected? It seems from Figure 5C that Arl4A and LAMP1 may exhibit higher overlap at later timepoints of EGF stimulation.

Response: We thank the reviewer for this comment. Our study showed that VPS36 preferentially interacts with the GDP-bound form of Arl4A and that Arl4A activity decreased after EGF treatment for 15 minutes (Fig. 5b). Furthermore, as shown in Fig. R2 below, we observed that the interaction of Arl4A and VPS36 did not significantly change over time after treatment with EGF for longer than 15 min. It appears that 15 min is an optimal time point to analyze the interaction between VPS36 and Arl4A. Thus, we selected 15 min for the analysis in Fig. 5c.

Fig. R2. EGF enhanced the interaction between VPS36 and Arl4A. Lysates of HeLa cells transfected with the indicated plasmids were immunoprecipitated with Myc-trap beads, and the amounts of bound proteins were analyzed by western blotting with anti-Myc and anti-Arl4A antibodies. The amount of coprecipitated Arl4A (right panel) was quantified by normalizing with Arl4A inputs and pulled-down Myc-VPS36. All the quantified values were then compared with the value at 0 min of EGF treatment. The results represent the mean \pm SD of three independent experiments (**, $P < 0.01$, using one-way ANOVA).

5. Some of the immunofluorescence experiments showing colocalization do not show evident overlap of the Arl4A and the other marker(s) examined. For example, Figure 4D (Arl4A WT), Figure 5A (Arl4A WT), Figure 5C, etc. It would be helpful to clarify how the Pearson's coefficient analysis was conducted (how ROIs were selected for analysis, etc). Showing some control analyses (for example, on pairs of images in which one of the channels is randomized by rotation 180 degrees or similar) would be helpful to delineate what is indeed colocalization. Response: We thank the reviewer for the comment. To confirm the imaging results, we repeated Fig. 4d experiment three times and added the Arl4A G2A (a myristoylation-deficient mutant) and VPS36 coexpression group as a negative control for immunofluorescence images in the new Fig. 4d. The Arl4A signal served as the mask for the (region of interest) ROI per cell. VPS36-Arl4A colocalized pixels/total VPS36 pixels within the mask were calculated to obtain the colocalization coefficients in the images. As a result, the colocalization coefficients in the paper represent the fractions of a protein that localizes on a referenced protein, not the same as the Pearson's coefficient. Overall, we found that except for T51N (the GDP-binding mutant) of Arl4A, Arl4A WT, Q79L, and G2A did not show clear colocalization preference toward VPS36.

For Fig. 5a and c, super resolution-based images were applied to improve image resolution and to replace the original ones in the new Fig. 5a. We used Imaris (Bitplane) software to quantify the 3D colocalization of Arl4A and Lamp1 in super-resolution images. The Arl4A signals

served as the mask for defining the ROI per cell. Using these masks, the 3D-colocalization coefficient values were calculated by Lamp1 pixels that colocalized with Arl4A/total Lamp1 pixels in these masks. The results showed that Arl4A gradually colocalized with the late endosome marker Lamp1, where GDP-bound Arl4A (T51N) is prone to localize, upon EGF treatment for 15 min. This was similar to the results of previous Fig. 5c. For control analysis, we rotated the Arl4A T51N image by 180 degrees and then used Imaris software to calculate its colocalization coefficient (Fig. R3B, for reviewer only). Our results showed that when the image was rotated, the colocalization coefficient between Lamp1 and Arl4A T51N decreased significantly. This control confirmed that the colocalization coefficients analyzed by the software reflect the colocalization of proteins rather than the overlapping of background signals.

Fig. R3. (A) For example: The colocalization coefficient value of VPS36 and Arl4A. Channel 1 represents the VPS36 signal, and Channel 2 represents the Arl4A signal. The VPS36 colocalization coefficient value is calculated by summing the pixels in the colocalized region (Quadrant 3) and then dividing by the sum of pixels in Channel 1 (Quadrant 1 + Quadrant 3). (B) The colocalization coefficient comparison of raw data and 180° misplaced data by rotating the signal from the Arl4A channel.

Also, some of the colocalization of Vps36 and Arl4A appear to have a Pearson's coefficient of nearly 1.0 (Figure 4D) in the Arl4A T51N condition – this would nearly perfect correlation of fluorescence intensities of these two channels, and it is difficult to imagine how this could be the case. These colocalization experiments should be carefully reviewed and revised.

Response: We thank the reviewer for this comment. As mentioned in the previous reply, we did not present the colocalization of VPS36 and Arl4A in Pearson's coefficient; thus, their

subcellular localizations were actually not of perfect positive correlation. In the image analysis, we used Zeiss software to determine the colocalization coefficient of VPS36 (Channel 1). The VPS36 colocalization coefficient value is calculated by summing the pixels in the colocalized region (Quadrant 3) and then dividing by the sum of pixels in Channel 1 (Quadrant 1 + Quadrant 3) (Fig. R4, for reviewer only). The colocalization coefficient values will range from 0 to 1. If the C1 value is “1”, it means that 100% VPS36 colocalized with Arl4A. If the value of C1 is 0, there is no colocalization between VPS36 and Arl4A.

Fig. R4. The colocalization coefficient value of VPS36 and Arl4A. Channel 1 represents the VPS36 signal, and Channel 2 represents the Arl4A signal. The VPS36 colocalization coefficient value is calculated by summing the pixels in the colocalized region (Quadrant 3) and then dividing by the sum of pixels in Channel 1 (Quadrant 1 + Quadrant 3). The Arl4A colocalization coefficient value is calculated by summing the pixels in the colocalized region (Quadrant 3) and then dividing by the sum of pixels in Channel 2 (Quadrant 2 + Quadrant 3).

6. Are there any differences in expression of (myc)-tagged Arl4A mutants versus wild-type?

Response: In this paper, we did not use any Myc-tagged Arl4A, so we did not compare the expression differences between myc-Arl4A and wild-type Arl4A. We infer that the reviewer might question the expression difference between Myc-tagged VPS36 and wild-type VPS36 (untagged). We did not use untagged VPS36 to perform any experiments. We have tried different VPS36 antibodies (Abcam #ab101378; Sigma #HPA039734; Invitrogen #PA5-60561), but none of them were suitable for immunofluorescence staining. In addition, no VPS36 commercialized Abs were found for IF application in previous studies. Therefore, we cannot use immunofluorescence staining to observe the differences between wild-type VPS36 (untagged) and Myc-tagged VPS36.

7. The C33A cell line appears to have very little EGFR (Figure S2A). Are these cells EGFR dependent under the conditions examined, for cell survival or proliferation? Some of these experiments were conducted under conditions in which EGF is added at 100 ng/mL, which should lead to rather rapid downregulation of EGFR near the start of the observation period. Is the effect seen under these conditions (BrdU, or apoptosis) sensitive to EGFR inhibition by TKIs? This would be relevant to allow interpretation of the results shown in Figure S7. It is difficult to demonstrate that the effects of Arl4A siRNA on cell viability are due to the regulation of EGFR by Arl4A, but the dependence of cell viability on EGFR under the conditions examined should at least be included.

Response: Thank you for your kind reminders. In this paper, we found that Arl4A prolongs the stability of growth receptor proteins (such as EGFR and c-Met), which could be the cause of decreased cell proliferation and increased cell apoptosis in Arl4A-depleted cells.

To validate these effects in EGFR low-expressing C33-A cells, we confirmed that gefitinib (a tyrosine kinase inhibitor) reduces C33-A cell growth (Fig. R5 A). These data suggested that the EGFR signaling pathway is important for C33-A cell growth. In addition, gefitinib suppressed cell growth exacerbated after Arl4A siRNA treatment and in C33-A cells (Fig. R5 B), indicating that depletion of Arl4A may not only inhibit EGFR-dependent cell growth but also hinder cell growth through other receptor-dependent signaling pathways.

Our data showed that Arl4A knockdown decreased the protein levels of growth factor receptors, such as EGFR and c-Met. Inhibition of c-Met oncogene signaling induces apoptosis in a variety of cancers. (Campos-Viguri et al., 2020; Hung et al., 2015; Wang et al., 2020; Zagouri et al., 2013). Previous studies also showed that reducing EGFR expression increases apoptosis (Okamoto et al., 2010; Yao et al., 2020). Therefore, we conclude that increasing apoptosis in Arl4A knockdown cells depends at least on EGFR but not specifically.

Fig. R5. Effect of Arl4 on cell proliferation. (A) C33-A cells treated with gefitinib for 24 hours. (B) C33-A cells were transfected with either the control or Arl4A siRNA. After 48 h of

transfection, cells were further treated with gfitinib (1 μ M) for 24 hours. (A-B) The WST-1 assay was performed to estimate cell proliferative activity. The results represent the mean \pm SD of three independent experiments (*, $P < 0.05$; ***, $P < 0.001$; ****, $P < 0.0001$ using one-way ANOVA).

References:

Campos-Viguri GE, Peralta-Zaragoza O, Jiménez-Wences H, et al. MiR-23b-3p reduces the proliferation, migration and invasion of cervical cancer cell lines via the reduction of c-Met expression. *Sci Rep.* 2020;10(1):3256.

Wang Y, Zhang H, Zhang Y, Li X, Hu X, Wang X. Decorin promotes apoptosis and autophagy via suppressing c-Met in HTR-8 trophoblasts. *Reproduction.* 2020;159(6):669-677.

Zagouri F, Bago-Horvath Z, Rössler F, et al. High MET expression is an adverse prognostic factor in patients with triple-negative breast cancer. *Br J Cancer.* 2013;108(5):1100-1105.

Okamoto K, Okamoto I, Okamoto W, et al. Role of survivin in EGFR inhibitor-induced apoptosis in non-small cell lung cancers positive for EGFR mutations. *Cancer Res.* 2010;70(24):10402-10410.

Yao N, Wang CR, Liu MQ, et al. Discovery of a novel EGFR ligand DPBA that degrades EGFR and suppresses EGFR-positive NSCLC growth. *Signal Transduct Target Ther.* 2020;5(1):214.

8. It is not clear how the enhanced arrival of EGFR to EEA1 endosomes (Figure S1A) at upon Arl4A silencing is related to a role for Arl4A in ESCRT sorting or EGFR deubiquitylation. Could the authors comment on this?

Response: We thank the reviewer for this comment. Our results showed that upon depletion of Arl4A, the speed of EGFR plasma membrane transport to early endosomes was accelerated, and the speed of early endosomes to late endosomes was also increased, thus accelerating the degradation of EGFR. We observed an interesting phenomenon in which the GDP-bound form of Arl4A localized to late endosomes and interacted with the ESCRT-II component VPS36. Therefore, in this study, we only focused on the role of Arl4A in the degradation of EGFR on late endosomes.

Our previous results indicated that Arl4A recruits ARNO to the plasma membrane to regulate Arf6 activity (Hofmann et al., 2007; Li et al., 2007). Arl4A also regulates the activity of CDC42 through Robo1 (Chiang et al., 2019). These proteins are all related to endocytosis (Francis et al., 2023; Hu et al., 2011; Izumi et al., 2004). Therefore, the regulation of these proteins may

be affected after knockdown of Arl4A, thus accelerating the internalization of EGFR. The mechanism of the Arl4A effect on EGFR internalization is still not clear. Therefore, we will further investigate the Arl4A-interacting protein, which is involved in the endocytosis process, in the future.

References:

Hofmann I, Thompson A, Sanderson CM, Munro S. The Arl4 family of small G proteins can recruit the cytohesin Arf6 exchange factors to the plasma membrane. *Curr Biol*.

2007;17(8):711-716.

Li CC, Chiang TC, Wu TS, Pacheco-Rodriguez G, Moss J, Lee FJ. ARL4D recruits cytohesin-2/ARNO to modulate actin remodeling. *Mol Biol Cell*. 2007;18(11):4420-4437.

Chiang, T.S., M.C. Lin, M.C. Tsai, C.H. Chen, L.T. Jang, and F.S. Lee. 2019. ADP ribosylation factor-like 4A interacts with Robo1 to promote cell migration by regulating Cdc42 activation. *Mol Biol Cell*. 30:69-81.

Francis, C.R., M.L. Bell, M.M. Skripnichuk, and E.J. Kushner. 2023. Arf6 Regulates Endocytosis and Angiogenesis by Promoting Filamentous Actin Assembly. *bioRxiv*.

Izumi G, Sakisaka T, Baba T, Tanaka S, Morimoto K, Takai Y. Endocytosis of E-cadherin regulated by Rac and Cdc42 small G proteins through IQGAP1 and actin filaments. *J Cell Biol*. 2004;166(2):237-248.

Hu J, Mukhopadhyay A, Craig AW. Transducer of Cdc42-dependent actin assembly promotes epidermal growth factor-induced cell motility and invasiveness. *J Biol Chem*. 2011;286(3):2261-2272.

9. In figure S2A, there appears to be very little Arl4A expressed in HeLa, A549, H1975 and PE089 cells. Nonetheless, there are effects of Arl4A knockdown for each of these cell lines shown in Figure 2. The detection of mRNA levels of Arl4A upon knockdown are helpful to appreciate that knockdown took place. The lack of evident detection of Arl4A protein in these cells makes some of the interpretation of the data less straightforward.

Response: We thank the reviewer for this suggestion. We had difficulty in detecting endogenous Arl4A in HeLa, A549, H1975, and PE089 cells, as shown in Fig. 2, when we harvested those cells and applied western blotting with 20-30 µg total protein/lane with our current antiserum. This result is reasonable because only cervical cancer C33-A cells were identified as the cancer cell line that exhibits the highest and detectable level of Arl4A protein among the 21 human cancer cell lines we examined previously (Chen et al., 2020) (Fig. R6A).

In response to this comment, we have increased the cell lysate amounts up to 120 µg total

protein/lane for western blotting to confirm the siRNA knockdown efficiency under the same conditions used in this study. Endogenous Arl4A was finally detected by western blotting, although with low signal-to-noise ratios (Fig. R6B). Western blotting analysis for HeLa, COS-7, A549, H1975, and PE089 cells has been added to the manuscript in new Supplementary Figs. 1, 9, 10, and 11.

Fig. R6. (A) Detection of endogenous Arl4A among the 21 human cancer cell lines examined. (B) Confirmation of Arl4A protein knockdown efficiency in HeLa, A549, H1975, and PE089 cells by western blotting.

References:

Chen KJ, Chiang TC, Yu CJ, Lee FS. Cooperative recruitment of Arl4A and Pak1 to the plasma membrane contributes to sustained Pak1 activation for cell migration. *J Cell Sci.* 2020;133(3):jcs233361.

10. The statement in the discussion on line 370: “The present study revealed that Arl4A acts as a negative regulator of EGFR” is confusing. For example, some of the data is summarized by the statement on line 357-358 “...our results show that lack of Arl4A results in strongly reduced levels of EGFR...” – this indicates that Arl4A is a positive regulator, at least of EGFR expression levels. Most of the experiments in this paper demonstrate that Arl4A enhances EGFR expression, consistent with enhanced EGFR function.

Response: We thank the reviewer for pointing out this error. We have corrected this discussion on original line 370 to “The present study revealed that Arl4A acts as a positive regulator of EGFR” on line 364 of the revised manuscript.

11. The statement on line 188 that “Arl4A depletion may reactivate EGFR mutant degradation...” could be revised. It is not clear that EGFR degradation is a binary switch that can only be in the on/off state and it is not clear that Arl4A silencing counteracts the specific perturbation of EGFR degradation seen in mutant receptors. More accurate would be to just

state that "EGFR degradation is enhanced by Arl4A depletion even in cells with EGFR mutations that have intrinsically reduced EGFR degradation rates."

Response: We thank the reviewer for rephrasing the statement for us. We have corrected this statement on line 185 in the revised manuscript.

12. In Figure S8A, what is the time of EGF stimulation? This is important to allow understanding of how this relates to the distinct effects of Arl4A silencing on EEA1 overlap of EGFR following 5 vs 15 min shown in Figure S1A.

Response: We thank the reviewer for this question. In Fig. S8A, cells were grown and fixed under conditions supplemented with normal growth medium: DMEM with 10% FBS, in which the EGF concentration is only approximately 15 ng/mL and does not trigger the EGFR degradation pathway. We therefore thought that this is why the colocalization coefficient of EGFR and EEA/Lamp1 was very low. To specifically correlate with EGF (100 ng/mL)-induced effects in the original Fig. S1A, costaining of EGFR and those markers upon EGF treatment at different time points has been added for A549, H1975, and PE089 cells in the **new Supplementary Figs. 9, 10 and 11** of the revised manuscript. The calculated colocalization coefficients of three lung cancer cells, similar to the trend observed in HeLa cells (Fig. S1A), supported that EGFR sorting accelerates upon EGF treatment in Arl4A-depleted cells.

13. A description of the quantification for the localization of EGFR to the lumen of enlarged endosomes shown in Figure 3E should be added beyond the brief statement in the figure legend for Figure 3E. For instance, was the lumen of the enlarged endosomes identified manually? What is the size distribution of these ROIs selected to represent the endosome lumen in each condition studied?

Response: We thank the reviewer for these questions. In Fig. 3e, endosomes with a size measured by ImageJ of no less than 2 μm were selected for quantification, and the ROIs of EGFR signals inside the enlarged MVB were manually referenced from Rab5 Q79L signals. The signal intensities were measured using the ImageJ "Measurement" function under the "Analyze" list. To improve image resolution, we replaced the data in Fig. 3E with images from super-resolution microscopy. The quantification methods of EGFR intensities inside the lumens of Rab5 Q79L-enlarged endosomes were the same as mentioned above. A brief description of the analysis methods is included in the figure legend of the **new Fig. 3c**.

14. It is perhaps more accurate (lines 331-333) to say of the experiments in Figure 7 that Arl4A binding to Vps36 regulates EGFR ubiquitinylation levels, rather than extending this conclusion

to state that this is specifically due to EGFR deubiquitylation modulation. A similar tempering of conclusions in the Results and Discussion section that indicate that effect of Arl4A is to regulate EGFR deubiquitylation is suggested throughout the manuscripts. This is due to the evidence to support this being largely limited to a displacement of USP8 from co-IP with CHMP2A with Arl4A overexpression (Figure 3C), which could be impacted by overexpression of these proteins and the effect of silencing of Arl4A on EGFR ubiquitylation duration, which could be affected by either ubiquitylation or de-ubiquitylation.

Response: We thank and agree with the reviewer's comments and concerns. We have corrected the overinterpretation of the effect of Arl4A on EGFR ubiquitination and changed the title of our manuscript to "Endosomal Arl4A attenuates EGFR degradation by binding to the ESCRT II component VPS36".

We have modified statements on lines 198-207 and 318-335 in the results and on lines 343-345 and 411-423 in the discussion.

Minor:

1. The sentence in lines 283-284 is awkward and difficult to understand: "Using an *in vitro* GST pulldown assay, we also found that GST-VPS36-WT-(330-386) interacted with Arl4A but not with the A3, A4, or A6 VPS36 mutants (Fig. 6B)"

Response: We thank the reviewer for pointing out this error. We have corrected this statement on lines 257-259 to "Using an *in vitro* GST pulldown assay, we confirmed that GST-VPS36-WT-(330-386) interacted with Arl4A but not with the A3, A4, or A6 VPS36 mutants (Fig. 6b)."

Reply to Reviewer #4

Reviewer #4 (Remarks to the Author):

In the manuscript "Endosomal Arl4A attenuates EGFR degradation by delaying its deubiquitination at the late endosome", Lin et al. report that the small GTPase Arl4A reduces EGFR degradation by binding to the ESCRT-II component VPS36. They further describe that Arl4A delays EGFR degradation by decreasing the deubiquitination of the receptor at late endosomes.

The findings reported in the paper are novel and of significance to the field; nevertheless, many of the data in the current form are not convincing and not able to support the authors' claims and conclusions. There are fundamental issues with the data presentation and interpretation. Specific points are listed below:

Response: We thank the reviewer for the positive comments and opportunity to improve the manuscript. We also appreciate the comments aimed at improving the quality of our manuscript,

and we have addressed these below, both in the comments for the author as well as the major and minor comment sections.

Major comments:

1) One of the main issues concerns all IF results presented in the manuscript. First, it is not clear whether the data presented derive from three independent experiments or from only one. This should be specified. If they originate from a single experiment, this is not sufficient to make any conclusion and the experiments should be repeated at least two more times. If they originate from at least three independently performed experiments, the results from independent experiments should not be pooled for statistical analysis.

Response: We thank the reviewer for the comments. All the dot plots in this paper represent the results from at least 3 biological repeats of the experiments (Figs. 3c, 4d, 5a, 7, and 8a; Supplementary Figs. 2, 9, 10, and 11). We have adjusted our quantified figures into super plots (Lord et al., 2020) to visually present and compare large amounts of data from multiple experiments with respective n numbers. We also clarified the biological repeats and the n numbers calculated in the plots in all the figure legends.

Reference

Lord, S.J., K.B. Velle, R.D. Mullins, and L.K. Fritz-Laylin. 2020. SuperPlots: Communicating reproducibility and variability in cell biology. *J Cell Biol.* 219.

Second, the image resolution is very low. For example, it is not possible to see the red signal on the merged, not magnified, images in suppl. Fig.8.

Response: We thank the reviewer for this comment. We agree with the reviewer that the way we present the endosomes (red signals) in large fields was not vision-friendly. To solve this issue, we first increased the image resolution from 516x516 pixels per field to 1024x1024 pixels per field. Meanwhile, super-resolution microscopy was used in some experiments to observe the tiny details inside the cell. The original images in Figs. 3e, 5a, and 5c have been replaced by new Figs. 3c, 5a, and 7c. Second, we improved our presentation by displaying enlarged views in the layouts to illustrate the localization of EGFR and the endosomes so that the red signal can clearly be observed. We noted that we expanded the experiments in Supplementary Fig. 8 to new Supplementary Figs. 9, 10 and 11 for different treatment time points in response to the reviewers' questions and to better illustrate the effect of Arl4A deficiency on EGFR endosomal dynamics in the context of EGF signaling.

Nevertheless, even when it is clearly visible that there is no (or poor) colocalization, the conclusion is often misleading. For example, the statement in lines 197-199: “Colocalization between EGFR and the late endosomal/lysosomal marker Lamp1 was barely observed in the control cells, suggesting that EGFR remains in early endosomal compartments (Supplementary Fig. 8A-B).” is not supported by data. Indeed, in Supplementary Fig. 8, the colocalization measured between EGFR and early endosomes is equally low as the colocalization between EGFR and late endosomes for control cells.

In the same figure, in PE089 cells, it is visible that EGFR colocalizes similarly with EEA1 in control and siArl4A cells, even though the EGFR signal is barely visible in siArl4A cells. It is not clear why the EGFR signal is barely visible in siArl4A cells stained for EEA1, but much stronger in the same cells stained for Lamp1 (but the Lamp staining is poorly visible). Based on these images, low Pearson's correlation coefficient, and lack of information about the number of experiments/statistics, it is impossible to draw any conclusion, including what is written in line 997, that “Depletion of Arl4A promotes EGFR mutant trafficking to lysosomes”. It might help to repeat these experiments by stimulating the cells with EGF as done in suppl. Fig.1.

Response: We thank the reviewer for the suggestion. We realized that the low colocalization issue is the main difficulty for readers to combine the data and the stated conclusions. The cells in the original Supplementary Fig 8 were cultured in medium with FBS, which contained low amounts of EGF that could bind and activate EGFR at basal levels. However, low concentrations of EGF do not trigger EGFR to go through the degradation pathway; alternatively, EGFR is recycled to the plasma membrane once it reaches early endosomes (Tanaka et al., 2018). Thus, without the synchronized trigger for EGFR trafficking, the colocalization coefficient of EGFR and the early endosomal or late endosomal marker was very low. To clearly distinguish the Arl4A deficiency-mediated effect on EGFR trafficking, we monitored EGFR localization under EGF treatment (new Supplementary Figs. 9, 10, and 11). We found that the EGFR-endosome colocalization differences became distinguishable in the siControl and siArl4A groups specifically at 5 min for the EEA1 compartment and at 30 min for the Lamp1 compartment in 3 different cell lines, supporting that depletion of Arl4A increases EGFR delivery to late endosomes. When EGFR colocalizes with Lamp1, it also indicates that EGFR is preparing to go through the degradation pathway. The decrease in EGFR levels will also cause decrease in the colocalization coefficient values. Therefore, in these cell lines, lower colocalization coefficient values were observed for EGFR and Lamp1. Several previous studies also showed similar phenomena (Gui et al., 2012; Huttunen et al., 2017). For example, the colocalization coefficients of EGFR and Lamp1 in A549 cells after the addition

of EGF for 1 hr was approximately 0.20 (Fig. R1A). Moreover, lower colocalization coefficients of EGFR and Lamp1 was also observed in various cell lines, such as RBE and MMNK-1 cells (Fig. R1B).

In the original supplementary Fig. 8b, for Lamp1 staining, EGFR intensity is shown to be decreased after Arl4A knockdown. However, in the enlarged image, EGFR intensity is found to be higher than that in siControl. In this case, our purpose was to clearly see the colocalization of EGFR and LAMP1, we therefore selected the region that was brighter for EGFR, thus causing reviewers to mistakenly think that EGFR's fluorescence intensity was higher after Arl4A KD. We apologize that the original Supplementary Fig. 8a-b we presented are not consistent enough, causing the reviewers to misunderstand the results.

This figure was redacted due to third-party rights. It can be found in doi: 10.1186/1471-2407-12-179, Figure 3b.

Fig. R1. (A) The colocalization coefficient of EGFR and Lamp1 in A549 cells after EGF stimulation for 1 hr. (B) The colocalization coefficient of EGFR and Lamp1 in RBE (black columns) and MMNK-1 (white columns) cells after EGF stimulation.

References:

Tanaka T, Zhou Y, Ozawa T, et al. Ligand-activated epidermal growth factor receptor (EGFR) signaling governs endocytic trafficking of unliganded receptor monomers by non-canonical phosphorylation. *J Biol Chem.* 2018;293(7):2288-2301.

Gui A, Kobayashi A, Motoyama H, Kitazawa M, Takeoka M, Miyagawa S. Impaired degradation followed by enhanced recycling of epidermal growth factor receptor caused by hypo-phosphorylation of tyrosine 1045 in RBE cells. *BMC Cancer.* 2012;12:179.

Huttunen M, Turkki P, Mäki A, Paavolainen L, Ruusuvoori P, Marjomäki V. Echovirus 1 internalization negatively regulates epidermal growth factor receptor downregulation. *Cell Microbiol.* 2017;19(3):10.1111/cmi.12671.

2) Another concern is that the results are based on the use of only one siRNA targeting Arl4A (only one experiment in suppl. Fig. 3 and 5 shows that the reintroduction of Arl4A rescues the defect). Arl4A rescue experiments need to be performed more consistently throughout the study (as done for Vps36), or at least a second siRNA targeting Arl4A should be included to confirm the specificity of the siRNAs used in this study.

Response: We thank the reviewer for raising this point. We used another Arl4A-targeting siRNA sequence (siArl4A#2) to confirm that Arl4A deficiency leads to decreased EGFR protein levels. Similar to the effect of Arl4A-targeting siRNA used in the original manuscript (siArl4A#1), siArl4A#2 reduced the expression of EGFR compared to the control in C33-A cells (**Fig. R2**). The siArl4A#2 sequence was validated and published in our previous paper (Chen et al., 2020). We have added these data to the current manuscript (**new Supplementary Fig. 4a**).

Fig. R2. Depletion of Arl4A alters the levels of EGFR protein. C33-A cells were transfected with the indicated siRNAs for 48 hrs.

References:

Chen, K.J., T.C. Chiang, C.J. Yu, and F.S. Lee. 2020. Cooperative recruitment of Arl4A and Pak1 to the plasma membrane contributes to sustained Pak1 activation for cell migration. *J Cell Sci.* 133.

3) *There are three isoforms of Arl4 (Arl4A, Arl4C, and Arl4D). Is the Arl4A siRNA used in this study specific for Arl4A isoform only?*

Response: We thank the reviewer for raising this point. We confirmed that the Arl4A siRNA sequence is specific for Arl4A but not for Arl4C and Arl4D. As the MUSCLE (Multiple Sequence Comparison by Log- Expectation) alignment result shows below, when importing the Arl4A siRNA sequence, only the Arl4A nucleotide sequence could be aligned as the targeting sequence.

siArl4A #1		siArl4A #2	
	GGAACTCATTGTCACT		TGCCTTCATTTCAGTCT
Arl4A	GGAACTCATTGTCACT	Arl4A	TGCCTTCATTTCAGTCT
siArl4A#1	GGAACTCATTGTCACT *****	siArl4A#2	TGCCTTCATTTCAGTCT *****
Arl4C	GGCATCAGCTGCCACT	Arl4C	TCCCTGCATATCGTCAT
siArl4A#1	GGAACTCATTGTCACT ** * ** ****	siArl4A#2	TGCCTTCATTTC----- * *** ** *
Arl4D	ATGGGGAACCACTTGACT	Arl4D	CGCCTCAAGTTCAAGGA
siArl4A#1	----GGAAC----- *****	siArl4A#2	TGCCTTCATTTC----- **** * ****

Fig. R3. The Muscle (Multiple Sequence Alignment) alignment results show that the siArl4A siRNA sequence does not match with Arl4C or Arl4D.

4) In many experiments, the WBs including control siRNA and Arl4A siRNA (or control and Arl4A overexpression) are shown on separate panels (e.g. Fig. 1f-g, Fig.3, suppl. Fig. 6A, etc.). For a direct comparison of the levels of Arl4A in the different conditions, these samples should be presented on the same membrane and not on separate panels.

Response: We thank the reviewer for raising this point. Arl4A depletion was found to lower EGFR protein levels. When comparing EGFR degradation in siControl and siArl4A cells, we therefore loaded more total cell lysate in siArl4A groups to make the antibody's ability to recognize EGFR the same at the starting point (0 minute), and EGFR levels were therefore detected on different membranes and separate panels. We therefore presented Arl4A signals in a consistent way. However, we did confirm the Arl4A knockdown efficiency by loading proteins equally in siControl and siArl4A cells, so we replaced the blots of Arl4A and tubulin in the original manuscript with separate panels. We also specified the loading differences for detecting EGFR in the figure legends (Figs. 1f-g, 3a-b, and Supplementary Fig. 7a).

5) Why the EGFR receptor is often barely degraded in control cells even at late time points (e.g. in Fig.1 f, Fig. 2, etc.)?

Response: Thank you for raising this question. We regarded slow EGFR degradation as a cell type-dependent issue. We observed that HeLa cells in Fig. 1G and A549 and H1975 cells in Fig. 2B have no such problem. In C33-A cells, the slower rate of EGFR degradation could be due to the abundance of Arl4A in contrast to HeLa cells, which possess comparably low amounts of Arl4A (new supplementary Fig. 3a). PE089 in Fig. 3D contains an exon 19 deletion similar to HCC827, which was recently shown to be highly resistant to EGF-induced EGFR downregulation compared to other lung cancer cell lines (Fig. R4A) (Wang et al., 2018). They also revealed that the exon 19 deletion mutant of EGFR is prone to accumulate in the endosome compartment and is somewhat resistant to EGF-induced endocytic degradation compared to wild-type EGFR. We inferred that PE089 may degrade EGFR slower by using the same strategy. In addition, the study also showed that different lung cancer cell types actually degrade EGFR at different speeds in response to EGF, which is a phenomenon commonly observed in other studies (Fig. R4B) (Shen et al., 2021).

This figure was redacted due to third-part rights. It can be found in <https://doi.org/10.1186/s12964-018-0245-y>, Figure 1.

This figure was redacted due to third-part rights. It can be found in <https://doi.org/10.3389/fcell.2021.642625>, Figure 1.

Fig. R4. (A) The cells were exposed to EGF at 20 ng/ml, and downregulation of EGFR was observed in all cell types, with the fastest degradation occurring in A549 cells that bear wild-type EGFR. The exon 19 deletion mutant of EGFR in HCC827 cells was very resistant to EGF-induced downregulation. Right panel; the exon 19-deleted EGFR in HCC827 cells is very insensitive to higher dosages of EGF stimulation (100 ng/ml). (B) The degradation rate of EGFR after EGF stimulation is not the same for A549, H3255, and HeLa cells.

References:

Liu, X. Liu, S. Liu, Y. Zhang, Y. Wang, L. Zou, and H. Liu. 2018. The exon 19-deleted EGFR undergoes ubiquitylation-mediated endocytic degradation via dynamin activity-dependent and -independent mechanisms. *Cell Commun Signal.* 16:40.

Shen CH, Chou CC, Lai TY, et al. ZNRF1 Mediates Epidermal Growth Factor Receptor Ubiquitination to Control Receptor Lysosomal Trafficking and Degradation. *Front Cell Dev Biol.* 2021;9:642625.

And why two different concentrations of EGF (100 or 50 ng/ml) are used to stimulate the cells in different experiments?

Response: According to current studies, 100 ng/ml EGF is commonly used in EGF-induced EGFR degradation. In most of our experiments, we also applied 100 ng/ml EGF. However, in particular experiments, the VPS36 depletion-induced delay in EGFR degradation was not

obvious when 100 ng/ml EGF was used. We inferred that EGFR degradation could be too rapid to observe the phenotypic change caused by VPS36 depletion and therefore referred to a study that reduced the concentration of EGF to 50 ng/mL to lower the sensitivity of EGFR degradation (Malerød et al., 2007; Zeng and Carlin, 2019). Our results showed that VPS36 depletion could slow the degradation of EGFR in the window of 50 ng/ml EGF stimulation when compared to the control group. For this reason, we only adjusted the EGF concentration to 50 ng/ml in the experiments that required the knockdown of VPS36. We added the specification in the revised materials and methods.

References:

Zeng, X., and C.R. Carlin. 2019. Adenovirus early region 3 RID α protein limits NF κ B signaling through stress-activated EGF receptors. PLoS Pathog. 15:e1008017.

Malerød, L., S. Stuffers, A. Brech, and H. Stenmark. 2007. Vps22/EAP30 in ESCRT-II mediates endosomal sorting of growth factor and chemokine receptors destined for lysosomal degradation. Traffic. 8:1617-1629.

6) *It is not clear why in suppl. Fig. 6B, p-EGFR level relative to tubulin graph, the curve relative to siArl4A decreases over time, while in the blot shown in suppl. Fig. 6A there is a clear increase in the levels of p-EGFR at 15 minutes.*

Response: We thank the reviewer for raising this point. Due to the large error bar derived from three biological repeats, the representative amount of p-EGFR measured at the 15-minute time point did not match the statistical graph. We therefore repeated the experiment twice to confirm our results and used similar experimental values after combining with the previous statistical results to avoid misinterpretation from large variations. We found that p-EGFR/tubulin was significantly different between the siControl and siArl4A groups at 30 min but not at 15 min, as shown in the **new Supplementary Fig. 7**. The discrepancy between the previous results and the current results is due to the high background value leading to quantitative errors. The key point is that we confirmed that EGFR phosphorylation (pEGFR/ α -tubulin) was significantly different between the siControl and siArl4A groups after EGF stimulation.

7) *The only image showing that EGFR is present in Arl4A-positive endosomes is Fig.3E. However, in this experiment, the authors also express the constitutively active mutant of Rab5, which is very well known to dramatically affect the endosomal pathway and the sorting of EGFR (e.g. Wegener et al, 2010). GFP-Rab5 wt should be included as control.*

Response: We thank the reviewer for the comment. We agree with the reviewer that Rab5

Q79L affects the endosomal pathway. We also tried to use wild-type Rab5 to observe the transport of EGFR. In wild-type Rab5-expressing cells, we observed EGFR on Rab5-positive vesicles after EGF treatment (Fig. R5). However, Rab5 WT was not able to enlarge MVB lumens, allowing us to quantify the EGFR fluorescence intensity “inside” the endosomes and assess EGFR sorting differences; therefore, the Arl4A overexpression effect could not be compared by using Rab5 WT. For this reason, we only display the experimental data of Rab5 Q79L in the manuscript.

Fig. R5. The HeLa cells were transfected for 24 hours with the Rab5 wild type-mRFP and E.V. or Arl4A plasmid (1 μ g). Subsequently, the cells were incubated with 100 ng/ml EGF for 10 min at 37°C and then stained with anti-EGFR (orange) and anti-Arl4A (green).

In addition, this image is not fully supporting the conclusion stated in lines 233-236 as in the control cells are also visible enlarged endosomes without EGFR in the lumen, and in the siArl4A cells are also visible enlarged endosomes with EGFR in the lumen.

Response: We thank the reviewer for raising this concern. We admit that the image resolution in the original manuscript was not friendly for the readers to judge whether EGFR is around or inside the endosome lumens and therefore could not fully support our conclusion. We therefore reperformed this experiment with super-resolution microscopy in the revised manuscript (new Fig. 3c). We improved the image resolutions and calculated the EGFR signals inside the Rab5 Q79L enlarged endosomes to give clearer and more significant results to support that Arl4A expression enhances EGFR sorting into MVBs.

In addition, the analysis is also performed on only 18 cells, and as for the other IF experiments, it is not specified whether the results are from one experiment or more.

Response: We repeated the experiment in Fig. 3E to increase the sample size, and a total of 30 cells were subjected to analysis. In response to the other reviewers' suggestion, we also repeated the experiment by using super-resolution microscopy to confirm our results (new Fig. 3c). We

observed the same phenomenon in super-resolution microscopy and confocal microscopy, in which Arl4A overexpression prevented EGFR sorting into MVBs.

The increased sample size from super resolution-based images was 30 cells, and the cell numbers were acceptable in this circumstance for consideration of operation time and file size. Many recent studies also used sample sizes between 5 and 20 cells for the super resolution-based quantification analysis, (Carravilla et al., 2021; Chen et al., 2022; Han et al., 2017; Wang et al., 2019; Yang et al., 2020). We conducted the experiment for 3 biological repeats, each consisting of an average of 26 more endosome scans from ~10 cells (E.V.=100 endosomes; Arl4A=81 endosomes). The specification was written in the figure legend.

For other IF experiments, at least 3 biological repeats were conducted and presented in the format of a super plot for readers to clearly understand where the pooled quantitative dots actually come from. The biological repeats and the n numbers calculated per experiment are specified in the figure legends of the revised manuscript.

References:

Han Y, Li M, Qiu F, Zhang M, Zhang YH. Cell-permeable organic fluorescent probes for live-cell long-term super-resolution imaging reveal lysosome-mitochondrion interactions. Nat Commun. 2017 Nov 3;8(1):1307. **Number of cells = 5**

Carravilla P, Dasgupta A, Zhurgenbayeva G, et al. Long-term STED imaging of membrane packing and dynamics by exchangeable polarity-sensitive dyes [published correction appears in Biophys Rep (N Y). 2022 Jun 08;2(2):None]. **Number of cells = 5**

Wang C, Taki M, Sato Y, et al. A photostable fluorescent marker for the superresolution live imaging of the dynamic structure of the mitochondrial cristae. Proc Natl Acad Sci U S A. 2019;116(32):15817-15822. **Number of cells = 20**

Chen Y, Hu S, Wu X, et al. Synaptotagmin-1 is a bidirectional Ca²⁺ sensor for neuronal endocytosis. Proc Natl Acad Sci U S A. 2022;119(20):e2111051119. **Number of cells = 11-15**

Yang X, Yang Z, Wu Z, et al. Mitochondrial dynamics quantitatively revealed by STED nanoscopy with an enhanced squaraine variant probe. Nat Commun. 2020;11(1):3699. **Number of cells = 6**

8) In some experiments, both GTP- and GDP- bound Arl4A interacts with Vps36. However, in the co-IP (and IF), Vps36 seems to preferentially interact with the dominant negative mutant. How do the authors explain this?

Response: We observed that both GTP- and GDP-bound Arl4A interact with VPS36 both in the yeast two-hybrid system and *in vitro* (Fig. 4a and 4b) but not *in vivo* (co-IP and IF data). Since GTP-bound and GDP-bound Arl4 have distinct spatial distributions *in vivo*, we inferred that the membrane for Arl4A targeting is the cause of those discrepancies. N-myristoylation modification of Arl4 is important for consolidating membrane localization, and this modification does not exist in the yeast two-hybrid system and purified components from *E. coli* because Arl4s were tagged with either LexA or His at the N-terminus. Furthermore, the *in vitro* binding system contains no membrane environment. We speculated that the association of Arl4A and distinct membrane components may lead to conformational changes in Arl4A. The changes that accompany the binding of GDP and GTP could contribute an additional layer of specificity of Arl4A for effector proteins. As a result, we observed the physical interactions between VPS36 and Arl4A WT, Q79L, or T51N in the yeast nucleus and *in vitro*, but particular specificity occurred in mammalian cells (co-IP and IF), in which the membrane components were taken into consideration.

In addition, according to the analysis in Fig. 5C, there is an increase in colocalization between Arl4A and Lamp-1 45 and 60 minutes after EGF stimulation, and Fig. 5B shows that Arl4A is more active at these time point. How does this fit with the statement that the dominant negative mutant is mostly associated with the lysosome?

Response: We thank the reviewer for raising this question. We agree that the activity pulldown assay showing increased Arl4A interaction with PAK-PBD at 45 and 60 min is slightly counteracting our model that the Arl4A-GDP bound form is induced and located at the lysosomal compartment at the corresponding time point by IF. We explained that IF colocalization may not directly reflect the nucleotide binding states because it takes time to change the subcellular localization after the conversion of GDP/GTP. Additionally, the Arl4A protein may undergo various conformational changes and require the help of additional proteins or other components to properly localize. Consistently, EGF triggers Arl4A to reduce activity and translocate to late endosomal compartments.

Also, the images in Fig.5A do not support this conclusion, as no Arl4A T51N-positive vesicles are present in the magnified green channel. It is not clear how the authors measure colocalization, as the signal from Arl4A T51N in the magnified region shown is clearly cytosolic and not overlapping with Lamp-1-positive vesicles, even though the artificial-coloured image seems to illustrate that. It seems that this is rather the consequence of

measuring the background/cytosolic signal due to a too low threshold in the image analysis than a real result.

Response: We thank the reviewer for the comments. In the original Fig. 5a, we used Zen software to determine the colocalization coefficient of Arl4A and Lamp1. Briefly, the colocalization coefficient values were calculated as Arl4A-Lamp1 colocalized pixels/total Lamp1 pixels when using the Arl4A-positive area as the mask for the ROI. However, we agree that the image resolution in the original manuscript cannot fully support our conclusion. We improved the image quality by using super-resolution microscopy in the new Fig. 5a to replace the original Fig. 5a and 5c. Imaris (Bitplane) software was applied to quantify the 3D colocalization of Arl4A and Lamp1 in the super-resolution images. Using the “surface” module in the software, we adjusted the threshold to allow “seed points” to mask all the vesicles from the Lamp1 channel. Objects with values below the threshold were excluded. The pixels of masked Lamp1 were further applied to calculate the 3D-colocalization coefficient value, which was defined as the Lamp1-Arl4A colocalization pixel/total Lamp1 channel pixel. We observed that Lamp1-positive vesicles overlapped more with Arl4A T51N than with Arl4A WT. We also rotated the Arl4A T51N image by 180 degrees and then used Imaris software to calculate its colocalization coefficient (Fig. R6). Our results showed that when the image was randomly rotated, the colocalization coefficient between Lamp1 and Arl4A T51N decreased significantly. The results confirmed that Arl4A and Lamp1 partially colocalized, and the image analysis in the new Fig. 5A was not a consequence of the background/cytosolic signal.

Fig. R6. The region of colocalization between Lamp1 and Arl4A is labeled white in the immunofluorescence image according to the colocalization analysis performed using the Imaris 3D colocalization view.

The same for Fig. 5C, where the Person’s correlation coefficient is close to zero for the different conditions, indicating almost no colocalization. As many of the images seem to have a low signal-to-noise ratio, the colocalization values measured may simply be the consequence of a very low threshold level. It may help to reduce the Arl4A cytosolic signal by permeabilizing

the sample before fixation.

Response: We thank the reviewer for the suggestion. We fully agree with the reviewer's opinion that previously presented colocalization coefficients were low and images have low signal-to-noise ratios. As mentioned above, we therefore improved the image quality by using super-resolution microscopy and requantified the colocalization coefficients in the new Fig. 5a. In regard to the low colocalization coefficients, elevating the image threshold from the Arl4A channel and enhancing image resolution did help a lot. Regarding the low signal-to-noise ratio, we followed the suggestion from the reviewer and have tested the possible effects of permeabilizing the samples before fixation (Fig. R7 shown below). However, compared to the original staining method (Fig. R7 A), all the pre-permeabilized conditions we tried drastically decreased Lamp1 signal intensity (Fig. R7 B and C). We also simultaneously fixed and permeabilized the cells with paraformaldehyde and Triton X-100 (Fig. R7 D-J). The treatment reduced the background but was also devastating for both Arl4A and Lamp1 signals in Figure D and E (only when the laser gain was high could Lamp1 be detected). On the other hand, the signal-to-noise ratios did not improve in Figure F-J. Thus, we applied super-resolution microscopy with the same fixation conditions as before (Fig. R7 A). Overall, we found that the results from super-resolution-based imaging were consistent with our previous findings, which supports Arl4A translocation to Lamp1 vesicles upon EGF treatment.

Arl4A /Lamp1/DAPI

Fig. R7. HeLa cells were fixed and permeabilized using different methodologies. The detailed method is described in Table R1.

	First treatment	Second treatment	Lamp1 signal (Red)	Arl4A signal (Green)	Lower background noise
A	4% PFA for 15 min	0.2 % Triton X-100 for 5 min	Good	Good	No
B	0.1% Triton X-100 for 2min	4% PFA for 15 min	Decreased /Dispersed	Punctate/ weak	Yes
C	0.2% Triton X-100 for 2min	4% PFA for 15 min	weak/no	Punctate/ weak	Yes
D	2% PFA with 0.2% Triton X-100 for 10 min	non	Decreased /Dispersed	Punctate/ weak	Yes
E	2% PFA with 0.1% Triton X-100 for 10 min	non	Decreased /Dispersed	Punctate/ weak	Yes
F	2% PFA with 0.5% Triton X-100 for 2 min	4% PFA for 15 min	Decreased /Dispersed	Punctate/ tubular/ cytosol	NO
G	2% PFA with 0.2% Triton X-100 for 2 min	4% PFA for 15 min	Good	The localization is slightly different	NO
H	2% PFA with 0.1% Triton X-100 for 2 min	4% PFA for 15 min		uneven staining	NO
I	4% PFA with 0.5% Triton X-100 for 2 min	4% PFA for 15 min	Good	The localization is slightly different	NO
J	4% PFA with 0.2% Triton X-100 for 2 min	4% PFA for 15 min	Good	The localization is slightly different	NO

* B-E The red signal used higher values of laser and a high gain value

§ B-E The green signal used higher values of laser and a high gain value

Table. R1. HeLa cells were fixed and permeabilized using different methodologies. (A) Previously results using this protocol to fix and permeabilize cells. (B-C) Permeabilization of

the sample before fixation. (D-E) Simultaneous fixation and permeabilization of cells. (F-J) Repeat the formaldehyde fixation after the step of simultaneously fixing and permeabilizing.

9) Line 319: the conclusion “the VPS36-A6 mutant exhibited decreased colocalization with CHMP2A” is not supported by data as in the images in suppl. Fig. 10C, CHMP2A-positive vesicles are all clearly positive for Vps36 (wt or mutant).

Response: We apologize that the VPS36 A6 (mutant) representative image in suppl. Fig. 10c was inappropriate to present the overall images we quantified. We chose other images whose VPS36-CHMP2A colocalization efficiency was close to the mode in the quantitative plot to replace the original images in Suppl. Fig. 10c (new Fig. 8a). The new representative image showed that, compared to the VPS36 WT group, there were many CHMP2A vesicles negative for VPS36 A6 protein, which corresponded to our conclusion in the main text.

Fig. R8. HeLa cells were transfected for 24 h with the indicated plasmid. Confocal sections of HeLa cells stained with anti-Myc (VPS36-WT-Myc or VPS36-A6-Myc) (red) and DAPI (blue). Scale bar, 10 μ m.

10) In many IP experiments, a proper negative control is missing, making difficult to evaluate the specificity of the results. For example, in Fig. 3C-D and Fig. 4C, the negative control would include cells transfected with an empty HA-vector (and not untransfected cells), and in Fig. 5D, 6D, 9A and 9C, cells transfected with an empty myc-vector.

Response: We thank you for your comment. All these IP experiments were actually transfected with empty vectors (HA-vector or Myc-vector) to ensure that the cells were exposed to equal amounts of total DNA and transfection reagents. We have added labels to indicate we added

the negative controls (empty vectors) in the revised manuscript (Fig. 4c, 5c, 6d, 8c, 8d and Supplementary Fig. 13).

It would also be interesting to verify whether the silencing of Arl4A increases USP8 recruitment by CHMP2A.

Response: We thank you for this suggestion. We performed a co-IP experiment to assess the USP8 and CHMP2A interaction under Arl4A-depleted conditions. The results showed that Arl4A depletion increased the association of GFP-CHMP2A with HA-USP8 in HeLa cells. We added the co-IP results to the revised manuscript (supplementary Fig. 13).

11-1) Similarly, a negative control is missing in the pull down experiment in Fig. 4B. Also, the entire comassie gel for all the GST- and His- tagged purified proteins should be included to evaluate their purity.

Response: We thank the reviewer for the comment. In this experiment, we used GST protein as the negative pull-down control and Arl1 as the negative VPS36 interacting control. These data showed that His-tagged fusion proteins (Arl4A and Arl1) did not nonspecifically interact with GST alone. In addition, Arl1 did not interact with GST-VPS36. We repeated the experiment and have added the whole Coomassie blue staining image in Fig. 4b (left panel shown below). However, the Coomassie blue image makes it difficult to determine the purity of His-tagged Arl4A and Arl1 when only 5% of the input was loaded. We therefore showed 50% Arl4A and Arl1 inputs (500 ng/per lane), located near 25 kDa, in another membrane to confirm that the purity of His-proteins was higher than 90% (right panel shown below).

Fig. R9. His-tagged Arl4A WT, Q79L, T51N, and Arl1 were pulled down by GST (the negative control) or GST-VPS36 (201-386). Pulled-down Arl4A proteins were probed using an anti-His antibody. Equal inputs of GST fusion proteins used in the assay were visualized by Coomassie Blue staining (left panel). The right panel shows 50% of His-Arl4A and His-Arl1 inputs located at approximately 25 kDa after Coomassie Blue staining. * The star flag represents nonspecific proteins.

11-2) The pull-down shown in Fig. 6B should include anti-His or anti-Arl4A staining for the pull down membrane containing all the samples and not only the input, to verify that the amount of the protein present in the different sample is comparable.

Response: We thank the reviewer for this suggestion. As requested, we have added the Arl4A inputs for different experimental groups in the new Fig. 6b.

12) The membrane shown in Suppl. Fig. 9a should include anti-Arl4A to verify that the lack of difference is not a consequence of a poor silencing. Arl4A levels should also be shown in Suppl. Fig. 4.

Response: Thank you for your advice for improving this manuscript. We apologize for omitting the results of the Arl4A knockdown efficiency in the original manuscript. We had difficulties blotting Arl4A protein in low amounts of cell lysates (original supplementary Fig. 9a) with our Arl4A antiserum. Even so, we did confirm the knockdown of Arl4A by measuring the mRNA levels, which had at least a 50% reduction. As suggested by the reviewer, we have added the Arl4A mRNA level in the corresponding experiment in the new Supplementary Fig. 12a.

On the other hand, it was very difficult to detect the endogenous Arl4A protein of mouse liver tissue in Suppl. Fig. 4 because of high background and nonspecific bands from miscellaneous cell types. To better determine the endogenous Arl4A protein, we isolated hepatocytes from the liver tissue of Arl4A knockout mice for western blotting. We could still not detect the specific band of endogenous Arl4A in Arl4A knockout hepatocytes. As a result, we used mRNA levels to present the Arl4A level in Arl4A knockout mice in Supplementary Fig. 5b.

Fig. R10. Isolation of Arl4A knockout hepatocytes from the livers of mice. Cell lysates were analyzed by immunoblotting with anti-Arl4A and anti-GAPDH antibodies. The C33-A cell lysate served as the positive control group.

13) The authors use different cell lines for different experiments (e.g. HeLa, 293T, C33A, Cos-7, etc.), and it is not often clear why some experiments are performed with one particular cell line and others with a different cell line. This should be clarified. For example, why in Fig. 1

or Fig. 3 the EGFR degradation upon Arl4A overexpression is performed in HeLa while EGFR degradation upon Arl4A silencing is performed in C33A cells?

Response: Thank you for your comment. The expression of Arl4A in two tested epithelial cervical carcinoma cell lines, HeLa and C33-A, was significantly different. C33-A cells expressed much higher Arl4A levels than HeLa cells (new Supplementary Fig. 3a). Since the expression level of Arl4A in C33-A cells is abundant (compared to 21 cell lines examined) (Chen et al., 2020) and overexpression of Arl4A did not further increase the endogenous level in C33-A cells, we therefore performed a knockdown assay in C33-A cells and an overexpression assay in HeLa cells, as shown in original Figs. 1 and 3. In addition, we used the COS-7 cell line to clarify whether Arl4A plays a role in EGFR degradation across different species in the original Supplementary Fig. 3b. We have clarified the reasons for using/changing certain cell lines in the main text.

References:

Chen KJ, Chiang TC, Yu CJ, Lee FS. Cooperative recruitment of Arl4A and Pak1 to the plasma membrane contributes to sustained Pak1 activation for cell migration. *J Cell Sci.* 2020;133(3):jcs233361.

Why 293T cells are used for the experiment in fig. 3D?

Response: For the IP data in original Fig. 3 c and d, we had difficulty determining the interactions between USP8 and CHMP2A in C33-A and HeLa cells due to protein expression issues. When HeLa and C33-A cells were used, we were not able to see any interactions between USP8 and CHMP2A. 293T is a widely used cell line to efficiently produce proteins of interest from DNA plasmids (Tan et al., 2021). Then, we examined the protein expression level of USP8 in 293T cells compared to that in HeLa and C33-A cell lines (Fig. R11). We found that USP8 was most efficiently expressed in the 293T cell line, and its interaction with CHMP2A could therefore be easily assayed. Thus, we chose the alternative cell line 293T to perform the USP8-CHMP2A co-IP experiments. We also clarified the reason for using 293T cells in the main text.

Fig. R11. Expression of USP8 in different cell lines. The cells as indicated were transfected with 2 µg USP8-HA plasmid for 24 hours, and 20 µg/lane of total cell lysates were run for western blotting.

References:

Tan E, Chin CSH, Lim ZFS, Ng SK. HEK293 Cell Line as a Platform to Produce Recombinant Proteins and Viral Vectors. *Front Bioeng Biotechnol.* 2021;9:796991.

14) Not sufficient details are provided in the methods for the work to be reproduced. For example, information about the siRNAs used in this study is not provided. It is also not described which point mutations have been introduced to generate the RNAi-resistant clones.

Response: Thank you for your kind reminders. We have added the image quantification methods in the revised manuscript. The siRNA sequences of Arl4A, VPS36, and USP8 are included in the revised Materials and Methods (see below).

“The sequences of the siRNAs used in this study were as follows: Arl4A siRNA#1: GGAACTCATTGTCCT; Arl4A siRNA #2:TGCCTTCATTTTCAGTCT; VPS36 siRNA: GGGGAATAGCTAACCCAG; USP8 siRNA: TGAAATACGTGACTGTTTAT.”

We also added details about how we constructed RNAi-resistant clones. The Arl4A/VSP36 RNAi-resistant clones contained silent point mutations in the RNAi targeting sequences that decrease siRNA binding to the mRNA. These constructs were generated by two-step site-directed mutagenesis PCR primers as follows:

pSG5-Arl4A^{Res}

F: 5'-CAAGATTTGAGAACTCTTTGTCCTGTCGTTGG-3'

R: 5'-CTCAATTTCTGACAGTGACAAAGAGTTTCTCAAATCTTG -3'.

pcDNA3.1-VPS36^{Res}

F: 5'-TTGCTGAGCATGGGCATCGCGAACCCAGTTACCAGAGAAACC-3'

R: 5'-TTGCTGAGCATGGGCATCGCGAATCCGGTGACCAGAGAAACC-3'.

The description about how image analysis has been performed is also missing.

Response: We thank the reviewer for this comment. We apologize for the lack of image analysis methods in the original manuscript. We have added the image analysis details and quantification methods in the revised manuscript.

In Fig. 1a, the intensity of EGF-555 was measured using ImageJ software. The outline of EGF-555-positive areas was used to define a whole cell body or referred to as the region of interest (ROI). This mask was then used as a reference to quantify EGF-555 intensity/cell in the image.

In Fig. 3c, EGFR signals localized inside the Rab5 (Q79L)-enlarged endosomes (MVB) were quantified using ImageJ. With the GFP-Rab5 Q79L signals as the mask to define the outline of the endosomes, endosome diameters $>2 \mu\text{m}$ were taken for calculation of EGFR intensities.

In revised Figs. 4d, 7a-b, and 8a and **Supplementary Figs. 2, 9, 10, and 11**, we used ZEN software to determine the colocalization coefficient of two proteins of interest. In Fig. 4D, the Arl4A signal served as the mask for the ROI per cell. VPS36-Arl4A colocalized pixels/total VPS36 pixels within the mask were calculated to obtain the colocalization coefficients in the images. In Figs. 7a-b and 8a, we also used ZEN software to determine the colocalization coefficient of two proteins of interest. In Supplementary Figs. 2, 9, 10 and 11, the EGFR signals served as the mask for defining the ROI per cell. EGFR-EEA1 or -Lamp1 colocalized pixels/total EGFR pixels within the mask were calculated as the colocalization coefficients.

In the revised Fig. 5a, we used Imaris (Bitplane) software to quantify the 3D colocalization of Arl4A and Lamp1 in super resolution images. The Arl4A signals served as the mask for defining the ROI per cell. Using these masks, the 3D colocalization coefficient values were calculated by Lamp1 pixels that colocalize with Arl4A/total Lamp1 pixels in these masks.

In the revised Fig. 7c, we used Imaris (Bitplane) software to quantify the 3D colocalization of VPS36 and GFP-Rab5 Q79L in super-resolution images. The GFP-Rab5 Q79L signals served as the mask for defining the ROI per cell. Using these masks, the 3D-colocalization coefficient values were calculated by VPS36 pixels that colocalize with GFP-Rab5 Q79L/total VPS36 pixels in these masks.

Minor comments:

-line 51: the references are separated

Response: The references error has been corrected (Line 51).

- In Figure 6B, and in the relative figure legend, it is mentioned GST-VPS36 (201-386), however in the main text (line 283), this is referred as GST-VPS36 (330-386).

Response: We appreciate the reviewer's careful reading of our manuscript and have corrected the error (Line 257-259).

"Using an *in vitro* GST pulldown assay, we confirmed that GST-VPS36-WT-(201-386) interacted with Arl4A but not with the A3, A4, or A6 VPS36 mutants (Fig. 6b)"

- line 302 (and relative figure legend): expression of Arl4A dominant negative mutant (and not overexpression)

Response: The mistake has been corrected.

- line 372: "because"

Response: The mistake has been corrected on line 366.

-line 433: "Arl4Acan"

Response: The mistake has been corrected on line 431.

- line 866: Arl4A WT is not the GTP-bound form.

Response: The mistake has been corrected.

-line 953: "Cell lysates were analyzed by immunoblotting with anti-Arl4A", but this is not shown in suppl. Fig. 3D. It should be included.

Response: We appreciate the reviewer's careful reading of our manuscript and have corrected the error (new Supplementary Fig. 4e legend) The results of the previous Arl4A knockdown experiment are presented in mRNA. The Arl4A knockdown efficiency in A549, H1975, and PE089 cells is shown in Fig. 2A.

- line 956: "Alr4a"

Response: The mistake has been corrected.

- lines 1023-1025: The immunoprecipitation shown in suppl. Fig. 9c is not a GFP-TRAP. The figure legend should be corrected.

Response: The mistake has been corrected.

REVIEWER COMMENTS

Reviewer #1 (Remarks to the Author):

The manuscript has been thoroughly revised, included with new experiments. This has markedly improved the data and reinforced the conclusions of the study. However, there are some remaining issues that still need to be addressed.

There is an overall problem with data normalisation. Some quantifications (for example Figure 4C, Fig 8, Fig 9, etc) are normalised to controls leading to a loss of variation in the control conditions. The authors go on to perform statistical analysis comparing this condition to others but this statistical analysis is void due to this lack of error in the controls. The authors need to reanalyse the data, for example by normalising it to the average control results to retain the variation within this control condition before performing any statistical comparisons.

Additionally, some EGFR trafficking and co-localisation studies have been performed without prior pre-incubation in serum-free conditions. This makes these analyses unclear as there is a confounding effect of the potential growth factors present in the serum. The authors need to repeat these experiments including a pre-incubation step in serum-free media.

Other specific comments to some of the queries here below:

"Although SNAP-fused EGFR has been used to screen drug leads (Fu et al. 2021), we did not find published papers using SNAP-fused EGFR to study the EGF-induced EGFR endocytic route. This may be because the 182 residue polypeptide length of the SNAP-tag may interfere with EGF binding to the EGFR and/or EGF-induced EGFR signaling or degradation. In addition, a previous study indicated that selection of the dye for labeling SNAP-fusion proteins is crucial, as many dyes suffer from either rapid photobleaching or high levels of nonspecific staining (Bosch et al., 2014)."

I don't necessarily agree with this statement, we have used SNAP-EGFR before to monitor its trafficking without any issues, and have very good data with various commercial and non-commercial fluorescent SNAP-tag probes. The use of the anti-EGFR antibody in living cells is also a valid alternative approach so this is now acceptable.

Supply Fig 1 looks good but please include size bars

"Arl4A KD levels should be assessed by WB." OK, please also submit original uncropped blots for all WBs

"For the rescue experiment of c-Met in HeLa cells, we acknowledge that expressing Arl4ARes can only achieve 58% rescue of c-Met in Arl4A-depleted HeLa cells. We have modified our discussion, and a detailed description is presented in the revised manuscript (Lines 125-127)." Sorry but I could not find this partial rescue comment anywhere in the manuscript

"Results at 0 minutes on Fig 1F don't match those of Fig 1E? Same problem in fig 2A and the rest of Fig 2" OK, can this explanation be added to the manuscript? Thanks

"Suppl Fig 8: why is wt egfr in endosomes?" "PE089 pictures don't match quantification given for LAMP1 co-localisation." Experiments should be repeated after serum starvation to avoid confounding effects for Fig S9, S10 and S11

"Fig 3: why is Arl4A over expression not performed in C33-A?" Fig. R5, for reviewer only : please include this explanation and info in the manuscript

"Fig4: what about the mirystoylation mutant?" Ok, please add co-IP to the manuscript

"Fig6: why was A3 and A4 not further investigated?" Ok please include this information in the manuscript

Reviewer #2 (Remarks to the Author):

After reviewing the authors' responses to critiques and carefully reading the revised manuscript, I unfortunately continue to see this work as largely superficial (limiting its significance), failing to provide clear mechanistic insights into the observations made. As the work stands, I am convinced that Arl4 plays some role in the endolysosomal sorting of ubiquitin-modified EGF receptor (depletion of Arl4 clearly reduces EGFR signal relative to control cells, while overexpression of Arl4 increases the stability of ubiquitin-modified EGFR), but direct connections between Arl4 and its purported ability to modulate ESCRT function are tenuous at best. In the absence of some mechanistic understanding of the role that Arl4 plays in EGFR transport and turnover, I continue to believe the study is too preliminary for publication.

Major comments:

I previously asked that the authors provide some level of mechanistic insight into how overexpression of Arl4 could lead to increased levels of ubiquitin-modified EGFR in cells, but instead of experimentally exploring this question, the authors chose to cite other studies that have suggested that removal of ubiquitin is important for cargo sorting into intraluminal vesicles. This is disappointing. Based on the data shown, Arl4 binding to ESCRT-II may negatively impact its function, but without any mechanistic study of this potential form of inhibition (which would be extremely interesting), it remains unclear whether this is a direct role for Arl4 in regulating ESCRT function, or some indirect role of Arl4.

Along these lines, Vps36 appears to interact directly with Arl4 (best in a GDP-bound state), which is increased with EGF stimulation. Is this interaction between Vps36 and Arl4 phosphorylation dependent? Such a finding could help strengthen the possibility that Arl4 regulates the function of ESCRT-II directly. As it stands, the authors show that Arl4 (in a largely GDP-bound state) co-localizes with Vps36 in cells, and they define a mutant form of Vps36 (A6) that diminishes their ability to bind. Indeed, the A6 mutant appears to associate less well with endosomes and depletion of Arl4 impairs Vps36 localization to endosomes – how then does overexpression of the A6 mutant enhance EGFR degradation? Is this mediated by an endosomal function of ESCRT-II or some non-endosomal function of Vps36? Again, without mechanistic insights, the observations shown raise internal inconsistencies that are difficult to resolve.

The A6 mutant is also shown to associate less well with CHMP2A. Why? The region mutated has not previously been shown to play any role in the association between ESCRT-II and ESCRT-III. Is this due to the inability of the A6 mutant to localize to endosomes? In a similar vein, Arl4 overexpression appears to reduce the interaction between CHMP2A and USP8/UBPY and Arl4 depletion increases their association. How? Defining the mechanism(s) underlying these effects would demonstrate a direct relationship between the function of Arl4 and the ESCRT machinery, which would be highly interesting and worthy of publication in Nature Communications.

Lastly, several of the experiments now included basically repeat those of others. For example, the authors show that Vps36 depletion delays EGFR degradation. This has been known for decades. However, overexpression of the A6 mutant, which doesn't bind to Arl4 well, enhances EGFR degradation more than the wild-type form of Vps36. How is this accomplished, especially considering that the A6 mutant doesn't accumulate at endosomes well? Resolving these types of questions with more mechanistic studies is critical to unravel the role of Arl4 in the endolysosomal pathway.

Reviewer #3 (Remarks to the Author):

The revised manuscript addresses most of the comments raised during review of the initial submission. The revised manuscript is significantly improved, and represents novel and interesting findings that will be of interest to the field of receptor signalling and membrane traffic. Perhaps the authors can also consider the following:

Some of the data were analyzed using ANOVA; however, the post-hoc test used does not appear to be described; this post-hoc test is what would have been used to allow determination of p-values. This information should be included.

Many of the comments were addressed with data that is presented only in the "Response to reviewer comments" document. While this document should also be public, being able to reference some of these new data from the manuscript text would help the reader better appreciate context for some of the data in the manuscript. Understandably, there may be space limitations. If there is an opportunity to include it, key data such as the sensitivity to gefinitib (Fig. R5) may be worth including in a supplemental figure. Also, stating whether these data in Figure R5 are from cells in the presence or absence of EGF stimulation would be useful.

Reviewer #4 (Remarks to the Author):

The authors have greatly improved the quality of the manuscript and addressed the majority of the reviewer's concerns. However, it is still unclear whether depletion of Arl4A increases EGFR delivery to late endosomes. Lamp1 signal intensity is very high in the samples where higher colocalization is measured and very low when low colocalization is measured. Therefore, it is difficult to conclude whether the obtained values are a consequence of the saturation of the signal or of a real colocalization (Suppl. Fig. 9; Suppl. Fig. 10). Also, In Suppl. Fig. 11 it is difficult to compare control and siArl4A cells, as in the panel where Lamp1 staining is shown, the siArl4A cells at 45 and 60 minutes have very weak staining for both markers compared to control cells. Finally, the concern about the use of only one siRNA targeting Arl4A (or rescue experiments) still remains as in the revised version the authors only added a second siRNA targeting Arl4 in one experiment, to confirm that it reduces EGFR expression

Replies to Reviewers

Reply to Reviewer #1

We sincerely appreciate the reviewer's constructive comments and have addressed each of the specific points as described below:

Reviewer #1 (Remarks to the Author):

The manuscript has been thoroughly revised, included with new experiments. This has markedly improved the data and reinforced the conclusions of the study. However, there are some remaining issues that still need to be addressed.

There is an overall problem with data normalisation. Some quantifications (for example Figure 4C, Fig 8, Fig 9, etc) are normalised to controls leading to a loss of variation in the control conditions. The authors go on to perform statistical analysis comparing this condition to others but this statistical analysis is void due to this lack of error in the controls. The authors need to reanalyse the data, for example by normalising it to the average control results to retain the variation within this control condition before performing any statistical comparisons.

Response: As suggested by the reviewer, we have reanalyzed the data with control conditions without error bars and used the same method for our newly included data in the current version of Figs. 1 (c, d, e), 2a, 4c, 5 (b, c), 6 (b, d), 8 (b, c, d, e), 9e, S5a, S14, S16, S17, and S19. We normalize each experimental value to the experimental value that is closest to the average value per experiment to obtain the variations within that control condition before making statistical comparisons.

Additionally, some EGFR trafficking and co-localization studies have been performed without prior pre-incubation in serum-free conditions. This makes these analyses unclear as there is a confounding effect of the potential growth factors present in the serum. The authors need to repeat these experiments including a pre-incubation step in serum-free media.

Response: Thank you for your comment. All EGFR trafficking experiments under EGF stimulation were pre-starved for at least 3-6 hours in serum-free medium. These data are included in Figs. 1a, 3c, 5a, and Supplementary Figs. 1, 2, 10, 11 and 12, and the serum-starving process was also indicated in the figure legends. To avoid confusion among readers, we have emphasized that serum starvation meant the serum-free condition in the EGFR degradation assay in Materials and Methods (line 528).

In Fig. 4d, Fig. 7, and Fig. 8a, on the other hand, we observed the co-localization only under normal conditions (steady-state). Therefore, we performed the experiments without EGF stimulation and did not include the process of serum starvation.

Other specific comments to some of the queries here below:

“Although SNAP-fused EGFR has been used to screen drug leads (Fu et al. 2021), we did not find published papers using SNAP-fused EGFR to study the EGF-induced EGFR endocytic route. This may be because the 182 residue polypeptide length of the SNAP-tag may interfere with EGF binding to the EGFR and/or EGF-induced EGFR signaling or degradation. In addition, a previous study indicated that selection of the dye for labeling SNAP-fusion proteins is crucial, as many dyes suffer from either rapid photobleaching or high levels of nonspecific staining (Bosch et al., 2014).”

I don't necessarily agree with this statement, we have used SNAP-EGFR before to monitor its trafficking without any issues, and have very good data with various commercial and non-commercial fluorescent SNAP-tag probes. The use of the anti-EGFR antibody in living cells is also a valid alternative approach so this is now acceptable.

Supply Fig 1 looks good but please include size bars

Response: Thank you for the correction. The omission has been corrected in Supplementary Fig. 1.

“Arl4A KD levels should be assessed by WB.” OK, please also submit original uncropped blots for all WBs.

Response: We thank the reviewer for this suggestion. All original uncropped blots can be seen in Supplementary Figs. 20-38.

“For the rescue experiment of c-Met in HeLa cells, we acknowledge that expressing Arl4ARes can only achieve 58% rescue of c-Met in Arl4A-depleted HeLa cells. We have modified our discussion, and a detailed description is presented in the revised manuscript (Lines 125-127).” Sorry but I could not find this partial rescue comment anywhere in the manuscript

Response: Thank you for your kind comment. We have added this description in lines 122-124.

“Not all cells take up plasmids with transient transfection; thus, the result of the rescue experiment reached only 58% recovery.”

“Results at 0 minutes on Fig 1F don't match those of Fig 1E? Same problem in fig 2A and the rest of Fig 2” OK, can this explanation be added to the manuscript? Thanks

Response: Thank you for your kind comment. We have added this remark in the legend of Figures 1f-g, 2b-d, S4e and S18c.

“To compare EGFR degradation rates, we adjusted EGFR protein levels at time 0 in each group to be the same for western blot analysis.”

“Suppl Fig 8: why is wt egfr in endosomes?” “PE089 pictures don’t match quantification given for LAMP1 co-localisation.” Experiments should be repeated after serum starvation to avoid confounding effects for Fig S9, S10 and S11

Response: Thank you for your kind comment. In **original Supplementary Figs. 9, 10, and 11**, we first performed a 6-hour serum starvation before proceeding with EGF treatment. We have indicated this experimental step in the figure legends of **new Supplementary Figs. 10, 11, and 12**.

“cells were transfected with siControl or siArl4A for 48 h and then serum starved for 6 h before 100 ng/ml EGF treatment for indicated times.”

“Fig 3: why is Arl4A over expression not performed in C33-A?” Fig. R5, for reviewer only : please include this explanation and info in the manuscript

Response: Thank you for your kind comment. We mention the highly expressed Arl4A of C33-A in lines **112-115**, and the reason for using HeLa cells for the Arl4A overexpression experiment is also given in lines **137-139**. For the information in the previous Fig. R5, we explain why we assayed the effect of Arl4A expression on the CHMP2-DUBs interaction in 293T for that 293T is much better for DUBs expression (in lines **298-299**).

Lines **112-115** “The cervical cancer cell line C33-A has been shown to exhibit a high level of Arl4A protein (Supplementary Fig. 3a) (Chen et al., 2020). To confirm Arl4A is involved in the lysosomal degradation of membrane receptors, we examined the protein levels of EGFR, hepatocyte growth factor receptor (c-Met) and transferrin receptor (TfR) in Arl4A-depleted C33-A cells.”

Lines **137-139** “Since the endogenous level of Arl4A in C33-A cells is high (Supplementary Fig. 3a) but low in HeLa cells, we knocked down Arl4A in C33-A cells and overexpressed it in HeLa cells.”

Lines **298-299** “We assayed the recruitment in Arl4A-overexpressing 293T cells for that 293T cell line has better ability to express DUBs.”

“Fig4: what about the mirystoylation mutant?” Ok, please add co-IP to the manuscript
Response: We thank the reviewer for this suggestion. We have added the co-IP data in Supplementary Fig. 13, and the corresponding description in lines 224-226.

Lines 224-226 “We found their interaction requires Arl4A membrane targeting for Arl4A G2A was unable to colocalize and be co-immunoprecipitated with VPS36 (Fig. 4d, Supplementary Fig.13)”

“Fig6: why was A3 and A4 not further investigated?” Ok please include this information in the manuscript

Response: Thank you for your kind comment. We have added this remark in lines 247-249 as follows;

Line 247-249 “We aligned and found that the amino acid region ³⁵²LAKER³⁵⁶ in which the A6 mutations reside was highly conserved among species (Fig. 6c), we therefore used the A6 mutant for further functional studies.”

Reply to Reviewer #2

We sincerely appreciate the reviewer’s comments and have addressed each of the specific points as described below:

Reviewer #2 (Remarks to the Author):

After reviewing the authors’ responses to critiques and carefully reading the revised manuscript, I unfortunately continue to see this work as largely superficial (limiting its significance), failing to provide clear mechanistic insights into the observations made. As the work stands, I am convinced that Arl4 plays some role in the endolysosomal sorting of ubiquitin-modified EGF receptor (depletion of Arl4 clearly reduces EGFR signal relative to control cells, while overexpression of Arl4 increases the stability of ubiquitin-modified EGFR), but direct connections between Arl4 and its purported ability to modulate ESCRT function are tenuous at best. In the absence of some mechanistic understanding of the role that Arl4 plays in EGFR transport and turnover, I continue to believe the study is too preliminary for publication.

Major comments:

I previously asked that the authors provide some level of mechanistic insight into how overexpression of Arl4 could lead to increased levels of ubiquitin-modified EGFR in cells, but instead of experimentally exploring this question, the authors chose to cite

other studies that have suggested that removal of ubiquitin is important for cargo sorting into intraluminal vesicles. This is disappointing. Based on the data shown, Arl4 binding to ESCRT-II may negatively impact its function, but without any mechanistic study of this potential form of inhibition (which would be extremely interesting), it remains unclear whether this is a direct role for Arl4 in regulating ESCRT function, or some indirect role of Arl4.

Response: We thank the reviewer for this critical comment. To better answer this question, we have added several new results to demonstrate a direct role of Arl4A in regulating ESCRT function. Together with our previous data, our evidences support two important conclusions:

Conclusion I “Arl4A-binding retains VPS36 to the late endosome, causing delayed EGFR deubiquitination and degradation.”

Evidences:

1. GDP-bound Arl4A directly interacts with VPS36 and co-localizes at the endosomes (Fig. 4b-d).
2. The interaction of VPS36 to Arl4A stabilizes its endosomal localization, for that either with VPS36 A6 expression or Arl4A depletion decreases VPS36 endosomal detainment (Fig. 7a-b and Supplementary Fig. 15).
3. Without Arl4A binding, either with VPS36 A6 expression or Arl4A depletion, VPS36 forms less stable complex with CHMP2A and CHMP6 (ESCRT-III components) (Fig. 8a-b, new Fig. 8c, new Supplementary Fig. 16).
4. Without Arl4A binding, VPS36-driven EGFR degradation and deubiquitination levels are accelerated (Fig. 9).

Conclusion II - “Arl4A-VPS36 interaction modulates the level of ubiquitin-modified EGFR in a USP8-dependent manner.”

Evidences:

1. Without Arl4A binding, the deubiquitinating enzyme USP8-CHMP2A interaction is specifically enhanced (Fig. 8d-e, Supplementary Fig. 17).
2. Depletion of USP8 suppresses ligand-induced EGFR degradation accelerated by Arl4A depletion (Supplementary Fig. 18).
3. Depletion of USP8 suppresses EGFR deubiquitination levels accelerated by Arl4A depletion (new Supplementary Fig. 19).

In conclusion, Arl4A-VPS36 interaction restrains VPS36 to the endosomal compartment and causes a more stable transit of ESCRT-II-ESCRT-III, which is detrimental for further recruitment of the deubiquitinating enzyme to complete the final

ESCRT cycle toward MVBs. These facts explain the initial phenomenon: how Arl4A could potentially attenuate EGFR degradation and prolong ubiquitination (Fig. 1-3).

The Emr group has previously reported that ESCRT-II conformational changes are critical to activate and mediate the architecture of ESCRT-III Snf7/CHMP4b assembly (Henne et al., 2012). Therefore, we propose that binding of Arl4 to the ESCRT II-component VPS36 stick ESCRT-II-ESCRT-III complex, disadvantaging the architecture of ESCRT-III for recruiting USP8 /UBPY. This slows the ESCRT assembly and disassembly cycle as well as ligand-induced EGFR degradation. We agree with the reviewer that a detailed mechanistic study on how Arl4A binding alters the assembly of ESCRT complex is extremely interesting; however, it needs in-depth structural examination and/or *in vitro* reconstitution analysis of ESCRT complex thus is beyond the scope of this study. We hope the reviewer could appreciate the punchline of this work: a novel regulation for turnover rate of multiple receptor tyrosine kinases (EGFR and c-MET) from GDP-bound Arl4A protein at endosomes.

References:

Henne, W.M., N.J. Buchkovich, Y. Zhao, and S.D. Emr. 2012. The endosomal sorting complex ESCRT-II mediates the assembly and architecture of ESCRT-III helices. *Cell*. 151:356-371.

Along these lines, Vps36 appears to interact directly with Arl4 (best in a GDP-bound state), which is increased with EGF stimulation. Is this interaction between Vps36 and Arl4 phosphorylation dependent? Such a finding could help strengthen the possibility that Arl4 regulates the function of ESCRT-II directly.

Response: We thank the reviewer for these comments. We previously found that the phosphorylation of Arl4 on S143 is induced by fibronectin-PAK signaling, and would enhance the plasma membrane localization and interaction with chaperone protein HYPK. Our preliminary data also shows this phosphorylation mimic enhanced the Arl4A-VPS36 interaction (Fig. R1). Therefore, it is very likely that EGF signaling could regulate the interaction between Arl4A and Vps36 through phosphorylation or other PTM on these two proteins. However, we did not detect S143 phosphorylation of Arl4A under EGF-stimulated signaling.

We agree with the reviewer that such finding could strengthen and provide detailed mechanistic insight on the regulation of ESCRT complex under EGF signaling. It will need a thorough PTM screen and analysis on both Arl4A and VPS36, and perhaps other ESCRT components, thus is beyond the scope of this work.

Fig. R1. HeLa cells were transfected with phospho-mimetic Arl4A (S143D), phospho-defective mutant (S143A), and VPS36-Myc for 16-18 hours. VPS36-Myc was immunoprecipitated and resuspended in sample buffer for western blotting and detection of Arl4A signals.

References:

Lin, M.C., C.J. Yu, and F.S. Lee. 2022. Phosphorylation of Arl4A/D promotes their binding by the HYPK chaperone for their stable recruitment to the plasma membrane. *Proc Natl Acad Sci U S A.* 119:e2207414119.

As it stands, the authors show that Arl4 (in a largely GDP-bound state) co-localizes with Vps36 in cells, and they define a mutant form of Vps36 (A6) that diminishes their ability to bind. Indeed, the A6 mutant appears to associate less well with endosomes and depletion of Arl4 impairs Vps36 localization to endosomes – how then does overexpression of the A6 mutant enhance EGFR degradation? Is this mediated by an endosomal function of ESCRT-II or some non-endosomal function of Vps36? Again, without mechanistic insights, the observations shown raise internal inconsistencies that are difficult to resolve.

Response: We thank the reviewer for these comments We show that A6 expression, which exhibits higher endosomal dynamics (new Fig. 7c), promotes the deubiquitination and degradation of EGFR (Fig. 9). We believe the effect of VPS36 A6 is an endosomal function of ESCRT-II.

The A6 mutant is also shown to associate less well with CHMP2A. Why? The region mutated has not previously been shown to play any role in the association between ESCRT-II and ESCRT-III. Is this due to the inability of the A6 mutant to localize to endosomes?

Response: We thank the reviewer for these comments. Previous studies have shown that the ESCRT-II and ESCRT-III complexes are transiently associated with the endosomal membrane (Babst et al., 2002a; Babst et al., 2002b) because they are immediately

disassembled by VPS4 into the cytoplasm after cargo sorting (Saksena et al., 2009). From these studies, the cycling of ESCRT-II/III components for cargo sorting are transient, and only by knockdown or disruption of VPS4 function can their endosomal puncta be clearly observed in the steady state (Jouvenet et al., 2011). In our **new Fig 7c**, we show that siVPS4 could restore the endosomal targeting of VPS36 A6, demonstrating that **VPS36 A6 is still capable of endosomal targeting** for driving EGFR degradation. Consequently, VPS36 A6 sounds to trigger faster cargo sorting than VPS36 WT, it thus cycles more rapidly between the cytosolic and endosome compartments, leading to greater difficulty in detecting the endosomal localization of A6 than WT in the steady state (**Fig. 7a-b and Fig. 8a**).

In the same context, the reason why the the A6 mutant associates less well with CHMP2A/6 in co-immunoprecipitation (**Fig. 8b and new Supplementary Fig. S16a**) is most likely due to the rapid and transient but not abrogated association of A6 with the endosomal compartment. Similarly, by knocking down Arl4A to mimic A6 expression, VPS36 decreased its association with the ESCRT-III components CHMP2A/CHMP6, and decreased its co-localization with the CD63 positive endosomes (**Fig. 8c, new Supplementary Fig. 15 and new Supplementary Fig. 16b**) imply a faster cycled ESCRT pipeline.

We greatly appreciate the reviewer's question, as it helps us to clarify our model in more detail. We have therefore revised our description of the A6 mutant and Arl4A knockdown when discussing the association of ESCRT components in the results and discussion below.

“VPS36-Myc also localized to small punctate structures that were positive for the MVB marker CD63 and Rab5 Q79L. In comparison, VPS36-A6-Myc puncta showed lower MVB localization (Fig. 7a-b). In addition, Arl4A depletion that simulates perturbed Arl4A-VPS36 interaction also decreased the targeting of VPS36 at the MVBs (Supplementary Fig. 15)” (**Lines 267-271**)

“We next asked whether Arl4A binding may affect the association of ESCRT-II and ESCRT-III. We found that, compared with VPS36-WT, VPS36-A6 exhibited both decreased colocalization (Fig. 8a) and association with the ESCRT-III component CHMP2A (Fig. 8b). In addition, Arl4A knockdown also abated VPS36 and CHMP2A interaction (Fig. 8c). These phenomena were also observed with other members of the ESCRT-III complex, such as CHMP6 (Supplemental Fig. 16). Taken together, these results suggested that Arl4A binding to VPS36 affected the ESCRT-II and ESCRT-III

association.” (Lines 283-289)

“Currently, we hypothesize that Arl4A binding to VPS36 alters the association of VPS36 and the ESCRT-III component CHMP2A, which delays CHMP2A-deubiquitinating enzyme USP8 recruitment, that in turn affects EGFR ubiquitination levels (Fig. 9g).” (Lines 335-338)

References:

Babst, M., D.J. Katzmann, E.J. Estepa-Sabal, T. Meerloo, and S.D. Emr. 2002a. Escrt-III: an endosome-associated heterooligomeric protein complex required for mvb sorting. *Dev Cell*. 3:271-282.

Babst, M., D.J. Katzmann, W.B. Snyder, B. Wendland, and S.D. Emr. 2002b. Endosome-associated complex, ESCRT-II, recruits transport machinery for protein sorting at the multivesicular body. *Dev Cell*. 3:283-289.

Saksena, S., J. Wahlman, D. Teis, A.E. Johnson, and S.D. Emr. 2009. Functional reconstitution of ESCRT-III assembly and disassembly. *Cell*. 136:97-109.

Jouvenet, N., M. Zhadina, P.D. Bieniasz, and S.M. Simon. 2011. Dynamics of ESCRT protein recruitment during retroviral assembly. *Nat Cell Biol*. 13:394-401.

*In a similar vein, Arl4 overexpression appears to reduce the interaction between CHMP2A and USP8/UBPY and Arl4 depletion increases their association. **How?** Defining the mechanism(s) underlying these effects would demonstrate a direct relationship between the function of Arl4 and the ESCRT machinery, which would be highly interesting and worthy of publication in Nature Communications.*

Response: We thank the reviewer for these comments. In our discussion, we infer that Arl4A binding to VPS36 sustains the association of VPS36 and ESCRT-III component (CHMP2A and CHMP6), resulting in a reduction of CHMP2A-USP8 interaction, which could further affect EGFR ubiquitination levels. These comments are sequentially supported by Fig. 8b, new Supplementary Fig. 16a, and Fig. 9. We recently also used Arl4 knockdown system to confirm the Fig. 8b that Arl4A existence affects the ESCRT-II and ESCRT-III association (Fig. 8c and new Supplementary Fig. 16b).

The Emr group has proposed that ESCRT-III has distinct architectural stages modulated by ESCRT-II to mediate cargo capture and vesicle formation through ordered assembly (Henne et al., 2012; Teis et al., 2010). How stronger association between ESCRT-II and ESCRT-III affects the interaction of the CHMP2A-USP8 is unknown. However, it has been reported that interference with the ESCRT recruitment process could block subsequent recruitment processes (Henne et al., 2012; Teis et al., 2010). Additional structure-based details of the ESCRT-III component CHMP2A-USP8 interaction in the

presence or absence of Arl4A require cryo-EM and structure-based analyses to move one step closer to understanding the effects of Arl4A binding to VPS36 on the conformational dynamics and molecular architecture of the ESCRT-III complex. This detailed information will require years of study and is more likely to be found in future publications. We would appreciate the reviewer's understanding that the attenuation of EGFR degradation by Arl4A-VPS36 interactions is the main content that we have carefully tried to address in this manuscript.

References:

Teis, D., S. Saksena, B.L. Judson, and S.D. Emr. 2010. ESCRT-II coordinates the assembly of ESCRT-III filaments for cargo sorting and multivesicular body vesicle formation. *Embo j.* 29:871-883.

Henne, W.M., N.J. Buchkovich, Y. Zhao, and S.D. Emr. 2012. The endosomal sorting complex ESCRT-II mediates the assembly and architecture of ESCRT-III helices. *Cell.* 151:356-371.

Lastly, several of the experiments now included basically repeat those of others. For example, the authors show that Vps36 depletion delays EGFR degradation. This has been known for decades. However, overexpression of the A6 mutant, which doesn't bind to Arl4 well, enhances EGFR degradation more than the wild-type form of Vps36. How is this accomplished, especially considering that the A6 mutant doesn't accumulate at endosomes well? Resolving these types of questions with more mechanistic studies is critical to unravel the role of Arl4 in the endolysosomal pathway.

Response: Thank you for your kind reminder. We feel we were attempting to answer our new hypothesis based on solid foundations, which served as indispensable positive controls in many experiments. For the example mentioned by the reviewer, we confirmed that VPS36 plays a role in EGFR degradation, whereas substitution of an Arl4A-binding defective VPS36 (A6) in cells accelerates EGFR degradation. This conclusion cannot be drawn without appropriate comparisons with the previous results. We appreciate that the reviewer raises the issue of the cytosolic-prone mutant of VPS36 A6. The ESCRT sorting pathway is composed of component assembly and finally disassembly after sorting the cargos into the multivesicular body (MVB). After disassembly, ESCRT components are dissociated into the cytosol by VPS4 (Im and Hurley, 2008; Quinney et al., 2019). Based on the fact that VPS36 A6 accelerated EGFR degradation under EGF treatment conditions and exhibited less colocalization with the MVB, we hypothesize that VPS36 A6 facilitates rapid sorting of cargo into MVBs, leading to rapid dissociation of VPS36 A6 from the MVB into the cytosol, and its endosome localization/association is therefore difficult to detect. To confirm this, we blocked the final step of ESCRT sorting by knocking down VPS4A and VPS4B. The

results showed that VPS36 A6 is able to accumulate well and colocalize with MVB markers (see Fig. 7c). Overall, it seems that subtle differences in the localization of VPS36 and VPS36 A6 point to fine-tuning in the localization rather than to all- or nothing ‘on/off’ switch. Our result confirms that Arl4 A6 can indeed associate with the MVB compartment and that ESCRT sorting for EGFR degradation is faster without the Arl4A-VPS36 interaction.

References:

Im, Y.J., and J.H. Hurley. 2008. Integrated structural model and membrane targeting mechanism of the human ESCRT-II complex. *Dev Cell*. 14:902-913.

Quinney, K.B., E.B. Frankel, R. Shankar, W. Kasberg, P. Luong, and A. Audhya. 2019. Growth factor stimulation promotes multivesicular endosome biogenesis by prolonging recruitment of the late-acting ESCRT machinery. *Proc Natl Acad Sci U S A*. 116:6858-6867.

Reply to Reviewer #3

We sincerely appreciate the reviewer’s comments and have tried to address each of the specific points as described below:

Reviewer #3 (Remarks to the Author):”

The revised manuscript addresses most of the comments raised during review of the initial submission. The revised manuscript is significantly improved, and represents novel and interesting findings that will be of interest to the field of receptor signalling and membrane traffic. Perhaps the authors can also consider the following:

Some of the data were analyzed using ANOVA; however, the post-hoc test used does not appear to be described; this post-hoc test is what would have been used to allow determination of p-values. This information should be included.

Response: We thank the reviewer for this suggestion. We used GraphPad Prism for one-way ANOVA statistical analysis with Tukey’s test as post hoc test to value the significance between each paring of tested groups. We have added this detail in all the figure legends with ANOVA analysis and noted in the Materials and Methods in lines 647-650.

“Statistical comparisons between treatments were performed by parametric t tests (Student’s t tests) or one-way analysis of variance (ANOVA) in GraphPad Prism 8. For the post hoc test, we used Tukey’s method to compare all possible group pairings. Significant differences are indicated in the figure (*, P < 0.05. **, P < 0.01. ***, P < 0.001. ****, P < 0.0001).” (Lines 647-650).

Many of the comments were addressed with data that is presented only in the “Response to reviewer comments” document. While this document should also be public, being able to reference some of these new data from the manuscript text would help the reader better appreciate context for some of the data in the manuscript. Understandably, there may be space limitations. If there is an opportunity to include it, key data such as the sensitivity to gefitinib (Fig. R5) may be worth including in a supplemental figure.

Response: As suggested by the reviewer, we added two comments and related data in our 2nd revised manuscript.

1. We added the result of relative Arl4A protein level in HeLa cells after overexpression of Arl4A compared to C33A cells in the **new Supplemental Fig. 4f** and supplemented the statement in the Results section as follows,

“Consistent with this observation, HeLa cells that express about three times more Arl4A protein than C33-A cells (Supplementary Fig. 4f), exhibit an increased EGFR half-life (Fig. 1g).” (Lines 142-144).

2. We have included the results on the sensitivity of gefitinib (**new Supplementary Fig. 9**) in the 2nd revised manuscript and added the following to the statement in the Results section,

“Though EGFR level is relatively low in C33A cells (Supplementary Fig. 3a), we evaluated the EGFR-dependent effect by treating cells with gefitinib (an EGFR inhibitor) and observed a decrease in cell growth in an inhibitor a dose-dependent manner (Supplementary Fig. 9a). Further, the gefitinib inhibition of cell growth was exacerbated by concomitant Arl4A knockdown, with or without EGF stimulation (Supplementary Fig. 9b-d).” (Lines 157-162).

Reply to Reviewer #4

We sincerely appreciate the reviewer’s comments and have tried to address each of the specific points as described below:

Reviewer #4 (Remarks to the Author):

The authors have greatly improved the quality of the manuscript and addressed the majority of the reviewer’s concerns.

However, it is still unclear whether depletion of Arl4A increases EGFR delivery to late endosomes. Lamp1 signal intensity is very high in the samples where higher colocalization is measured and very low when low colocalization is measured. Therefore, it is difficult to conclude whether the obtained values are a consequence of

the saturation of the signal or of a real colocalization (Suppl. Fig. 9; Suppl. Fig. 10). Also, In Suppl. Fig. 11 it is difficult to compare control and siArl4A cells, as in the panel where Lamp1 staining is shown, the siArl4A cells at 45 and 60 minutes have very weak staining for both markers compared to control cells.

Response: We thank the reviewer for the comment. We found that under normal conditions, Lamp1 expression level and pattern between cells are not uniform (Fig. R2a). We apologize that the images we presented raise such concerns. To address the concerns raised by siArl4A, we also confirmed that the total amount of Lamp 1 protein is the same between siCtrl and siArl4A cells (Fig. R2b). Therefore, we will include the images with more comparable Lamp1 as new representative images in Supplementary Figs. 10-12 to avoid misunderstanding.

Fig.R2. (a) Lamp1 expression is not uniform in cells under normal conditions. A549, H1975, and PE089 cells were stained with anti-Lamp1 (red) and DAPI (blue). (b) The C33-A cells were transfected with either the control or Arl4A siRNA. Cell lysates were analyzed by immunoblotting with anti-EEA1, anti-Lamp1, anti-Arl4A, and anti- α -tubulin antibodies.

Finally, the concern about the use of only one siRNA targeting Arl4A (or rescue experiments) still remains as in the revised version the authors only added a second siRNA targeting Arl4 in one experiment, to confirm that it reduces EGFR expression.

Response: We thank the reviewer for the comment. We have included the Arl4 #2 siRNA for the rescue experiments in the 2nd revised manuscript (Supplementary Fig. 4a). In addition, we tested the effect of Arl4A#2 siRNA in ligand-induced receptor degradation pathway. We found that the half-life of EGFR was decreased as the effect resulted from siArl4A mainly used in the study by transfection with Arl4A#2 siRNA (Supplementary Fig. 4e).

REVIEWERS' COMMENTS

Reviewer #1 (Remarks to the Author):

The authors have now addressed my queries and I am happy to recommend publication of the manuscript.

Reviewer #2 (Remarks to the Author):

The revised manuscript is clearly improved. Although the level of mechanistic insight remains somewhat limited, the numerous lines of evidence shown are strongly suggestive that Arl4 regulates cargo sorting through an ability to interact with and regulate ESCRT-II. I anticipate members of the community will find these results intriguing and investigate further the mechanisms by which Arl4 directs ESCRT-II activity, which will move the field forward.

Reviewer #4 (Remarks to the Author):

I appreciate that the authors tried to address most of the reviewers' comments. However, my previous concern regarding the colocalization analysis with Lamp1 shown in Suppl. Fig.10b remains unfortunately unanswered. Why the intensity of Lamp1 signal differs over time (and in a different way between control and Arl4A silenced A549 cells)? How does this issue affect the colocalization analysis?